# KALEIDOSCOPE: IN-LANGUAGE EXAMS FOR MASSIVELY MULTILINGUAL VISION EVALUATION

Israfel Salazar[* 2]    Manuel Fernández Burda[* 3]    Shayekh Bin Islam[* 4,31]
Arshia Soltani Moakhar[* 4]    Shivalika Singh[* 1]    Fabian Farestam[* 17]
Angelika Romanou[* 30]    Danylo Boiko[4,9]    Dipika Khullar[4]    Mike Zhang[2,10,28]
Dominik Krzemiński[4]    Jekaterina Novikova[4,26]    Luísa Shimabucoro[18]
Joseph Marvin Imperial[19,27]    Rishabh Maheshwary[4]    Sharad Duwal[4]
Alfonso Amayuelas[5]    Swati Rajwal[6]    Jebish Purbey[4,7]    Ahmed Ruby[25]
Nicholas Popovič[11,12]    Marek Suppa[22,23]    Azmine Toushik Wasi[4]
Ram Mohan Rao Kadiyala[7,8]    Olga Tsymboi[13]    Maksim Kostritsya[15,16]
Bardia Soltani Moakhar[4]    Gabriel da Costa Merlin[18]    Otávio Ferracioli Coletti[18]
Maral Jabbari Shiviari[4]    MohammadAmin Farahani Fard[4]    Silvia Fernandez[4,33]
María Grandury[21]    Dmitry Abulkhanov[4]    Drishti Sharma[4]
Andre Guarnier De Mitri[18]    Leticia Bossatto Marchezi[20]    Setayesh Heydari[4]
Johan Obando-Ceron[4,24,29]    Nazar Kohut[14]    Beyza Ermis[1]
Desmond Elliott[† 2,28]    Enzo Ferrante[† 3,32]    Sara Hooker[† 1]    Marzieh Fadaee[† 1]

[1]Cohere Labs    [2]University of Copenhagen    [3]CONICET & Universidad de Buenos Aires
[4]Cohere Labs Community    [5]UC Santa Barbara    [6]Emory University    [7]ZeroGrad
[8]Traversaal.ai    [9]Taras Shevchenko National University of Kyiv    [10]Aalborg University
[11]Karlsruhe Institute of Technology    [12]ScaDS.AI, TU Dresden    [13]T-Tech
[14]Lviv Polytechnic National University    [15]HSE University    [16]HiveTrace    [17]ETH Zürich
[18]University of São Paulo    [19]National University Philippines    [20]Federal University of São Carlos
[21]SomosNLP    [22]Cisco    [23]Comenius University in Bratislava    [24]Mila – Québec AI Institute
[25]Uppsala University    [26]Vanguard    [27]University of Bath    [28]Pioneer Center for AI
[29]Université de Montréal    [30]EPFL    [31]KAIST    [32]Anyone AI    [33]AI Circle

Correspondence: israfel.salazar@di.ku.dk, mburda@dc.uba.ar, marzieh@cohere.com

## ABSTRACT

The evaluation of vision-language models (VLMs) has mainly relied on English-language benchmarks, leaving significant gaps in both multilingual and multicultural coverage. While multilingual benchmarks have expanded, both in size and language, many rely on translations of English datasets, failing to capture cultural nuances. In this work, we propose KALEIDOSCOPE, as the most comprehensive exam benchmark to date for the multilingual evaluation of vision-language models. KALEIDOSCOPE is a large-scale, in-language multimodal benchmark designed to evaluate VLMs across diverse languages and visual inputs. KALEIDOSCOPE covers 18 languages and 14 different subjects, amounting to a total of 20,911 multiple-choice questions. Built through an open science collaboration with a diverse group of researchers worldwide, KALEIDOSCOPE ensures linguistic and cultural authenticity. We evaluate top-performing multilingual vision-language models and find that they perform poorly on low-resource languages and in complex multimodal scenarios. Our results highlight the need for progress on culturally inclusive multimodal evaluation frameworks.

## 1 INTRODUCTION

Evaluations are the backbone of measuring progress in machine learning, yet many benchmarks – especially for language models – remain English and Western-centric (Joshi et al., 2020; Fan et al.,

---

* First authors    † Senior authors

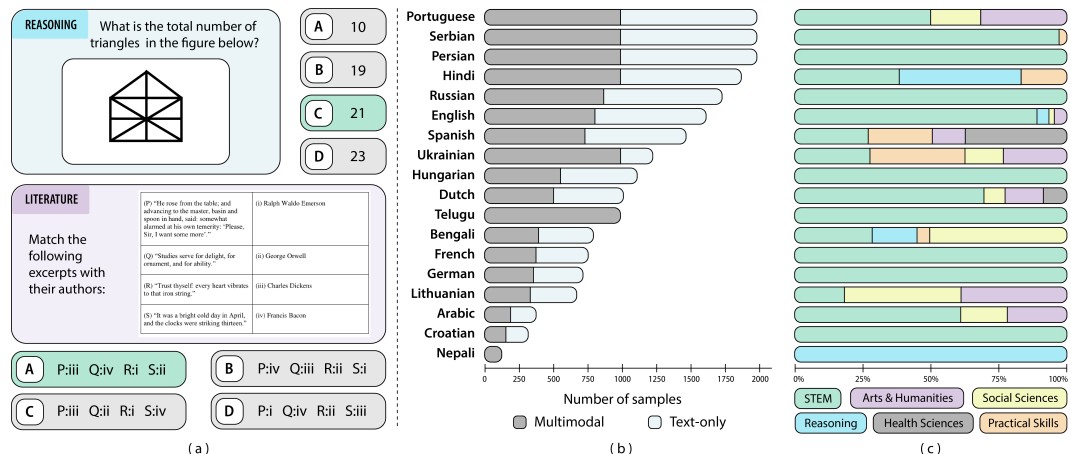

Figure 1: **Overview of the** KALEIDOSCOPE **Benchmark.** (a) Multilingual-Multimodal MCQ Samples (b) Language and Multimodal Samples Distribution. (c) Exam Category Breakdown.

2020; Dodge et al., 2021; Liu et al., 2021; Chung et al., 2022; Gehrmann et al., 2022; Lucy et al., 2024). This imbalance becomes even more striking at frontier of AI, where generative models are rapidly expanding into multimodal territory (OpenAI et al., 2024; Google et al., 2024; Anthropic, 2024; Deitke et al., 2024; Yue et al., 2025; Qwen-Team, 2025), seeking to represent a richer world made up of different modalities such as image, text, sound. In recent years, the community has made promising strides toward broader multilingual text evaluation (Ahuja et al., 2023; Singh et al., 2024b;a; Aakanksha et al., 2024; Pozzobon et al., 2024; Romanou et al., 2024; Singh et al., 2025; Adelani et al., 2024), and multimodal benchmarks are starting to take shape (Bugliarello et al., 2022; Fu et al., 2023; Yue et al., 2024a;b; Li et al., 2024a; Xu et al., 2025). Yet reliable evaluation at the intersection of multilingual and multimodal tasks remains rare. This gap motivates our work.

A common but imperfect solution is translating English benchmarks into other languages. While convenient, this often falls short of capturing cultural context and nuance. Translated datasets can easily reinforce Western-centric knowledge and assumptions (van Miltenburg et al., 2017; Frank et al., 2018; Singh et al., 2025; Longpre et al., 2025) limiting their ability to assess performance across diverse settings. Moreover, automated data curation pipelines frequently amplify existing quality issues (Luccioni & Viviano, 2021; Caswell et al., 2020; Kreutzer et al., 2022), with translation artifacts such as *translationese* muddying evaluations (Koppel & Ordan, 2011; Zhang & Toral, 2019; Bizzoni et al., 2020; Vanmassenhove et al., 2021). While translated data has its place, especially for some particularly low-resource tasks (Zhou et al., 2021; Thapliyal et al., 2022; Qiu et al., 2022; Ramos et al., 2024; Geigle et al., 2025; Dang et al., 2024; Üstün et al., 2024; Aakanksha et al., 2024), it is an imperfect substitute for genuinely diverse, in-language benchmarks.

In this work, we introduce the largest benchmark of real-world, in-language exam questions blending image and text modalities. Our dataset pushes beyond simple captioning, challenging models to reason about visual content in various topics, the way humans are evaluated in exams worldwide. Through a large-scale open science effort across 18 languages, we construct KALEIDOSCOPE (see Figure 1), featuring a diverse selection of knowledge domains across 14 subjects. With 55% of the total 20,911 questions requiring image understanding for accurate resolution, our work establish a comprehensive, and inclusive evaluation framework for multimodal language models. We evaluate a wide range of state-of-the-art models on KALEIDOSCOPE, including Claude 3.5 Sonnet (Anthropic, 2024), GPT-4o (OpenAI et al., 2024), and Gemini-V (Google et al., 2024), as well as smaller open-weight VLMs, such as Aya-Vision model family (Cohere-For-AI-Team, 2025), Molmo (Deitke et al., 2024) Pangea (Yue et al., 2025), and Qwen2.5-VL model family (Qwen-Team, 2025). Our key contributions and findings are highlighted here:

KALEIDOSCOPE **Benchmark**: We present the largest multilingual multimodal exam set, covering high resource (e.g., English, Spanish) to underrepresented languages (e.g., Bengali, Telugu) across diverse subjects from sociology to STEM. Most languages (10/18) include 5+ topics, with the rest focusing on multi-subtopics like mathematics or engineering. Questions emphasize vision grounded

reasoning through tasks like interpreting graphs, pictures, and region-specific diagrams, supported by fine-grained metadata for model diagnostics.

**Modality-Specific Performance Disparities:** All models perform substantially better on text-only questions, revealing a clear disparity across modalities. The gap widens in larger modelsl; for instance, GPT-4o shows a 21.6% difference between text-only and multimodal performance, while smaller models like Molmo exhibit a much narrower gap of 3.69%. (Section 4.1). Furthermore, multimodal performance varies significantly by visual data type: models are more capable of answering questions about tables (76.5%) and photographs (81.5%) compared to diagrams (62.9%).

**Domain-Specific Performance Disparities:** We observe a significant performance gap between questions requiring knowledge of Humanities & Social Sciences and those focused on STEM subjects (Section 4.4). On average, models present accuracy of 83.7% for humanities versus 59.2% for STEM (based on the best scores across models). Models struggle more with STEM questions, suggesting that while they can often recognize visual content and retrieve related knowledge, they lack the reasoning capabilities needed to arrive at the correct answers in STEM domains.

**Crosslingual Performance Disparities:** Model performance varies across languages, with better results in high-resource languages and weaker performance in mid- and low-resource ones (Section 4.3). Crosslingual transfer appears to play a role, as models perform better on average in languages using Latin scripts compared to those with non-Latin scripts.

## 2 THE KALEIDOSCOPE BENCHMARK

The KALEIDOSCOPE Benchmark is a global collection of multiple-choice questions sourced from real-world exams, with the goal of evaluating multimodal and multilingual understanding in VLMs. The collected exams are in a Multiple-choice question answering (MCQA) format which provides a structured framework for evaluation by prompting models with predefined answer choices (Hendrycks et al., 2021; Lu et al., 2023; Wang et al., 2024a; Yue et al., 2024a; Romero et al., 2024; Romanou et al., 2024), closely mimicking conventional human testing methodologies. Our work is built around three core design principles that guide the selection, curation, processing, and addition of exams:

🖼 **Multimodality**: Images are central to KALEIDOSCOPE, as we aim to evaluate how VLMs integrate and reason about visual information to answer questions. We prioritize multimodal questions with diverse image types, complemented by a similar proportion of text-only questions for a complete assessment and comparison.

🌐 **Multilinguality**: The benchmark contains questions in 18 languages, with a focus on under-represented mid- and low-resource languages (e.g., Nepali, Lithuanian) alongside high-resource languages (e.g., English, Spanish) for a thorough evaluation across a broad range of languages.

👥 **Diversity**: Our goal is to collect exams covering as wide a range of topics as possible ranging from Mathematics and Sociology, to Medicine and Driving Licenses, ensuring comprehensive evaluation across various domains. The final collection includes exams from 14 different domains, collected from 18 countries and with varying educational levels, allowing detailed clustering and comprehensive evaluation.

### 2.1 GLOBAL COLLABORATION

Our work entailed an extensive, open science process to manually collect data by working directly with native speakers of different languages (Elliott et al., 2016; Liu et al., 2021; Thapliyal et al., 2022; Li et al., 2024c; Üstün et al., 2024; Singh et al., 2024b). This is acutely needed in the field of machine learning, where recent studies have highlighted that dataset creators remain predominantly Western-centric (Longpre et al., 2025). The manual curation of datasets is a costly process that requires careful attention to detail in every language to ensure high-quality, contextually relevant content for evaluation. In this work, we engage in a large-scale open science collection process, which brings together contributors spanning 20 nations across four continents to ensure linguistic and cultural authenticity. For related participatory research see Appendix C.1.

## 2.2 DATA PIPELINE

**Collection:** We collected KALEIDOSCOPE following guidelines on type of exams and questions required, formatting, specifications, and quality control measures. Data was collected through a global call for contributions and distributed across global communities, with the majority of contributors being independent researchers in the open science community. This effort resulted in 20,911 questions from 18 countries and languages, sourced in their original languages to maintain linguistic authenticity. We prioritized original, domain-expert-written questions (e.g., from teachers), ensuring real-world relevance and quality. The exams were gathered from various repositories, including official government websites, question banks, and other publicly available repositories with educational materials. Throughout the process, contributors also annotated associated licenses with each dataset to allow for documentation of data provenance (Longpre et al., 2024).

**Processing:** The annotation process involves two stages. First, we perform automated parsing and extraction. For directly parsable text, we use PDF or web parsers, while for non-parsable text, we employ OCR API's, such as Mathpix[1], along with vision-language models such as GPT-4o. These tools allow us to extract both text and image elements from exam source formats, which are then converted into structured outputs in LaTeX, Markdown, and JSON formats, as required. Since automated parsing can sometimes result in misaligned images and text, in the second stage we refine the extracted text. Applying heuristic rules, as well as high-performing LLMs (Claude 3.5 Sonnet and GPT-4o), we restructure the output, ensuring proper alignment of questions, text, and answer choices. Human verification follows, ensuring images are correctly linked to the corresponding questions, and checking that extracted formulas match the expected equation format.

**Quality Assessment:** Maintaining reliable and high quality data is essential, especially given the large-scale international collaboration in this project. To ensure integrity, we include manual validation in three stages of the collection and annotation pipeline. First, at the end of the collection stage, two independent annotators validate each exam to ensure conformity with the guidelines. We include a strict revision to confirm compliance with the distribution license requirements. Only exams approved by both independent annotators are included in the dataset. Next, following the annotation process, a validation script checks for JSON formatting errors, duplicates, and malformed strings that do not conform to identified entry specifications (see Appendix E.3). Finally, at the last stage, two separate validators perform a final manual review of the collected files before merging them into KALEIDOSCOPE.

Quality control also extends to the evaluation, where we analyze the most prominent failure modes. During inference, suspicious outputs, such as ambiguous answers, no response, or consistent failures across models, are flagged for manual review. If an issue is identified, the entire exam containing the problematic question is reviewed for correction or removal. This process guarantees that any errors in the benchmark questions are identified and addressed, further enhancing the reliability of the dataset.

## 2.3 DATA STATISTICS

The final KALEIDOSCOPE benchmark contains 20,911 questions across 18 languages belonging to 8 language families. A total of 11,459 questions require an image to be answered (55%), while the remaining 9,452 (45%) are text-only. The dataset covers 14 different subjects, grouped into 6 broad domains. Figure 1 presents an overview of the dataset; detailed statistics can be found in Appendix B.3. The majority of questions in KALEIDOSCOPE are multimodal, with the exact proportion varying across languages, ranging from 50% to 100%, with some languages always requiring images for resolution.

Each exam question contains 17 fields, including source country, language, license, educational level, category, and multimodal information. These fields are detailed in Appendix E.3. The questions are formatted in MCQA format with 4 options and a single correct answer. The subject is labeled in both English and the source language. The educational level (e.g., high school, university entrance, professional licensing) is also included to ensure diverse representation. Multimodal questions additionally specify the type of image, such as graphs, tables, or diagrams. Additionally, each entry includes metadata such as source details, licensing status, and ISO 639-1 language codes. For a fine-grained analysis, each question includes detailed metadata, with examples provided in Appendix E.1.

---

[1] https://mathpix.com/convert

The metadata allows us to evaluate how visual and textual elements interact in multimodal reasoning tasks, making the benchmark valuable for evaluating models across diverse scenarios.

KALEIDOSCOPE covers a wide range of languages, including low- and mid-resource languages such as Nepali, Lithuanian, Bengali, Telugu, Persian, Ukrainian, Croatian, Serbian, and Hungarian, as well as high-resource languages such as English, Spanish, Portuguese, Russian, French, German, Arabic, Hindi, and Dutch. This selection allows us to evaluate how performance is affected by the amount of resources available for a given language. The dataset spans 8 different language families, providing a broad linguistic range. The number of questions per language varies significantly, from 126 for Nepali to 2000 for Portuguese, Serbian, and Persian. The linguistic diversity present in KALEIDOSCOPE enables a robust evaluation of models across both widely spoken and underrepresented languages, making the dataset suitable for comprehensive multilingual assessment.

## 3 EXPERIMENTAL SETUP

### 3.1 MODELS

We benchmark both open-weights and closed multimodal vision-language models on KALEIDOSCOPE, focusing on lighter open-weight models and larger closed models to assess performance across a wide range of model sizes. The open-weight models[2] include Aya-Vision-8B and 32B (Cohere-For-AI-Team, 2025), Molmo-7B-D (Deitke et al., 2024), Pangea-7B (Yue et al., 2025), and all sizes of Qwen2.5-VL-Instruct (Qwen-Team, 2025) (3B, 7B, 32B, and 72B) to analyze the impact of model scale on KALEIDOSCOPE. All models have image and multilingual support; Aya-Vision supports 23 languages, Qwen2.5-VL supports 29 languages, and Pangea was trained on a dataset spanning 39 different languages, making them strong candidates for multimodal and multilingual evaluation. For the closed models, we evaluate GPT-4o (OpenAI et al., 2024) (2024/08/06), Claude 3.5 Sonnet (Anthropic, 2024) (2024/10/22), and Gemini 1.5 Pro (Google et al., 2024).

### 3.2 EVALUATION SETUP

We designed two distinct evaluation setups to accommodate VLMs' varying reasoning and instruction-following capabilities. For closed models, we employed zero-shot prompts using the Chain-of-Thought (CoT) method (Wei et al., 2022), instructing the model to reason step-by-step before selecting the final answer within specific `<ANSWER> </ANSWER>` tags—a natural approach aligned with real-world MCQ applications. Using a common template (see Appendix D.5), we ensured equal evaluation conditions across models, with instructions translated into all evaluated languages for a fully in-language setup, following Romanou et al. (2024). For smaller open-weight models, which showed limited CoT effectiveness in preliminary experiments (Appendix D.2), we instead used a direct answer generation approach, prompting models to output their choice in a JSON-structured `{'choice': ...}` field. This simplified the task by reducing reasoning or formatting errors, with instructions always in English regardless of question language. Further discussion on model output errors is in section 5. Due to KALEIDOSCOPE's evaluating nature, we employ accuracy as our main metric (further evaluation metrics details can be found in Appendix D.1).

## 4 RESULTS

### 4.1 OVERALL PERFORMANCE

We benchmark a wide variety of models on KALEIDOSCOPE, with results summarized in Table 1. Claude 3.5 Sonnet and Gemini 1.5 Pro lead among closed models, while GPT-4o's performance is impacted by high format errors—particularly in the multimodal split—though its accuracy improves significantly when considering only valid answers (see Section 5). Among open-weight models, Qwen2.5-VL-72B achieves the highest accuracy, followed by lightweight models like Qwen2.5-VL-7B, which leads the 7-8B category.

---

[2] All open-weight models are evaluated locally using 1×NVIDIA Ampere A100 GPU with 64GB of memory for models up to 8B, and 4×A100 for models on the range 32B–72B. Closed models were accessed via API. To ensure a consistent evaluation environment, we set the temperature to 0.7, the maximum token generation to 1024, and the image size to 512×512 for all models.

Table 1: **Performance Evaluation on** KALEIDOSCOPE. Results are reported as macro-averaged accuracy (%) across all languages (equal weight per language). **Acc.**: Accuracy over all samples; **F.E.**: Format Error rate (invalid responses); **Valid Acc.**: Accuracy excluding invalid responses. Metrics are shown for the full dataset (**Overall**), multimodal inputs (**Multimodal**), and text-only inputs (**Text-only**).

| | Overall | | | Multimodal | | | Text-only | | |
|---|---|---|---|---|---|---|---|---|---|
| | | Valid Responses | | | Valid Responses | | | Valid Responses | |
| Model | Acc. | F.E. | Acc. | Acc. | F.E. | Acc. | Acc. | F.E. | Acc. |
| Claude 3.5 Sonnet | **62.91** | 1.78 | **63.87** | **55.63** | 3.24 | **57.24** | **73.54** | 0.02 | **73.57** |
| Gemini 1.5 Pro | 62.10 | 1.62 | 62.95 | 55.01 | 1.46 | 55.71 | 72.35 | 1.81 | 73.45 |
| GPT-4o | 58.32 | 6.52 | 62.10 | 49.80 | 10.50 | 55.19 | 71.40 | 1.71 | 72.39 |
| Qwen2.5-VL-72B | 52.94 | 0.02 | 53.00 | 48.40 | 0.03 | 48.41 | 60.00 | 0.02 | 60.01 |
| Aya-Vision-32B | 39.27 | 1.05 | 39.66 | 35.74 | 1.49 | 36.28 | 44.73 | 0.51 | 45.00 |
| Qwen2.5-VL-32B | 48.21 | 0.88 | 48.64 | 44.90 | 0.28 | 45.05 | 53.77 | 1.61 | 54.60 |
| Aya-Vision-8B | 35.09 | 0.07 | 35.11 | 32.35 | 0.05 | 32.36 | 39.27 | 0.10 | 39.30 |
| Molmo-7B-D | 32.87 | 0.04 | 32.88 | 31.43 | 0.06 | 31.44 | 35.12 | 0.01 | 35.13 |
| Pangea-7B | 31.31 | 7.42 | 34.02 | 27.15 | 13.52 | 31.02 | 37.84 | 0.03 | 37.86 |
| Qwen2.5-VL-7B | 39.56 | 0.08 | 39.60 | 36.85 | 0.04 | 36.88 | 43.91 | 0.11 | 43.96 |
| Qwen2.5-VL-3B | 35.56 | 0.19 | 35.63 | 33.67 | 0.32 | 33.79 | 38.51 | 0.03 | 38.53 |

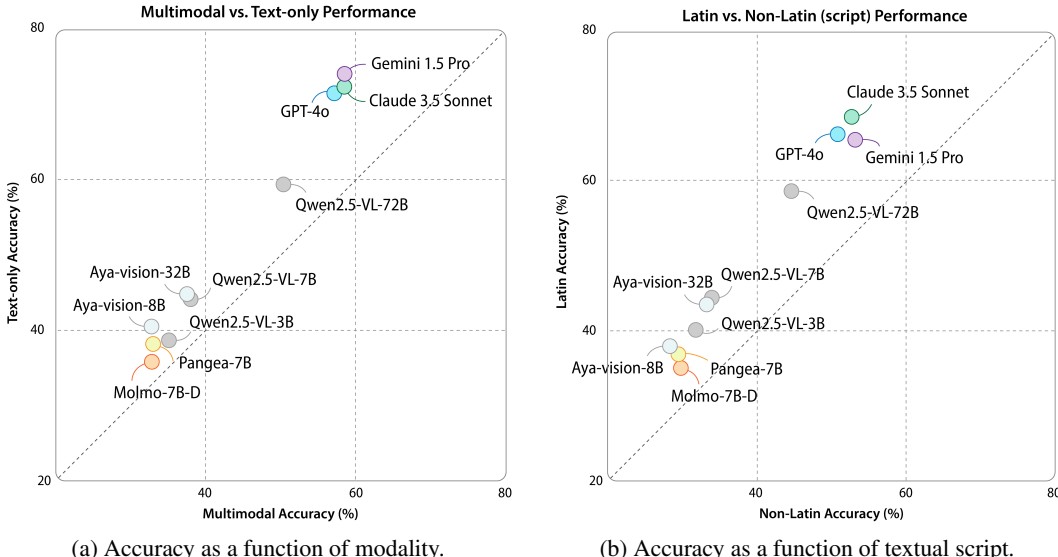

(a) Accuracy as a function of modality.  (b) Accuracy as a function of textual script.

Figure 2: **Model Performance Analysis on** KALEIDOSCOPE. (a) Accuracy (%) of models on multimodal and text-only questions, highlighting low performance on multimodal samples. (b) Accuracy (%) by script type, revealing biases for latin scripts. Accuracy over valid responses is used to generate both figures. Identity line is added to show parity.

A key trend is the performance drop in multimodal questions compared to text-only ones, with closed models showing the largest gaps (e.g., GPT-4o's steep decline). In contrast, open-weight models exhibit smaller gaps, suggesting greater robustness across modalities despite lower overall scores. This gap narrows further for smaller models, with Molmo displaying the most balanced performance. This lightweight model's consistency is depicted in Figure 2a (by being closer to the identity line), reinforcing that open models while less specialized, handle multimodal tasks more uniformly.

Table 2: **Model Performance Breakdown by Image Type in** KALEIDOSCOPE. Accuracy (%) over valid answers across image type. Bold values indicate top-performing model.

| Model | Diagram (2,182) | Figure (6,178) | Graph (733) | Map (392) | Photo (631) | Formula (487) | Table (597) | Text (257) |
|---|---|---|---|---|---|---|---|---|
| Claude 3.5 Sonnet | **62.9** | 50.5 | **74.2** | **80.1** | 77.8 | 52.1 | 75.0 | 85.2 |
| Gemini 1.5 Pro | 59.4 | **51.3** | 67.9 | 69.4 | 75.8 | **68.3** | 76.0 | 85.2 |
| GPT-4o | 59.6 | 48.2 | 68.4 | 78.8 | **81.5** | 64.4 | **76.5** | **86.2** |
| Qwen2.5-VL-72B | 51.1 | 43.9 | 59.4 | 66.1 | 70.5 | 48.7 | 61.5 | 86.0 |
| Aya-Vision 32B | 38.6 | 33.4 | 42.0 | 50.0 | 60.2 | 32.4 | 33.1 | 68.8 |
| Qwen2.5-VL-32B | 46.7 | 41.0 | 53.1 | 58.2 | 65.0 | 47.3 | 58.0 | 82.5 |
| Aya-Vision 8B | 32.7 | 29.9 | 37.2 | 38.6 | 42.3 | 29.2 | 34.1 | 54.9 |
| Molmo-7B-D | 30.3 | 31.5 | 36.7 | 37.8 | 45.0 | 25.1 | 30.6 | 56.8 |
| Pangea-7B | 31.0 | 31.0 | 32.9 | 38.5 | 45.0 | 32.2 | 29.4 | 66.3 |
| Qwen2.5-VL-7B | 38.0 | 34.0 | 44.3 | 48.0 | 53.9 | 34.9 | 40.9 | 76.3 |
| Qwen2.5-VL-3B | 32.8 | 32.3 | 40.2 | 41.2 | 48.2 | 34.7 | 35.2 | 72.8 |

## 4.2 NOT ALL IMAGE TYPES ARE EQUAL

KALEIDOSCOPE contains eight visual information types, with accuracy varying significantly by complexity (Table 2). Simpler inputs like text-rich images (Qwen2.5-VL-7B: 76.3%; GPT-4o: 86.2%) and photos score higher than technical categories like Formulas and Diagrams (Qwen2.5-VL-7B: 38.0%; GPT-4o: 62.9%). Notably, Qwen2.5-VL-72B ranks second in text-rich images, surpassing both Gemini and Claude. Larger models show specialized strengths: Gemini 1.5 Pro dominates Formulas and Figures, GPT-4o leads in text-rich images, and Claude 3.5 Sonnet achieves the highest scores in Diagrams, Graphs, and Maps. In contrast, Qwen2.5-VL-7B consistently outperforms all lightweight models across categories, demonstrating broader capability despite lower absolute scores. The results reveal a clear hierarchy: models handle simple visuals well but struggle with structured or symbolic data, a pattern consistent across architectures but more pronounced in smaller models.

## 4.3 RESOURCE AND SCRIPT SENSITIVITY IN MODELS

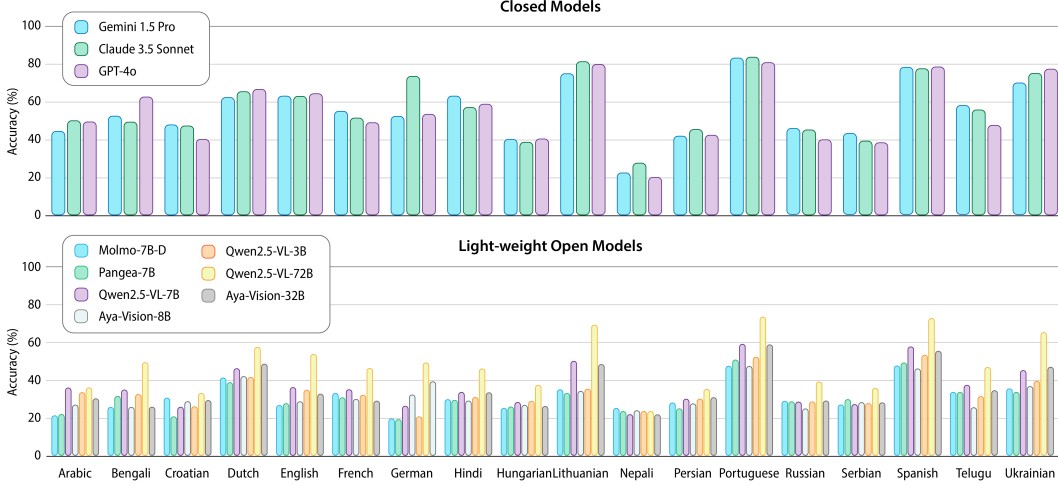

Figure 3: **Multimodal Accuracy by Language in** KALEIDOSCOPE. Reports performance (accuracy %) for closed models and open-weight models on multimodal questions.

Performance in KALEIDOSCOPE varies widely across all 18 languages (see Figure 3). Models generally perform well in high-resource languages (e.g., English, Spanish, German) but struggle with lower- and mid-resource ones, such as Nepali and Telugu. This can be attributed to the limited

training data for these languages, complex scripts, and the exclusive use of multimodal samples for these languages (see Appendix B.3), which are inherently more challenging. Lithuanian, despite being mid-resource language, stands out as the highest-performing language, with Claude 3.5 Sonnet leading in accuracy. This might be due the fact that all Lithuanian questions belong to `College Graduation Exams`, and have a major subject composition of Social Sciences and Humanities in opposition to STEM subjects, which may align well with the models' capabilities. Closed models show similar performance within each language, except for German, where Claude excels. In contrast, Qwen2.5-VL-7B consistently leads all lightweight models for almost every language, and the heavier Qwen2.5-VL-72B shows the benefits of model scale.

The results show that all models are biased towards Latin script languages. As shown in Figure 2b, all models are above the parity line, exhibiting consistent higher performance for Latin scripts compared to non-Latin scripts. Full results can be found in Appendix D.3.

### 4.4 STEM Questions Expose Model Deficiencies

KALEIDOSCOPE consists of exams covering 14 subjects and domains. We observe that all models perform significantly better on Humanities & Social Science questions compared to other domains. The closed models achieve high accuracy in areas like Sociology (Claude: 93.4%, GPT-4o: 93.2%), Social Sciences (GPT-4o: 88.1%, Gemini: 85.7%), and Language (GPT-4o: 85.8%, Claude: 85.5%). In contrast, performance in STEM subjects, including Mathematics, Physics, and Engineering, is notably lower, with most models scoring below 50%. This suggests that while they are generally capable of recognizing visual content and retrieving surface-level knowledge, they fall short when it comes to performing the multi-step reasoning and problem-solving required in STEM subjects. Answering these questions often demands not just factual recall but also the ability to interpret complex diagrams, apply mathematical concepts, and reason through scientific principles – capabilities that current models have yet to fully master. This highlights a key gap in their ability to bridge perception and reasoning, particularly in tasks that require deeper analytical thinking. Refer to Appendix D.3, Table 9, for multimodal and complete results of model performance across subjects.

## 5 Analysis

### 5.1 How Sensitive Are VLMs to Missing or Incorrect Images?

Table 3: **Image Relevance Analysis for Qwen2.5-VL-7B on** KALEIDOSCOPE. Model performance across the standard multimodal, *Random Image*, and *No-Image* setups to assess the impact of visual information on question-answering accuracy.

| Setup | Accuracy | Valid Responses | |
| --- | --- | --- | --- |
| | | Format Error | Accuracy |
| Standard Multimodal | 36.85 | 0.04 | 36.88 |
| Random Image | 32.56 | 3.12 | 33.53 |
| No Image | 33.44 | 0.03 | 33.45 |

To evaluate the dependency of multimodal questions on images, and the impact of incorrect image associations, we conducted an experiment using the multimodal split of KALEIDOSCOPE. Following Elliott (2018); Thomason et al. (2019), we created two modified versions of the dataset: (1) a *'No Image'* split, where all images were removed, and (2) a *'Random Image'* split, where images were randomly reassigned to questions. The aim of this experiment is to assess how much the models rely on the visual information. We evaluate the performance of Qwen2.5-VL-7B on these modified splits, and the results are shown in Table 3.

We observe that the model performs above the random baseline (25%) across all three splits, indicating some ability to reason from text alone. However, there is a drop in performance ($-3.41\%$ in Total Accuracy) when questions are presented without images, suggesting that the model does rely on visual information for accurate answers. The performance drop is similar for both modifications; however, we observe a significantly larger format error when the model is tested with irrelevant images. In several of these cases, the model actually acknowledges that the image does not correspond to the

question. In contrast, in experiments with no images, the format error rate is almost zero, indicating that the model attempts to answer even when visual inputs are missing.[3]

## 5.2 Scaling Model Size Improves Performance

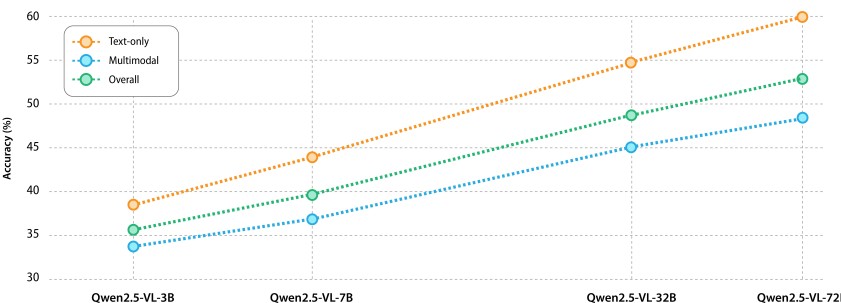

Figure 4: **Model Size Analysis for Qwen2.5-VL Models.** Performance improvement across three model sizes (3B, 7B, 32B, and 72B parameters) on KALEIDOSCOPE's multimodal tasks, demonstrating consistent gains from increased model capacity. Note that x-axis is shown in log-scale.

To analyze the impact of model size on KALEIDOSCOPE performance, we evaluated all four variants of Qwen2.5-VL. We selected this model family for its well-distributed size range, as well as being the best performing model in the open weight model category. We follow the same experimental setup for all model versions.

Figure 4 shows the performance of Qwen2.5-VL variants on KALEIDOSCOPE. Model size is shown in the x-axis (log-scale), while the y-axis displays accuracy for multimodal and text-only splits, and overall score. We observe a linear relationship between the logarithm of the model size and accuracy, with larger models showing significant gains. The largest open model model evaluated, Qwen2.5-VL-72B, still underperforms the closed models, however, these results highlight the effectiveness of scaling for open models, with clear and predictable improvements at each size tier.

## 5.3 Format Errors

While our experimental setup ensures a majority of answers were extracted from model outputs, we observe occasional failures: models struggle to follow instructions, outputs contain formatting errors, or models refuse to answer (particularly for health-related or ethical questions). Appendix Figure 5 shows that unanswered questions concentrate in mid- to low-resource languages, and the distribution accumulates over non-latin scripts, likely due to tokenization challenges, insufficient language-specific training data, or visual-textual alignment difficulties. Pangea-7B shows the highest refusal rates, especially for Telugu (452), Hindi (130), Persian (160), and Serbian (125). While other open models show minimal unanswered counts, indicating better format adherence. Closed models (Claude 3.5 Sonnet, GPT-4o) display distinct behavior: their refusals concentrate on non-Latin, low-resource languages, but they also show high error rates for English questions, primarily health/medical queries due to policy constraints. This underscores the trade-off between content moderation and benchmark performance.

## 6 Related Work

While VLMs excel in multimodal tasks, existing benchmarks (Li et al., 2024b; Vayani et al., 2024; Nayak et al., 2024; Schneider et al., 2025) predominantly evaluate high-resource languages (e.g., English (Zang et al., 2024; Schneider et al., 2025), Chinese (Fu et al., 2023; He et al., 2024)), neglecting linguistic diversity and cultural nuances (Hengle et al., 2024; Bird, 2022). Translating benchmarks via tools like ChatGPT (Lai et al., 2023), GPT-4 (Yue et al., 2025) or Google Translate (Li et al., 2023),

---

[3]We observed that Qwen2.5-VL-7B tends to hallucinate when no image is present. In a simple experiment using the prompt "`Describe the following image`", the model correctly describes the input image when provided. However, when no image is passed, the model hallucinates and generates a random description.

often introduces errors and cultural mismatches (Singh et al., 2024a; Huang et al., 2025). Recent multilingual benchmarks attempt to bridge this gap: MMLU-ProX (Xuan et al., 2025) covers reasoning in 13, CVQA (Romero et al., 2024) integrates cultural visuals across 31, and PangeaBench aggregates 47 languages (Yue et al., 2025). However, cultural benchmarks like MaRVL (binary evaluation) (Liu et al., 2021) and CULTURALVQA (English-only open-ended questions) (Nayak et al., 2024) remain limited in scope or format. Moreover, the MaXM benchmark (Changpinyo et al., 2023) addresses bias and provides multilingual, multimodal assessment across 7 languages but does not focus on cultural aspects, an area where KALEIDOSCOPE offers added value. KALEIDOSCOPE advances these efforts by combining regionally sourced multimodal exam questions with MCQA structure, enabling granular, culturally conscious evaluation across 18 languages. Exam-style benchmarks assess VLMs under structured multilingual settings. M3Exam (Zhang et al., 2023) uses real exams in 9 languages but only 23% image-dependent questions. EXAMS-V (Das et al., 2024) spans 11 languages with multimodal STEM content, yet 75% of its 20,932 questions are text-only. While M5 (Schneider & Sitaram, 2024) evaluates 41 languages across vision-language tasks, it avoids MCQA formats. KALEIDOSCOPE surpasses these by combining 18 languages, STEM & cultural coverage, and 55% image-dependent MCQA (Table C.2), offering comprehensive multilingual exam evaluation.

## 7  CONCLUSION

As generative models become increasingly multimodal and multilingual, the need for robust and culturally grounded evaluation benchmarks has never been more urgent. We take a step toward closing this gap by introducing the largest benchmark of real-world, in-language multimodal exam questions. By grounding evaluation in authentic exam settings from around the world, our benchmark challenges models to reason about images in ways that mirror human assessment, capturing both linguistic and cultural complexity. Our findings highlight the limitations of current models in handling this intersection of skills: multilingual understanding, visual reasoning, and culturally aware problem-solving. We hope this benchmark serves not only as a valuable tool for measuring progress but also as a call to action for developing models that are truly capable of operating across languages, cultures, and modalities. Continued investment in representative, high-quality evaluation datasets will be essential to ensure that future AI systems are equitable and globally relevant.

## 8  ACKNOWLEDGMENTS

We acknowledge the EuroHPC Joint Undertaking for awarding this project access to the EuroHPC supercomputer LEONARDO, hosted by CINECA (Italy) and the LEONARDO consortium through an EuroHPC Development Access call (ID:EUHPC_D12_071). This work was supported by research grant (VIL53122) from Villum Fonden, and by the European Union's Horizon 2020 research and innovation program under grant agreement No. 101135671 (TrustLLM). EF gratefully acknowledges the support of the Googler Initiated Grants and the Google Award for Inclusion Research programs. MZ is supported by the research grant (VIL57392) from Villum Fonden and also received funding from the Danish Government to Danish Foundation Models (4378-00001B). JMI is supported by the National University Philippines and the UKRI Centre for Doctoral Training in Accountable, Responsible, and Transparent AI [EP/S023437/1] of the University of Bath. MS is funded by the EU NextGenerationEU through the Recovery and Resilience Plan for Slovakia under the project No. 09I02-03-V01-00029.

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

APPENDIX

APPENDIX CONTENTS

## A   LIMITATIONS

While our benchmark represents an important step toward more representative multilingual multi-modal evaluations, several limitations still remain. First, the dataset is inherently imbalanced across languages. Coverage varies depending on the availability and accessibility of exam sources, with some languages significantly underrepresented. Second, difficulty levels are not uniformly controlled. Since questions are drawn directly from real-world exams across diverse educational systems, variations in exam design, curricular focus, and intended grade levels introduce potential inconsistency in task complexity across languages and modalities. Further the chosen MCQA question format, inherent to many exams, has issues, see Appendix C.3. For instance: **Exploitation of biases:** Models may guess correct answers by exploiting statistical patterns or poorly designed distractors, inflating performance metrics without demonstrating genuine understanding. **Limited real-world applicability:** Unlike open-ended queries typical in real-world applications, MCQA provides predefined options, which may not reflect natural user interactions. **Choice-order sensitivity:** Performance can vary based on the order of answer choices, introducing inconsistencies unrelated to model capability. Finally, while the dataset expands coverage beyond English, the overall language diversity remains limited. Many languages, especially those spoken in low-resource regions, are still missing due to the scarcity of suitable exam material and annotators.

**Intended Use.**   KALEIDOSCOPE is designed as an evaluation-only benchmark for assessing multi-lingual and multimodal reasoning under exam-style, multiple-choice conditions. Appropriate uses include: diagnosing modality gaps between image-text and text-only settings; analyzing model behavior across languages, scripts, subjects, and image types; and studying cross-lingual or culturally grounded biases within a controlled MCQA format. KALEIDOSCOPE is not intended for evaluating free-form generation, long-context reasoning, conversational or interactive capabilities, or open-ended problem solving. It is also not designed for model training or fine-tuning.

## B   DATA COLLECTION DETAILS

### B.1   LICENSE

To ensure ethical data usage, we prioritize sources that permit redistribution and academic use. During data collection, we filter out content from sources with restrictive licensing policies. Additionally, our dataset does not include personally identifiable information, and all collected exams are either publicly available or obtained under appropriate agreements. To further guarantee compliance, we employ a two-stage validation process in which two blinded annotators independently verify the license of each exam included in our dataset. Only items that pass both validations are included in the final dataset.

### B.2   DIFFICULTY LEVELS

To better contextualize cross-language performance variability, we introduce a unified four-tier difficulty taxonomy (Basic, Intermediate, Advanced, Expert) derived from the original heterogeneous level metadata. To construct the mapping, we sampled 30 examples for each "level" category and determined the typical age and educational stage at which humans would encounter similar material. This served as a proxy for aligning the original categories to our four standardized bins.

Table 4 reports the resulting difficulty distribution for each language in KALEIDOSCOPE. Each cell lists both the raw count of questions and the corresponding percentage of that language's total. While the difficulty profiles vary across languages, reflecting differences in the types of publicly available exams, the overall benchmark is centered around Intermediate and Advanced material, corresponding broadly to high-school and undergraduate-level content.

We provide these difficulty assignments as additional metadata while preserving the original level tags, allowing the community to refine or reinterpret the taxonomy according to their research needs. For full transparency, we include the final mapping used to convert the original level categories into our four-tier difficulty scheme in our code.

Table 4: **Difficulty distribution across languages in** KALEIDOSCOPE. Each cell shows percentages with raw counts in grey.

| Language | Basic | | Intermediate | | Advanced | | Expert | | Total |
|---|---|---|---|---|---|---|---|---|---|
| Arabic | 0.0% | 0 | 100.0% | 382 | 0.0% | 0 | 0.0% | 0 | 382 |
| Bengali | 0.0% | 0 | 72.6% | 581 | 18.0% | 144 | 9.4% | 75 | 800 |
| German | 0.0% | 0 | 0.0% | 0 | 100.0% | 722 | 0.0% | 0 | 722 |
| English | 0.0% | 0 | 65.4% | 1065 | 17.0% | 277 | 17.6% | 286 | 1628 |
| Spanish | 0.0% | 0 | 58.9% | 873 | 3.8% | 56 | 37.3% | 553 | 1482 |
| Persian | 0.0% | 0 | 76.3% | 1526 | 23.7% | 474 | 0.0% | 0 | 2000 |
| French | 0.0% | 0 | 100.0% | 762 | 0.0% | 0 | 0.0% | 0 | 762 |
| Hindi | 13.1% | 248 | 28.6% | 540 | 21.2% | 399 | 37.1% | 699 | 1886 |
| Croatian | 0.0% | 0 | 100.0% | 324 | 0.0% | 0 | 0.0% | 0 | 324 |
| Hungarian | 0.0% | 0 | 100.0% | 1120 | 0.0% | 0 | 0.0% | 0 | 1120 |
| Lithuanian | 0.0% | 0 | 100.0% | 680 | 0.0% | 0 | 0.0% | 0 | 680 |
| Nepali | 0.0% | 0 | 100.0% | 126 | 0.0% | 0 | 0.0% | 0 | 126 |
| Dutch | 0.0% | 0 | 24.2% | 246 | 75.8% | 772 | 0.0% | 0 | 1018 |
| Portuguese | 0.0% | 0 | 0.0% | 0 | 100.0% | 2000 | 0.0% | 0 | 2000 |
| Russian | 0.0% | 0 | 100.0% | 1744 | 0.0% | 0 | 0.0% | 0 | 1744 |
| Serbian | 0.0% | 0 | 100.0% | 2000 | 0.0% | 0 | 0.0% | 0 | 2000 |
| Telugu | 0.0% | 0 | 0.0% | 0 | 100.0% | 1000 | 0.0% | 0 | 1000 |
| Ukrainian | 35.3% | 437 | 0.0% | 0 | 64.7% | 800 | 0.0% | 0 | 1237 |
| **All languages** | **3.3%** | **685** | **62.8%** | **12,869** | **39.5%** | **8,094** | **9.4%** | **1,929** | **20,911** |

## B.3 DATASET STATISTICS

Table 5: **Statistics of the** KALEIDOSCOPE **Dataset.** Breakdown of subjects (Subjects), total questions (Total), multimodal questions (Visual), and text-only questions (Text) per language. Languages are covered by multiple sources with single-subject cases containing specialized subdomains. 🖼: Supports evaluation of both multimodal (image+text) and unimodal (text-only) capabilities. 🌐: Languages are classified by resource level (high/mid/low) following Joshi et al. (2019); Singh et al. (2024b). 👥: Enables granular analysis of model performance across modalities, languages, and subject domains.

| Language | Code | Subjects | Total | Visual | Text | Resources | Family |
|---|---|---|---|---|---|---|---|
| Portuguese | pt | 11 | 2000 | 1000 | 1000 | High | Italic |
| Serbian | sr | 1 | 2000 | 1000 | 1000 | High | Balto-Slavic |
| Persian | fa | 5 | 2000 | 1000 | 1000 | High | Iranian |
| Hindi | hi | 12 | 1886 | 1000 | 886 | High | Indo-Aryan |
| Russian | ru | 1 | 1744 | 872 | 872 | High | Balto-Slavic |
| English | en | 9 | 1628 | 814 | 814 | High | Germanic |
| Spanish | es | 6 | 1482 | 741 | 741 | High | Italic |
| Hungarian | hu | 1 | 1120 | 560 | 560 | High | Uralic |
| Dutch | nl | 10 | 1018 | 509 | 509 | High | Germanic |
| French | fr | 1 | 762 | 381 | 381 | High | Italic |
| German | de | 1 | 722 | 361 | 361 | High | Germanic |
| Arabic | ar | 10 | 382 | 191 | 191 | High | Semitic |
| Croatian | hr | 1 | 324 | 162 | 162 | High | Balto-Slavic |
| Ukrainian | uk | 8 | 1237 | 1000 | 237 | Mid | Balto-Slavic |
| Bengali | bn | 6 | 800 | 400 | 400 | Mid | Indo-Aryan |
| Lithuanian | lt | 6 | 680 | 340 | 340 | Mid | Balto-Slavic |
| Telugu | te | 1 | 1000 | 1000 | 0 | Low | South Dravidian |
| Nepali | ne | 1 | 126 | 126 | 0 | Low | Indo-Aryan |
| **Total** | **(18)** | **14** | **20,911** | **11,457** | **9,454** | **–** | **–** |

## C  EXPANDED RELATED WORK

### C.1  PARTICIPATORY OPEN SCIENCE PROJECTS

Participatory research empowers diverse communities to actively contribute to research processes, capturing linguistic subtleties and cultural nuances directly from native speakers. Prior participatory NLP research has primarily targeted region-specific tasks such as translation, character recognition, and audio transcription. We highlight notable initiatives here which served as our motivation and backbone framework for building KALEIDOSCOPE.

In Africa, the **Masakhane**[4] community exemplifies impactful participatory NLP by focusing on grassroots-led data collection, annotation, and model creation for African languages. Nekoto et al. (2020) demonstrated that communities in low-resource environments significantly contribute to NLP, even without formal training. Subsequent efforts by Adelani et al. (2023) have further advanced dataset curation and model development for underrepresented African languages using similar participatory frameworks. Similarly, the **MaRVL** dataset (Multicultural Reasoning over Vision and Language; Liu et al., 2021) employed native speakers from diverse linguistic backgrounds (*Indonesian, Swahili, Tamil, Turkish*, and *Mandarin Chinese*) to contribute culturally representative images, subsequently annotated by professional linguists. Despite its cultural richness, MaRVL's modest scale (under 8,000 data points) limits broader applicability beyond evaluation.

In Latin America, participatory research has also emerged and is continuously growing through the help of communities. Recent works include Hernandez Mena & Meza Ruiz (2022), which developed eight open-access linguistic resources via structured social service programs, engaging student volunteers in transcription and segmentation tasks. Concurrently, Cañete et al. (2020) and Guevara-Rukoz et al. (2020) spearheaded crowd-sourced corpora addressing dialectal diversity and resource scarcity specific to Latin American Spanish.

Table 6: **Comparison of Multimodal Benchmarks.** [†]All but 40 questions are in English that measure machine translation capability from Chinese to English.

| Benchmark | Languages | Samples | Multimodal | Modalities | Human Annotation | Answer type |
|---|---|---|---|---|---|---|
| **MMMU** (Yue et al., 2024a) | 1 | 11,550 | 11,264 | Image-Text | Yes | MCQA |
| **SEED-Bench** (Li et al., 2024a) | 1 | 19,242 | 19,242 | Image-Text, Video-Text | Partial | MCQA |
| **MME** (Fu et al., 2023) | 1[†] | 2,194 | 0 | Image-Text | Partial | Y/N |
| **M3Exam** (Zhang et al., 2023) | 9 | 12,317 | 2,816 | Image-Text | Yes | MCQA |
| **EXAMS-V** (Das et al., 2024) | 11 | 20,932 | 5,086 | Image-Text | Yes | MCQA |
| **M5** (Schneider & Sitaram, 2024) | 41 | 237,094 | 1,422 | Image-Text | Yes | Mix |
| **KALEIDOSCOPE** | **18** | **20,911** | **11,459** | Image-Text | Yes | MCQA |

In Southeast Asia, **Project SEALD**[5], a collaboration between AI Singapore and Google Research, facilitated multilingual dataset collection to support regional Large Language Models (LLMs). Outputs from SEALD underpin open-source multilingual models such as *SEA-LION*[6], *Wangchan-Lion* (Phatthiyaphaibun et al., 2024), and *Sahabat-AI*[7]. Related initiatives include **NusaCrowd** for aggregating and standardizing Indonesian NLP datasets (Cahyawijaya et al., 2023) and the **SEACrowd** and **SEA-VL** projects aimed at comprehensive evaluation and benchmarking of LLMs across Southeast Asian languages (Cahyawijaya et al., 2025; Lovenia et al., 2024).

On a global scale, the **CVQA dataset** (Romero et al., 2024) was created using a participatory approach, involving native speakers and cultural experts from over 30 countries. Annotators were

---

[4]https://www.masakhane.io/

[5]**S**outh**e**ast **A**sian **L**anguages in One Network **D**ata; https://aisingapore.org/aiproducts/southeast-asian-languages-in-one-network-data-seald/

[6]https://sea-lion.ai

[7]https://sahabat-ai.com

selected for their fluency in local languages and cultural familiarity. Many contributors were also recognized as co-authors based on their level of involvement, reinforcing a collaborative, community-driven effort. The **Aya Initiative** employed participatory methods, engaging over 3,000 contributors to curate instruction datasets across 114 languages, resulting in one of the largest multilingual datasets for language model training (Singh et al., 2024b; Üstün et al., 2024). Similarly, the INCLUDE benchmark (Romanou et al., 2024) leveraged participatory approaches closely aligned with our methodology. The **BigScience ROOTS corpus**, developed collaboratively for the BLOOM model, exemplifies large-scale participatory data collection. Approximately 62% of ROOTS data was crowd-sourced via global hackathons and open submissions, involving over 1,000 researchers from 60 countries and more than 250 institutions, resulting in 1.6 terabytes of multilingual data (Laurençon et al., 2022). Additionally, Uzuner et al. (2010) underscored the viability of community-driven annotation for complex, domain-specific NLP tasks like clinical text annotation, highlighting broader applicability of participatory frameworks beyond general NLP domains.

Participatory methods have also successfully extended into reinforcement learning from human feedback (RLHF). For instance, the **OpenAssistant** project, led by LAION, utilized global crowdsourcing to construct a multilingual corpus comprising over 161,000 messages annotated by 13,500 volunteers. This dataset facilitated robust training of dialogue-aligned language models through extensive human feedback annotations (Köpf et al., 2023).

## C.2 COMPARISON WITH OTHER BENCHMARKS

Table 6 offers a concise comparison of key multimodal benchmarks. MMMU (Yue et al., 2024a), SEED-Bench (Li et al., 2024a), and MME (Fu et al., 2023) are single-language datasets focused mainly on image-text pairs, with SEED-Bench also incorporating video-text. MME is notably smaller and only partially human-annotated, using mostly true/false formats. In contrast, M3Exam (Zhang et al., 2023), EXAMS-V (Das et al., 2024), and M5 (Schneider & Sitaram, 2024) introduce multilingualism—M5 being the most extensive with 41 languages—though much of its content is not multiple-choice and lacks verified annotations.

KALEIDOSCOPE stands out by offering a balanced composition of 20,911 samples across 18 languages, with a strong focus on multimodal reasoning (11,459 Image-Text samples), comprehensive human annotation, and a consistent multiple-choice setup. Compared to existing benchmarks, KALEIDOSCOPE is more linguistically diverse than M3Exam and EXAMS-V, includes more multimodal samples than M5, and ensures higher quality through expert-verified annotations, making it a robust and equitable benchmark for evaluating multilingual multimodal models.

## C.3 EVALUATION METRICS AND THE MCQA FRAMEWORK

Traditional evaluation metrics for VLMs, such as exact match accuracy, BLEU (Papineni et al., 2002), ROUGE (Lin, 2004), and CIDEr (Vedantam et al., 2015), rely on surface-level n-gram comparisons that often penalize semantically equivalent answers phrased differently from reference texts. In contrast, the multiple-choice question answering (MCQA) framework (Hendrycks et al., 2021; Romero et al., 2024; Lu et al., 2022; Yue et al., 2024a) offers a more human-like evaluation paradigm by providing predefined answer options. This reduces ambiguity in scoring and facilitates the creation of evaluation datasets that capture both domain knowledge and linguistic/cultural nuances across languages. Although concerns regarding oversaturation and reliance on superficial cues in MCQA exist (Du et al., 2023; Yuksekgonul et al., 2022), these can be mitigated by extending the answer option space and applying rigorous filtering strategies (Wang et al., 2024b; Yue et al., 2024a). Our primary challenge lies in the scarcity of questions that are both multimodal and culturally agnostic. As demonstrated by results from KALEIDOSCOPE and related studies (Maaz et al., 2024; Nayak et al., 2024), oversaturation is not a prevalent issue. Consequently, bridging this evaluation gap is of key importance. To ensure high data quality, source data in KALEIDOSCOPE are manually verified by qualified processors in accordance with established criteria (2.2), maintaining a clear distinction between verified and unverified data.

# D  EXPERIMENT DETAILS & ADDITIONAL EXPERIMENTS

## D.1  EVALUATION METRICS

Given the multiple-choice nature of the task, we use accuracy as the primary evaluation metric. We report overall accuracy across all questions, as well as accuracy on the subset of questions where the model produces valid responses. A response is considered valid if the model successfully provides an answer in the expected format and selects a valid option (i.e., one of the letters A, B, C, D). Invalid responses typically result from missing the selected choice, selecting an invalid option, or refusal to answer. To quantify these cases, we report the *Format Error Rate*, which measures the proportion of questions for which the model fails to generate a valid answer. For grouped results, we report the macro average of valid answer accuracy across languages, i.e. all languages have equal weight when computing the score.

## D.2  PROMPT ABLATION: CoT VS. DIRECT APPROACHES

To benchmark the models, we initially designed a CoT prompt instructing them to think step-by-step and then provide the correct answer, marking the choice with the tags `<ANSWER> </ANSWER>`. However, in preliminary experiments, we found this instruction too complex for mid- to small-sized models (32B–3B), which struggled to follow it consistently.

In Table 7, we compare results using the CoT prompt versus the direct English-language prompt adopted in our final evaluation. The error rate was considerably higher for most models under the CoT setup, even after cleaning and extracting answers with regex matching their typical output formats. Two exceptions were Pangea and Molmo, which showed lower error rates with the CoT prompt; however, this occurred because both models ignored the reasoning instruction and directly output the selected option, making extraction easier. Overall, prompt choice significantly impacted performance: the direct English prompt improved results across all models except Pangea, whose performance remained unchanged.

For closed models, we also compare the performance of CoT versus the direct prompt on GPT-4o in Table 8. We observe an interesting trade-off: while CoT improves accuracy, it also increases the format error rate. As detailed in Section 5.3, these errors are tightly related with implicit refusals, especially in questions involving medical or safety-relevant content. We hypothesize that CoT gives the model more room to reason toward the correct answer but also increases the likelihood of refusals, thereby raising the format-error rate.

Based on these findings, we select the best-performing prompting strategy for each class of models: the direct prompt for open-weight models and CoT for closed-weight models.

Table 7: **Comparison of CoT and direct English prompting on** KALEIDOSCOPE **for small models**. Reported values are macro-averaged accuracy (%) across all languages.

| Model | Overall CoT | | | Overall In-English | | |
|---|---|---|---|---|---|---|
| | | Valid Responses | | | Valid Responses | |
| | Acc. | F.E. | Valid Acc. | Acc. | F.E. | Valid Acc. |
| Aya-Vision-32B | 38.94 | 8.04 | 42.06 | 39.27 | 1.05 | 39.66 |
| Aya-Vision-8B | 33.08 | 6.22 | 35.15 | 35.09 | 0.07 | 35.11 |
| Molmo-7B-D | 32.86 | 0.01 | 32.87 | 32.87 | 0.04 | 32.88 |
| Pangea-7B | 31.24 | 5.61 | 33.45 | 31.31 | 7.42 | 34.02 |
| Qwen2.5-VL-7B | 35.18 | 6.34 | 37.64 | 39.56 | 0.08 | 39.60 |
| Qwen2.5-VL-3B | 32.90 | 1.40 | 33.33 | 35.56 | 0.19 | 35.63 |

Table 8: **Comparison of different prompting strategies on GPT-4o.** Results are shown disaggregated by language. Global accuracy, valid answer accuracy and format error rate are reported.

| Language | Direct Prompt | | | CoT Prompt | | |
|---|---|---|---|---|---|---|
| | **Total Acc.** | **Valid Acc.** | **F.E.** | **Total Acc.** | **Valid Acc.** | **F.E.** |
| Arabic | 49.7 | 50.4 | 1.3 | 52.9 | **57.7** | 8.4 |
| Bengali | 57.4 | 57.5 | 0.2 | 65.6 | **67.3** | 2.5 |
| Croatian | 33.3 | 33.8 | 1.2 | 49.7 | **52.6** | 5.6 |
| Dutch | 57.1 | 58.1 | 1.8 | 58.9 | **62.4** | 5.6 |
| English | 63.9 | 64.0 | 0.1 | 60.8 | **73.4** | 17.1 |
| French | 37.7 | 37.7 | 0.1 | 61.8 | **64.6** | 4.3 |
| German | 70.8 | 70.8 | 0.0 | 71.6 | **72.6** | 1.4 |
| Hindi | 48.6 | 48.7 | 0.2 | 60.1 | **64.0** | 6.1 |
| Hungarian | 34.0 | 34.4 | 1.1 | 47.2 | **50.6** | 6.7 |
| Lithuanian | 83.4 | 83.4 | 0.0 | 86.5 | **88.4** | 2.2 |
| Nepali | 23.8 | **24.0** | 0.8 | 19.0 | 20.5 | 7.1 |
| Persian | 40.2 | 40.3 | 0.3 | 47.0 | **47.9** | 2.0 |
| Portuguese | 75.4 | 75.4 | 0.0 | 82.6 | **85.2** | 3.0 |
| Russian | 36.6 | 36.6 | 0.1 | 51.5 | **54.5** | 5.5 |
| Serbian | 33.1 | 33.4 | 1.0 | 47.0 | **52.6** | 10.6 |
| Spanish | 75.4 | 75.5 | 0.2 | 77.7 | **80.1** | 3.0 |
| Telugu | 42.1 | 44.3 | 4.9 | 41.6 | **47.9** | 13.2 |
| Ukrainian | 71.4 | 72.2 | 1.1 | 68.1 | **75.3** | 9.5 |
| Overall | 51.9 | 52.2 | 0.7 | 58.3 | **62.1** | 6.5 |

## D.3 COMPLETE RESULTS

We report full results on multimodal performances grouped by subject in table Tables 9, and full results for all questions grouped by subject, Table 10, and language, Table 11. Each table reports, for each model and category; **Total Accuracy %**: the accuracy over all samples, **Valid Accuracy %**: the accuracy over successfully extracted answers and **Format Error % (FE)**: the proportion of unextracted answers.

Table 9: **Subject-wise Performance on** KALEIDOSCOPE**'s Multimodal Questions.** Valid accuracy (%) across examination subjects for only multimodal samples, with bold highlighting top-performing models.

| | Closed Weights | | | Open Weights | | | | | | |
|---|---|---|---|---|---|---|---|---|---|---|
| | Gemini | Claude | GPT-4o | Qwen2.5-3B | Molmo-7B | Pangea-7B | Qwen2.5-7B | Aya V-8B | Aya V-32B | Qwen2.5-72B |
| *Humanities & Social Sciences* | | | | | | | | | | |
| Economics | 64.1 | 63.8 | **66.7** | 37.7 | 27.5 | 33.9 | 42.7 | 30.9 | 29.8 | 58.8 |
| Geography | 72.8 | **81.5** | 80.4 | 40.7 | 37.6 | 36.7 | 51.0 | 39.5 | 50.5 | 70.4 |
| History | 78.7 | 83.7 | **86.4** | 48.9 | 42.1 | 42.4 | 52.9 | 45.6 | 61.4 | 77.1 |
| Language | 83.5 | 85.5 | **85.8** | 72.2 | 60.1 | 66.0 | 75.7 | 56.6 | 71.2 | 85.1 |
| Social Sciences | 85.7 | 82.9 | **88.1** | 52.9 | 52.2 | 53.8 | 68.6 | 58.0 | 64.3 | 80.0 |
| Sociology | 92.3 | **93.4** | 93.2 | 64.1 | 61.0 | 57.3 | 73.1 | 57.7 | 70.5 | 87.2 |
| *STEM* | | | | | | | | | | |
| Biology | 60.3 | 62.9 | **63.9** | 37.6 | 35.3 | 33.4 | 42.6 | 35.4 | 40.7 | 53.8 |
| Chemistry | **60.4** | 59.7 | 52.9 | 33.2 | 33.5 | 34.1 | 38.5 | 28.0 | 34.8 | 50.0 |
| Engineering | 57.3 | **64.4** | 56.3 | 28.9 | 24.4 | 24.2 | 32.4 | 30.3 | 34.8 | 48.4 |
| Mathematics | **48.6** | 44.4 | 44.0 | 30.4 | 28.8 | 29.0 | 30.1 | 28.6 | 29.6 | 40.3 |
| Physics | 57.8 | **58.7** | 54.7 | 33.7 | 26.7 | 28.9 | 34.7 | 27.1 | 33.0 | 42.3 |
| *Reasoning, Health Science, and Practical Skills* | | | | | | | | | | |
| Reasoning | 52.0 | **53.3** | 51.0 | 27.4 | 27.5 | 26.6 | 29.5 | 25.1 | 27.6 | 42.3 |
| Medicine | 70.2 | 73.8 | **75.6** | 36.7 | 40.4 | 38.4 | 45.8 | 35.4 | 52.3 | 63.3 |
| Driving License | 64.4 | 64.2 | **73.1** | 39.0 | 44.9 | 39.4 | 44.9 | 41.6 | 47.1 | 54.5 |

Table 10: **Total Accuracy %**, **Valid Accuracy %** and **Format Error % (FE)** grouped by Subject in KALEIDOSCOPE for multimodal samples.

| | | Biology | Chemistry | Driving License | Economics | Engineering | Geography | History | Language | Mathematics | Medicine | Physics | Reasoning | Social Sciences | Sociology |
|---|---|---|---|---|---|---|---|---|---|---|---|---|---|---|---|
| **Gemini 1.5 Pro** | Total Acc. | 60.1 | 60.2 | 64.4 | 64.1 | 57.0 | 72.8 | 78.7 | 83.5 | 46.9 | 69.6 | 57.4 | 51.2 | 85.7 | 92.3 |
| | Valid Acc. | 60.3 | 60.4 | 64.4 | 64.1 | 57.3 | 72.8 | 78.7 | 83.5 | 48.6 | 70.2 | 57.8 | 52.0 | 85.7 | 92.3 |
| | FE | 0.3 | 0.4 | 0.0 | 0.0 | 0.4 | 0.0 | 0.0 | 0.0 | 3.5 | 0.8 | 0.7 | 1.7 | 0.0 | 0.0 |
| **Claude 3.5 Sonnet** | Total Acc. | 61.6 | 59.0 | 64.2 | 63.4 | 50.0 | 81.4 | 83.7 | 85.1 | 43.7 | 72.9 | 58.4 | 52.2 | 82.9 | 91.0 |
| | Valid Acc. | 62.9 | 59.7 | 64.2 | 63.8 | 64.4 | 81.5 | 83.7 | 85.5 | 44.4 | 73.8 | 58.7 | 53.3 | 82.9 | 93.4 |
| | FE | 2.1 | 1.2 | 0.0 | 0.8 | 22.4 | 0.2 | 0.0 | 0.5 | 1.5 | 1.2 | 0.6 | 2.0 | 0.0 | 2.6 |
| **GPT-4o** | Total Acc. | 60.7 | 47.1 | 65.5 | 64.1 | 45.7 | 76.2 | 70.8 | 78.3 | 39.4 | 64.6 | 51.9 | 43.9 | 74.3 | 87.2 |
| | Valid Acc. | 63.9 | 52.9 | 73.1 | 66.7 | 56.3 | 80.4 | 86.4 | 85.8 | 44.0 | 75.6 | 54.7 | 51.0 | 88.1 | 93.2 |
| | FE | 5.1 | 11.1 | 10.4 | 3.8 | 18.9 | 5.2 | 18.1 | 8.7 | 10.5 | 14.6 | 5.1 | 13.9 | 15.7 | 6.4 |
| **Qwen2.5-VL-72B** | Total Acc. | 37.6 | 33.2 | 39.0 | 37.4 | 28.8 | 40.6 | 48.9 | 72.2 | 30.1 | 36.7 | 33.7 | 27.4 | 52.9 | 64.1 |
| | Valid Acc. | 37.6 | 33.2 | 39.0 | 37.7 | 28.9 | 40.7 | 48.9 | 72.2 | 30.4 | 36.7 | 33.7 | 27.4 | 52.9 | 64.1 |
| | FE | 0.0 | 0.0 | 0.0 | 0.8 | 0.2 | 0.2 | 0.0 | 0.0 | 0.9 | 0.0 | 0.0 | 0.0 | 0.0 | 0.0 |
| **Aya-Vision-32B** | Total Acc. | 40.2 | 34.6 | 47.1 | 29.8 | 34.6 | 50.5 | 61.4 | 70.5 | 28.8 | 52.1 | 31.8 | 27.5 | 64.3 | 70.5 |
| | Valid Acc. | 40.7 | 34.8 | 47.1 | 29.8 | 34.8 | 50.5 | 61.4 | 71.2 | 29.6 | 52.3 | 33.0 | 27.6 | 64.3 | 70.5 |
| | FE | 1.3 | 0.5 | 0.0 | 0.0 | 0.7 | 0.0 | 0.0 | 0.2 | 2.8 | 0.4 | 3.6 | 0.3 | 0.0 | 0.0 |
| **Aya-Vision-8B** | Total Acc. | 53.8 | 50.0 | 54.5 | 58.8 | 48.4 | 70.4 | 77.1 | 84.9 | 40.3 | 63.3 | 42.3 | 42.3 | 80.0 | 87.2 |
| | Valid Acc. | 53.8 | 50.0 | 54.5 | 58.8 | 48.4 | 70.4 | 77.1 | 85.1 | 40.3 | 63.3 | 42.3 | 42.3 | 80.0 | 87.2 |
| | FE | 0.0 | 0.0 | 0.0 | 0.0 | 0.0 | 0.0 | 0.0 | 0.2 | 0.1 | 0.0 | 0.0 | 0.0 | 0.0 | 0.0 |
| **Molmo-7B-D** | Total Acc. | 35.2 | 33.5 | 44.9 | 27.5 | 24.4 | 37.6 | 42.1 | 60.1 | 28.7 | 40.4 | 26.7 | 27.5 | 51.4 | 60.3 |
| | Valid Acc. | 35.3 | 33.5 | 44.9 | 27.5 | 24.4 | 37.6 | 42.1 | 60.1 | 28.8 | 40.4 | 26.7 | 27.5 | 52.2 | 61.0 |
| | FE | 0.1 | 0.0 | 0.0 | 0.0 | 0.1 | 0.0 | 0.0 | 0.0 | 0.1 | 0.0 | 0.0 | 0.0 | 1.4 | 1.3 |
| **Pangea-7B** | Total Acc. | 30.9 | 22.0 | 38.2 | 32.1 | 21.4 | 35.6 | 42.1 | 65.8 | 24.8 | 35.0 | 24.7 | 21.9 | 50.0 | 55.1 |
| | Valid Acc. | 33.4 | 34.1 | 39.4 | 33.9 | 24.2 | 36.7 | 42.4 | 66.0 | 29.0 | 38.4 | 28.9 | 26.6 | 53.8 | 57.3 |
| | FE | 7.5 | 35.3 | 2.9 | 5.3 | 11.5 | 3.0 | 0.6 | 0.2 | 14.4 | 8.8 | 14.7 | 17.6 | 7.1 | 3.8 |
| **Qwen2.5-VL-7B** | Total Acc. | 34.3 | 16.2 | 41.2 | 22.1 | 30.3 | 38.7 | 45.3 | 56.6 | 28.6 | 35.4 | 27.1 | 24.3 | 57.1 | 57.7 |
| | Valid Acc. | 35.4 | 28.0 | 41.6 | 30.9 | 30.3 | 39.5 | 45.6 | 56.6 | 28.6 | 35.4 | 27.1 | 25.1 | 58.0 | 57.7 |
| | FE | 3.0 | 42.2 | 1.1 | 28.2 | 0.0 | 1.9 | 0.6 | 0.0 | 0.3 | 0.0 | 0.0 | 3.4 | 1.4 | 0.0 |
| **Qwen2.5-VL-3B** | Total Acc. | 42.6 | 38.5 | 44.9 | 42.7 | 32.4 | 51.0 | 52.9 | 75.7 | 30.1 | 45.8 | 34.7 | 29.5 | 68.6 | 73.1 |
| | Valid Acc. | 42.6 | 38.5 | 44.9 | 42.7 | 32.4 | 51.0 | 52.9 | 75.7 | 30.1 | 45.8 | 34.7 | 29.5 | 68.6 | 73.1 |
| | FE | 0.0 | 0.1 | 0.0 | 0.0 | 0.0 | 0.0 | 0.0 | 0.0 | 0.1 | 0.0 | 0.1 | 0.0 | 0.0 | 0.0 |

Table 11: **Total Accuracy %**, **Valid Accuracy %** and **Format Error % (FE)** grouped by Language in KALEIDOSCOPE for multimodal samples.

| | | Latin Script | | | | | | Non-Latin Script | | | | | | | | | | | |
|---|---|---|---|---|---|---|---|---|---|---|---|---|---|---|---|---|---|---|---|
| | | English | French | German | Dutch | Portuguese | Spanish | Arabic | Bengali | Croatian | Hindi | Hungarian | Lithuanian | Nepali | Persian | Russian | Serbian | Telugu | Ukrainian |
| **Gemini 1.5 Pro** | Total Acc. | 62.7 | 54.6 | 52.6 | 61.5 | 81.8 | 78.5 | 44.5 | 51.8 | 46.9 | 62.6 | 39.1 | 75.0 | 22.2 | 41.2 | 45.0 | 41.9 | 58.1 | 70.3 |
| | Valid Acc. | 63.2 | 55.2 | 52.6 | 62.5 | 83.4 | 78.5 | 44.7 | 52.7 | 48.1 | 63.2 | 40.5 | 75.0 | 22.8 | 42.1 | 46.2 | 43.6 | 58.3 | 70.3 |
| | FE | 0.9 | 1.0 | 0.0 | 1.6 | 1.9 | 0.0 | 0.5 | 1.8 | 2.5 | 0.9 | 3.4 | 0.0 | 2.4 | 2.1 | 2.6 | 3.8 | 0.4 | 0.0 |
| **Claude 3.5 Sonnet** | Total Acc. | 36.9 | 51.7 | 73.7 | 65.2 | 83.8 | 77.6 | 50.3 | 49.5 | 46.9 | 57.1 | 38.8 | 81.2 | 27.8 | 45.6 | 45.3 | 38.9 | 56.0 | 75.2 |
| | Valid Acc. | 63.2 | 51.7 | 73.7 | 65.6 | 83.8 | 77.7 | 50.3 | 49.6 | 47.5 | 57.2 | 38.9 | 81.4 | 28.0 | 45.6 | 45.4 | 39.6 | 56.0 | 75.2 |
| | FE | 41.6 | 0.0 | 0.0 | 0.6 | 0.0 | 0.1 | 0.0 | 0.2 | 1.2 | 0.1 | 0.4 | 0.3 | 0.8 | 0.0 | 0.2 | 1.8 | 0.0 | 0.0 |
| **GPT-4o** | Total Acc. | 42.6 | 46.2 | 52.4 | 60.1 | 76.6 | 73.8 | 41.9 | 60.2 | 36.4 | 53.0 | 36.4 | 76.2 | 19.0 | 41.3 | 37.6 | 32.4 | 41.6 | 68.5 |
| | Valid Acc. | 64.5 | 49.2 | 53.7 | 66.8 | 81.0 | 78.6 | 49.7 | 62.8 | 40.4 | 59.0 | 40.7 | 79.7 | 20.5 | 42.6 | 40.3 | 38.7 | 47.9 | 77.5 |
| | FE | 33.9 | 6.0 | 2.5 | 10.0 | 5.4 | 6.1 | 15.7 | 4.0 | 9.9 | 10.1 | 10.5 | 4.4 | 7.1 | 3.0 | 6.7 | 16.2 | 13.2 | 11.6 |
| **Qwen2.5-VL-72B** | Total Acc. | 53.8 | 46.2 | 49.3 | 57.4 | 73.3 | 72.5 | 36.1 | 49.5 | 33.3 | 46.2 | 37.5 | 69.1 | 23.8 | 35.4 | 39.3 | 35.9 | 47.0 | 65.2 |
| | Valid Acc. | 53.8 | 46.4 | 49.3 | 57.5 | 73.3 | 72.5 | 36.1 | 49.5 | 33.3 | 46.2 | 37.6 | 69.1 | 23.8 | 35.4 | 39.3 | 35.9 | 47.0 | 65.2 |
| | FE | 0.0 | 0.5 | 0.0 | 0.2 | 0.0 | 0.0 | 0.0 | 0.0 | 0.0 | 0.0 | 0.2 | 0.0 | 0.0 | 0.0 | 0.0 | 0.0 | 0.0 | 0.0 |
| **Aya-Vision-32B** | Total Acc. | 32.9 | 27.6 | 39.1 | 47.5 | 57.4 | 53.2 | 30.4 | 26.0 | 29.6 | 32.9 | 26.1 | 48.5 | 22.2 | 30.3 | 29.0 | 28.4 | 34.8 | 47.1 |
| | Valid Acc. | 33.0 | 29.3 | 39.5 | 48.7 | 58.8 | 55.5 | 30.5 | 26.5 | 29.6 | 33.7 | 26.5 | 48.5 | 22.2 | 31.1 | 29.4 | 28.5 | 34.8 | 47.1 |
| | FE | 0.4 | 6.0 | 1.1 | 2.4 | 2.4 | 4.2 | 0.5 | 0.2 | 0.0 | 2.3 | 1.6 | 0.0 | 0.0 | 2.5 | 1.3 | 0.4 | 0.1 | 0.0 |
| **Aya-Vision-8B** | Total Acc. | 28.9 | 30.2 | 32.4 | 42.2 | 47.6 | 46.3 | 27.2 | 18.2 | 29.0 | 29.2 | 27.1 | 34.4 | 23.8 | 27.7 | 25.2 | 28.5 | 11.1 | 36.6 |
| | Valid Acc. | 28.9 | 30.2 | 32.4 | 42.2 | 47.6 | 46.3 | 27.2 | 25.9 | 29.0 | 29.3 | 27.1 | 34.4 | 24.2 | 27.8 | 25.2 | 28.5 | 25.7 | 36.8 |
| | FE | 0.0 | 0.0 | 0.0 | 0.0 | 0.0 | 0.0 | 0.0 | 29.5 | 0.0 | 0.2 | 0.0 | 0.0 | 1.6 | 0.3 | 0.0 | 0.0 | 56.8 | 0.6 |
| **Molmo-7B-D** | Total Acc. | 26.9 | 33.1 | 19.9 | 41.5 | 47.6 | 47.8 | 21.5 | 26.0 | 30.9 | 30.0 | 25.5 | 35.3 | 25.4 | 28.3 | 29.2 | 27.2 | 33.9 | 35.8 |
| | Valid Acc. | 27.0 | 33.3 | 19.9 | 41.5 | 47.6 | 47.8 | 21.5 | 26.0 | 30.9 | 30.1 | 25.5 | 35.3 | 25.4 | 28.3 | 29.2 | 27.2 | 33.9 | 35.8 |
| | FE | 0.2 | 0.8 | 0.0 | 0.0 | 0.0 | 0.0 | 0.0 | 0.0 | 0.0 | 0.2 | 0.0 | 0.0 | 0.0 | 0.0 | 0.0 | 0.0 | 0.0 | 0.1 |
| **Pangea-7B** | Total Acc. | 24.7 | 25.5 | 17.2 | 37.3 | 48.8 | 46.2 | 20.4 | 27.8 | 17.9 | 24.3 | 23.9 | 32.6 | 17.5 | 21.2 | 25.5 | 26.4 | 18.6 | 33.0 |
| | Valid Acc. | 27.9 | 31.1 | 19.6 | 39.0 | 51.0 | 49.4 | 22.3 | 31.8 | 21.0 | 29.7 | 26.2 | 33.4 | 23.9 | 25.2 | 28.9 | 30.1 | 33.9 | 33.9 |
| | FE | 11.4 | 18.1 | 12.2 | 4.3 | 4.3 | 6.5 | 8.4 | 12.8 | 14.8 | 18.3 | 8.8 | 2.4 | 27.0 | 16.0 | 11.9 | 12.4 | 45.2 | 2.6 |
| **Qwen2.5-VL-7B** | Total Acc. | 36.4 | 35.2 | 26.6 | 46.4 | 59.2 | 57.8 | 36.1 | 35.0 | 25.9 | 33.8 | 28.6 | 50.3 | 22.2 | 30.3 | 28.8 | 27.5 | 37.6 | 45.4 |
| | Valid Acc. | 36.4 | 35.3 | 26.6 | 46.4 | 59.2 | 57.8 | 36.1 | 35.1 | 26.1 | 33.9 | 28.6 | 50.3 | 22.2 | 30.3 | 28.8 | 27.5 | 37.6 | 45.4 |
| | FE | 0.0 | 0.3 | 0.0 | 0.0 | 0.0 | 0.0 | 0.0 | 0.2 | 0.6 | 0.2 | 0.2 | 0.0 | 0.0 | 0.0 | 0.0 | 0.0 | 0.0 | 0.0 |
| **Qwen2.5-VL-3B** | Total Acc. | 35.0 | 32.3 | 21.1 | 41.8 | 52.2 | 53.3 | 33.5 | 32.8 | 25.9 | 31.3 | 28.2 | 35.6 | 23.8 | 30.3 | 29.0 | 27.8 | 31.8 | 40.1 |
| | Valid Acc. | 35.0 | 32.4 | 21.1 | 41.8 | 52.3 | 53.3 | 33.7 | 32.8 | 26.4 | 31.3 | 29.2 | 35.6 | 23.8 | 30.4 | 29.0 | 28.1 | 31.8 | 40.1 |
| | FE | 0.0 | 0.3 | 0.0 | 0.0 | 0.1 | 0.0 | 0.5 | 0.2 | 1.9 | 0.0 | 3.2 | 0.0 | 0.0 | 0.2 | 0.0 | 1.1 | 0.0 | 0.0 |

Table 12: **Valid Accuracy (%) for All Questions and All Models, by Language and General Category.** Languages are grouped by script. Empty cells indicate either no questions for that category, or only format errors in the answers from the model.

| General Category | Model | Latin Script | | | | | | Non-Latin Script | | | | | | | | | | | |
|---|---|---|---|---|---|---|---|---|---|---|---|---|---|---|---|---|---|---|---|
| | | English | French | German | Dutch | Portuguese | Spanish | Arabic | Bengali | Croatian | Hindi | Hungarian | Lithuanian | Nepali | Persian | Russian | Serbian | Telugu | Ukrainian |
| Health Sciences | Gemini 1.5 Pro | – | – | – | 0.562 | 1.000 | 0.781 | – | – | – | 1.000 | – | – | – | – | – | – | – | – |
| | Claude 3.5 Sonnet | – | – | – | 0.57 | 1.000 | 0.833 | – | – | – | 1.000 | – | – | – | – | – | – | – | – |
| | GPT-4o | – | – | – | 0.6 | 1.000 | 0.861 | – | – | – | 1.000 | – | – | – | – | – | – | – | – |
| | Qwen2.5-72B | – | – | – | 0.570 | 1.000 | 0.743 | – | – | – | 1.000 | – | – | – | – | – | – | – | – |
| | Aya-Vision-32B | – | – | – | 0.565 | 1.000 | 0.591 | – | – | – | 1.000 | – | – | – | – | – | – | – | – |
| | Aya-Vision-8B | – | – | – | 0.430 | 0.333 | 0.439 | – | – | – | 1.000 | – | – | – | – | – | – | – | – |
| | Molmo-7B-D | – | – | – | 0.488 | 1.000 | 0.432 | – | – | – | 0.000 | – | – | – | – | – | – | – | – |
| | Pangea-7B | – | – | – | 0.500 | 1.000 | 0.444 | – | – | – | 1.000 | – | – | – | – | – | – | – | – |
| | Qwen2.5-7B | – | – | – | 0.523 | 1.000 | 0.556 | – | – | – | 1.000 | – | – | – | – | – | – | – | – |
| | Qwen2.5-3B | – | – | – | 0.419 | 0.667 | 0.458 | – | – | – | 0.000 | – | – | – | – | – | – | – | – |
| Humanities & Culture | Gemini 1.5 Pro | 0.96 | – | – | 0.68 | 0.92 | 0.87 | 0.592 | – | – | 1.0 | – | 0.871 | – | – | – | – | – | 0.7 |
| | Claude 3.5 Sonnet | 0.97 | – | – | 0.741 | 0.925 | 0.875 | 0.631 | – | – | 1.000 | – | 0.94 | – | – | – | – | – | 0.79 |
| | GPT-4o | 0.971 | – | – | 0.732 | 0.924 | 0.861 | 0.59 | – | – | 1.000 | – | 0.93 | – | – | – | – | – | 0.84 |
| | Qwen2.5-72B | 0.956 | – | – | 0.706 | 0.906 | 0.888 | 0.530 | – | – | 0.333 | – | 0.787 | – | – | – | – | – | 0.701 |
| | Aya-Vision-32B | 0.779 | – | – | 0.671 | 0.824 | 0.737 | 0.506 | – | – | 0.333 | – | 0.592 | – | – | – | – | – | 0.490 |
| | Aya-Vision-8B | 0.721 | – | – | 0.486 | 0.723 | 0.659 | 0.398 | – | – | 0.333 | – | 0.414 | – | – | – | – | – | 0.352 |
| | Molmo-7B-D | 0.588 | – | – | 0.493 | 0.659 | 0.626 | 0.313 | – | – | 1.000 | – | 0.342 | – | – | – | – | – | 0.323 |
| | Pangea-7B | 0.676 | – | – | 0.479 | 0.671 | 0.736 | 0.349 | – | – | 0.000 | – | 0.375 | – | – | – | – | – | 0.348 |
| | Qwen2.5-7B | 0.809 | – | – | 0.528 | 0.796 | 0.832 | 0.422 | – | – | 0.333 | – | 0.479 | – | – | – | – | – | 0.424 |
| | Qwen2.5-3B | 0.765 | – | – | 0.493 | 0.701 | 0.816 | 0.494 | – | – | 0.333 | – | 0.346 | – | – | – | – | – | 0.420 |
| General Knowledge | Gemini 1.5 Pro | 0.55 | – | – | – | – | 0.712 | – | 0.792 | – | 0.634 | – | – | – | 0.582 | – | – | – | 0.57 |
| | Claude 3.5 Sonnet | 0.53 | – | – | – | – | 0.746 | – | 0.87 | – | 0.582 | – | – | – | 0.678 | – | – | – | 0.642 |
| | GPT-4o | 0.5 | – | – | – | – | 0.722 | – | 0.79 | – | 0.631 | – | – | – | 0.8 | – | 0.5 | – | 0.66 |
| | Qwen2.5-72B | 0.500 | – | – | – | – | 0.710 | – | 0.821 | – | 0.522 | – | – | – | 0.442 | – | – | – | 0.540 |
| | Aya-Vision-32B | 0.500 | – | – | – | – | 0.653 | – | 0.737 | – | 0.468 | – | – | – | 0.404 | – | – | – | 0.451 |
| | Aya-Vision-8B | 0.350 | – | – | – | – | 0.609 | – | 0.564 | – | 0.488 | – | – | – | 0.404 | – | – | – | 0.412 |
| | Molmo-7B-D | 0.500 | – | – | – | – | 0.634 | – | 0.385 | – | 0.373 | – | – | – | 0.519 | – | – | – | 0.419 |
| | Pangea-7B | 0.467 | – | – | – | – | 0.602 | – | 0.718 | – | 0.395 | – | – | – | 0.347 | – | – | – | 0.423 |
| | Qwen2.5-7B | 0.500 | – | – | – | – | 0.602 | – | 0.590 | – | 0.465 | – | – | – | 0.442 | – | – | – | 0.423 |
| | Qwen2.5-3B | 0.550 | – | – | – | – | 0.645 | – | 0.487 | – | 0.336 | – | – | – | 0.462 | – | – | – | 0.362 |
| Reasoning | Gemini 1.5 Pro | 0.31 | – | – | – | – | – | – | 0.58 | – | 0.7 | – | – | 0.23 | – | – | – | – | – |
| | Claude 3.5 Sonnet | 0.35 | – | – | – | – | – | – | 0.58 | – | 0.66 | – | – | 0.28 | – | – | – | – | – |
| | GPT-4o | 0.4 | – | – | – | – | – | – | 0.57 | – | 0.67 | – | – | 0.21 | – | – | – | – | – |
| | Qwen2.5-72B | 0.311 | – | – | – | – | – | – | 0.500 | – | 0.501 | – | – | 0.238 | – | – | – | – | – |
| | Aya-Vision-32B | 0.284 | – | – | – | – | – | – | 0.336 | – | 0.338 | – | – | 0.222 | – | – | – | – | – |
| | Aya-Vision-8B | 0.176 | – | – | – | – | – | – | 0.235 | – | 0.308 | – | – | 0.214 | – | – | – | – | – |
| | Molmo-7B-D | 0.176 | – | – | – | – | – | – | 0.379 | – | 0.273 | – | – | 0.254 | – | – | – | – | – |
| | Pangea-7B | 0.203 | – | – | – | – | – | – | 0.302 | – | 0.300 | – | – | 0.239 | – | – | – | – | – |
| | Qwen2.5-7B | 0.230 | – | – | – | – | – | – | 0.409 | – | 0.353 | – | – | 0.222 | – | – | – | – | – |
| | Qwen2.5-3B | 0.230 | – | – | – | – | – | – | 0.280 | – | 0.295 | – | – | 0.238 | – | – | – | – | – |
| STEM | Gemini 1.5 Pro | 0.72 | 0.69 | 0.7 | 0.58 | 0.83 | 0.8 | 0.51 | 0.53 | 0.58 | 0.64 | 0.54 | 0.85 | – | 0.52 | 0.58 | 0.58 | 0.58 | 0.76 |
| | Claude 3.5 Sonnet | 0.75 | 0.67 | 0.83 | 0.6 | 0.83 | 0.76 | 0.52 | 0.47 | 0.56 | 0.58 | 0.5 | 0.87 | – | 0.52 | 0.57 | 0.53 | 0.56 | 0.77 |
| | GPT-4o | 0.74 | 0.65 | 0.73 | 0.58 | 0.79 | 0.77 | 0.53 | 0.65 | 0.53 | 0.6 | 0.51 | 0.89 | – | 0.47 | 0.54 | 0.53 | 0.48 | 0.77 |
| | Qwen2.5-72B | 0.641 | 0.507 | 0.673 | 0.507 | 0.627 | 0.709 | 0.386 | 0.478 | 0.373 | 0.487 | 0.410 | 0.742 | – | 0.395 | 0.427 | 0.403 | 0.470 | 0.687 |
| | Aya-Vision-32B | 0.484 | 0.290 | 0.519 | 0.378 | 0.461 | 0.485 | 0.328 | 0.276 | 0.285 | 0.361 | 0.281 | 0.472 | – | 0.299 | 0.285 | 0.291 | 0.348 | 0.457 |
| | Aya-Vision-8B | 0.407 | 0.265 | 0.471 | 0.353 | 0.384 | 0.442 | 0.270 | 0.248 | 0.259 | 0.329 | 0.280 | 0.395 | – | 0.301 | 0.292 | 0.270 | 0.346 | 0.404 |
| | Molmo-7B-D | 0.379 | 0.301 | 0.285 | 0.339 | 0.343 | 0.457 | 0.219 | 0.288 | 0.284 | 0.284 | 0.279 | 0.379 | – | 0.272 | 0.302 | 0.272 | 0.339 | 0.346 |
| | Pangea-7B | 0.424 | 0.287 | 0.322 | 0.342 | 0.408 | 0.443 | 0.249 | 0.299 | 0.240 | 0.285 | 0.290 | 0.397 | – | 0.266 | 0.286 | 0.286 | 0.339 | 0.347 |
| | Qwen2.5-7B | 0.480 | 0.362 | 0.422 | 0.376 | 0.473 | 0.540 | 0.352 | 0.324 | 0.300 | 0.345 | 0.301 | 0.581 | – | 0.312 | 0.298 | 0.282 | 0.376 | 0.484 |
| | Qwen2.5-3B | 0.453 | 0.331 | 0.355 | 0.325 | 0.398 | 0.477 | 0.313 | 0.354 | 0.305 | 0.317 | 0.305 | 0.331 | – | 0.276 | 0.303 | 0.271 | 0.318 | 0.425 |
| Social Sciences | Gemini 1.5 Pro | 1.0 | – | – | 0.8 | 0.9 | – | 0.65 | 0.66 | – | 0.75 | – | 0.8 | – | – | – | – | – | 0.76 |
| | Claude 3.5 Sonnet | 0.9 | – | – | 0.84 | 0.92 | – | 0.7 | 0.66 | – | 0.88 | – | 0.86 | – | – | – | – | – | 0.87 |
| | GPT-4o | 1.0 | – | – | 0.88 | 0.91 | – | 0.7 | 0.7 | – | 0.88 | – | 0.84 | – | – | – | – | – | 0.85 |
| | Qwen2.5-72B | 0.923 | – | – | 0.813 | 0.874 | – | 0.591 | 0.605 | – | 0.875 | – | 0.765 | – | – | – | – | – | 0.688 |
| | Aya-Vision-32B | 0.154 | – | – | 0.738 | 0.788 | – | 0.538 | 0.333 | – | 0.500 | – | 0.535 | – | – | – | – | – | 0.561 |
| | Aya-Vision-8B | 0.154 | – | – | 0.525 | 0.677 | – | 0.515 | 0.290 | – | 0.375 | – | 0.392 | – | – | – | – | – | 0.428 |
| | Molmo-7B-D | 0.250 | – | – | 0.475 | 0.632 | – | 0.258 | 0.256 | – | 0.375 | – | 0.348 | – | – | – | – | – | 0.376 |
| | Pangea-7B | 0.182 | – | – | 0.550 | 0.652 | – | 0.379 | 0.354 | – | 0.500 | – | 0.397 | – | – | – | – | – | 0.337 |
| | Qwen2.5-7B | 0.462 | – | – | 0.600 | 0.759 | – | 0.470 | 0.387 | – | 0.250 | – | 0.529 | – | – | – | – | – | 0.543 |
| | Qwen2.5-3B | 0.308 | – | – | 0.600 | 0.680 | – | 0.523 | 0.331 | – | 0.375 | – | 0.352 | – | – | – | – | – | 0.370 |

Table 13: **Valid Accuracy (%) for Multimodal Questions for All Models, by Language and General Category.** Languages are grouped by script. Empty cells indicate either no questions for that category, or only format errors in the answers from the model.

| General Category | Model | Latin Script | | | | | | Non-Latin Script | | | | | | | | | | | |
|---|---|---|---|---|---|---|---|---|---|---|---|---|---|---|---|---|---|---|---|
| | | English | French | German | Dutch | Portuguese | Spanish | Arabic | Bengali | Croatian | Hindi | Hungarian | Lithuanian | Nepali | Persian | Russian | Serbian | Telugu | Ukrainian |
| Health Sciences | Gemini 1.5 Pro | – | – | – | 0.538 | 1.000 | 0.719 | – | – | – | – | – | – | – | – | – | – | – | – |
| | Claude 3.5 Sonnet | – | – | – | 0.538 | 1.000 | 0.761 | – | – | – | – | – | – | – | – | – | – | – | – |
| | GPT-4o | – | – | – | 0.565 | 1.000 | 0.779 | – | – | – | – | – | – | – | – | – | – | – | – |
| | Qwen2.5-72B | – | – | – | 0.500 | 1.000 | 0.648 | – | – | – | – | – | – | – | – | – | – | – | – |
| | Aya-Vision-32B | – | – | – | 0.481 | 1.000 | 0.524 | – | – | – | – | – | – | – | – | – | – | – | – |
| | Aya-Vision-8B | – | – | – | 0.286 | 0.500 | 0.400 | – | – | – | – | – | – | – | – | – | – | – | – |
| | Molmo-7B-D | – | – | – | 0.393 | 1.000 | 0.400 | – | – | – | – | – | – | – | – | – | – | – | – |
| | Pangea-7B | – | – | – | 0.423 | 1.000 | 0.372 | – | – | – | – | – | – | – | – | – | – | – | – |
| | Qwen2.5-7B | – | – | – | 0.429 | 1.000 | 0.457 | – | – | – | – | – | – | – | – | – | – | – | – |
| | Qwen2.5-3B | – | – | – | 0.250 | 1.000 | 0.376 | – | – | – | – | – | – | – | – | – | – | – | – |
| Humanities & Culture | Gemini 1.5 Pro | 1.000 | – | – | 0.691 | 0.902 | 0.872 | 0.75 | – | – | – | – | 0.805 | – | – | – | – | – | 0.701 |
| | Claude 3.5 Sonnet | 1.000 | – | – | 0.794 | 0.893 | 0.872 | 0.375 | – | – | – | – | 0.878 | – | – | – | – | – | 0.792 |
| | GPT-4o | 1.000 | – | – | 0.785 | 0.912 | 0.863 | 0.5 | – | – | – | – | 0.843 | – | – | – | – | – | 0.839 |
| | Qwen2.5-72B | 0.667 | – | – | 0.731 | 0.908 | 0.888 | 0.625 | – | – | – | – | 0.720 | – | – | – | – | – | 0.701 |
| | Aya-Vision-32B | 0.500 | – | – | 0.687 | 0.779 | 0.737 | 0.500 | – | – | – | – | 0.622 | – | – | – | – | – | 0.490 |
| | Aya-Vision-8B | 0.500 | – | – | 0.529 | 0.686 | 0.659 | 0.375 | – | – | – | – | 0.463 | – | – | – | – | – | 0.352 |
| | Molmo-7B-D | 0.167 | – | – | 0.574 | 0.644 | 0.626 | 0.250 | – | – | – | – | 0.341 | – | – | – | – | – | 0.323 |
| | Pangea-7B | 0.333 | – | – | 0.529 | 0.640 | 0.736 | 0.250 | – | – | – | – | 0.375 | – | – | – | – | – | 0.348 |
| | Qwen2.5-7B | 0.333 | – | – | 0.603 | 0.776 | 0.832 | 0.500 | – | – | – | – | 0.488 | – | – | – | – | – | 0.424 |
| | Qwen2.5-3B | 0.167 | – | – | 0.574 | 0.699 | 0.816 | 0.375 | – | – | – | – | 0.402 | – | – | – | – | – | 0.420 |
| General Knowledge | Gemini 1.5 Pro | 0.526 | – | – | – | – | 0.625 | – | 0.917 | – | 0.827 | – | – | – | 0.577 | – | – | – | 0.557 |
| | Claude 3.5 Sonnet | 0.5 | – | – | – | – | 0.875 | – | 0.917 | – | 0.707 | – | – | – | 0.673 | – | – | – | 0.567 |
| | GPT-4o | 0.462 | – | – | – | – | 1.000 | – | 0.917 | – | 0.795 | – | – | – | 0.8 | – | – | – | 0.636 |
| | Qwen2.5-72B | 0.474 | – | – | – | – | 0.875 | – | 0.917 | – | 0.632 | – | – | – | 0.442 | – | – | – | 0.493 |
| | Aya-Vision-32B | 0.474 | – | – | – | – | 0.625 | – | 0.833 | – | 0.571 | – | – | – | 0.404 | – | – | – | 0.388 |
| | Aya-Vision-8B | 0.316 | – | – | – | – | 0.875 | – | 0.667 | – | 0.586 | – | – | – | 0.404 | – | – | – | 0.373 |
| | Molmo-7B-D | 0.474 | – | – | – | – | 0.875 | – | 0.250 | – | 0.462 | – | – | – | 0.519 | – | – | – | 0.413 |
| | Pangea-7B | 0.429 | – | – | – | – | 0.375 | – | 0.917 | – | 0.476 | – | – | – | 0.347 | – | – | – | 0.311 |
| | Qwen2.5-7B | 0.474 | – | – | – | – | 0.625 | – | 0.833 | – | 0.557 | – | – | – | 0.442 | – | – | – | 0.373 |
| | Qwen2.5-3B | 0.526 | – | – | – | – | 0.500 | – | 0.583 | – | 0.391 | – | – | – | 0.462 | – | – | – | 0.358 |
| Reasoning | Gemini 1.5 Pro | 0.31 | – | – | – | – | – | – | 0.333 | – | 0.67 | – | – | 0.228 | – | – | – | – | – |
| | Claude 3.5 Sonnet | 0.355 | – | – | – | – | – | – | 0.289 | – | 0.667 | – | – | – | 0.28 | – | – | – | – |
| | GPT-4o | 0.396 | – | – | – | – | – | – | 0.372 | – | 0.648 | – | – | – | 0.205 | – | – | – | – |
| | Qwen2.5-72B | 0.311 | – | – | – | – | – | – | 0.422 | – | 0.502 | – | – | – | 0.238 | – | – | – | – |
| | Aya-Vision-32B | 0.284 | – | – | – | – | – | – | 0.244 | – | 0.295 | – | – | – | 0.222 | – | – | – | – |
| | Aya-Vision-8B | 0.176 | – | – | – | – | – | – | 0.178 | – | 0.304 | – | – | – | 0.214 | – | – | – | – |
| | Molmo-7B-D | 0.176 | – | – | – | – | – | – | 0.311 | – | 0.296 | – | – | – | 0.254 | – | – | – | – |
| | Pangea-7B | 0.203 | – | – | – | – | – | – | 0.308 | – | 0.280 | – | – | – | 0.239 | – | – | – | – |
| | Qwen2.5-7B | 0.230 | – | – | – | – | – | – | 0.333 | – | 0.326 | – | – | – | 0.222 | – | – | – | – |
| | Qwen2.5-3B | 0.230 | – | – | – | – | – | – | 0.178 | – | 0.303 | – | – | – | 0.238 | – | – | – | – |
| STEM | Gemini 1.5 Pro | 0.658 | 0.552 | 0.526 | 0.583 | 0.774 | 0.785 | 0.438 | 0.513 | 0.481 | 0.542 | 0.405 | 0.775 | – | 0.412 | 0.462 | 0.436 | 0.583 | 0.76 |
| | Claude 3.5 Sonnet | 0.671 | 0.517 | 0.737 | 0.598 | 0.784 | 0.735 | 0.491 | 0.42 | 0.475 | 0.45 | 0.389 | 0.812 | – | 0.444 | 0.454 | 0.396 | 0.56 | 0.769 |
| | GPT-4o | 0.666 | 0.492 | 0.537 | 0.613 | 0.731 | 0.746 | 0.489 | 0.628 | 0.404 | 0.489 | 0.407 | 0.848 | – | 0.405 | 0.403 | 0.387 | 0.479 | 0.768 |
| | Qwen2.5-72B | 0.556 | 0.466 | 0.493 | 0.499 | 0.590 | 0.683 | 0.344 | 0.438 | 0.333 | 0.378 | 0.376 | 0.638 | – | 0.349 | 0.393 | 0.359 | 0.348 | 0.686 |
| | Aya-Vision-32B | 0.333 | 0.295 | 0.395 | 0.395 | 0.434 | 0.470 | 0.290 | 0.254 | 0.296 | 0.304 | 0.265 | 0.413 | – | 0.306 | 0.294 | 0.285 | 0.348 | 0.459 |
| | Aya-Vision-8B | 0.302 | 0.265 | 0.357 | 0.369 | 0.366 | 0.445 | 0.270 | 0.242 | 0.204 | 0.284 | 0.267 | 0.338 | – | 0.297 | 0.290 | 0.268 | 0.346 | 0.405 |
| | Molmo-7B-D | 0.275 | 0.332 | 0.199 | 0.381 | 0.331 | 0.439 | 0.202 | 0.284 | 0.309 | 0.259 | 0.255 | 0.400 | – | 0.270 | 0.292 | 0.272 | 0.339 | 0.347 |
| | Pangea-7B | 0.284 | 0.310 | 0.196 | 0.326 | 0.401 | 0.434 | 0.218 | 0.302 | 0.210 | 0.250 | 0.262 | 0.299 | – | 0.247 | 0.289 | 0.301 | 0.339 | 0.348 |
| | Qwen2.5-7B | 0.373 | 0.354 | 0.266 | 0.413 | 0.455 | 0.517 | 0.356 | 0.295 | 0.261 | 0.287 | 0.286 | 0.563 | – | 0.295 | 0.288 | 0.275 | 0.376 | 0.482 |
| | Qwen2.5-3B | 0.360 | 0.325 | 0.211 | 0.363 | 0.382 | 0.483 | 0.313 | 0.340 | 0.264 | 0.299 | 0.292 | 0.375 | – | 0.295 | 0.290 | 0.281 | 0.318 | 0.426 |
| Social Sciences | Gemini 1.5 Pro | 1.000 | – | – | 0.784 | 0.882 | – | 0.4 | 0.57 | – | – | – | 0.713 | – | – | – | – | – | 0.763 |
| | Claude 3.5 Sonnet | 0.9 | – | – | 0.838 | 0.902 | – | 0.65 | 0.624 | – | – | – | 0.785 | – | – | – | – | – | 0.867 |
| | GPT-4o | 1.000 | – | – | 0.867 | 0.877 | – | 0.562 | 0.678 | – | – | – | 0.756 | – | – | – | – | – | 0.849 |
| | Qwen2.5-72B | 0.923 | – | – | 0.811 | 0.843 | – | 0.400 | 0.557 | – | – | – | 0.702 | – | – | – | – | – | 0.688 |
| | Aya-Vision-32B | 0.154 | – | – | 0.716 | 0.673 | – | 0.350 | 0.228 | – | – | – | 0.455 | – | – | – | – | – | 0.561 |
| | Aya-Vision-8B | 0.154 | – | – | 0.514 | 0.608 | – | 0.450 | 0.242 | – | – | – | 0.320 | – | – | – | – | – | 0.428 |
| | Molmo-7B-D | 0.250 | – | – | 0.432 | 0.601 | – | 0.300 | 0.215 | – | – | – | 0.337 | – | – | – | – | – | 0.376 |
| | Pangea-7B | 0.182 | – | – | 0.527 | 0.570 | – | 0.250 | 0.287 | – | – | – | 0.331 | – | – | – | – | – | 0.337 |
| | Qwen2.5-7B | 0.462 | – | – | 0.581 | 0.660 | – | 0.350 | 0.389 | – | – | – | 0.483 | – | – | – | – | – | 0.543 |
| | Qwen2.5-3B | 0.308 | – | – | 0.595 | 0.614 | – | 0.526 | 0.338 | – | – | – | 0.326 | – | – | – | – | – | 0.370 |

Table 14: **Valid Accuracy (%) for All Models, by Language and Multimodal Content Type.** Languages are grouped by script. Empty cells indicate either no questions for that category, or only format errors in the answers from the model.

| General Category | Model | Latin Script | | | | | | Non-Latin Script | | | | | | | | | | | |
|---|---|---|---|---|---|---|---|---|---|---|---|---|---|---|---|---|---|---|---|
| | | English | French | German | Dutch | Portuguese | Spanish | Arabic | Bengali | Croatian | Hindi | Hungarian | Lithuanian | Nepali | Persian | Russian | Serbian | Telugu | Ukrainian |
| Diagram | Gemini 1.5 Pro | 0.535 | – | 0.531 | 0.601 | 0.769 | 0.833 | 0.311 | 0.483 | – | 0.609 | – | 0.750 | 0.228 | 0.667 | – | – | – | 0.734 |
| | Claude 3.5 Sonnet | 0.532 | – | 0.739 | 0.593 | 0.744 | 0.750 | 0.40 | 0.586 | – | 0.592 | – | 0.643 | 0.280 | 1.000 | – | – | – | 0.751 |
| | GPT-4o | 0.539 | – | 0.560 | 0.636 | 0.750 | 0.875 | 0.333 | 0.724 | – | 0.576 | – | 0.643 | 0.205 | 0.667 | – | – | – | 0.728 |
| | Qwen2.5-72B | 0.461 | – | 0.491 | 0.543 | 0.648 | 0.708 | 0.244 | 0.517 | – | 0.479 | – | 0.750 | 0.238 | 0.667 | – | – | – | 0.661 |
| | Aya-Vision-32B | 0.334 | – | 0.400 | 0.416 | 0.490 | 0.625 | 0.273 | 0.241 | – | 0.365 | – | 0.393 | 0.222 | 0.333 | – | – | – | 0.452 |
| | Aya-Vision-8B | 0.265 | – | 0.352 | 0.364 | 0.395 | 0.542 | 0.133 | 0.241 | – | 0.369 | – | 0.321 | 0.214 | 0.333 | – | – | – | 0.407 |
| | Molmo-7B-D | 0.271 | – | 0.201 | 0.399 | 0.349 | 0.583 | 0.089 | 0.207 | – | 0.331 | – | 0.393 | 0.254 | 0.667 | – | – | – | 0.318 |
| | Pangea-7B | 0.269 | – | 0.180 | 0.378 | 0.436 | 0.522 | 0.244 | 0.125 | – | 0.313 | – | 0.429 | 0.239 | 0.667 | – | – | – | 0.322 |
| | Qwen2.5-7B | 0.325 | – | 0.274 | 0.491 | 0.481 | 0.792 | 0.333 | 0.276 | – | 0.379 | – | 0.500 | 0.222 | 0.333 | – | – | – | 0.446 |
| | Qwen2.5-3B | 0.344 | – | 0.204 | 0.382 | 0.407 | 0.583 | 0.311 | 0.310 | – | 0.316 | – | 0.214 | 0.238 | 0.000 | – | – | – | 0.424 |
| Figure | Gemini 1.5 Pro | 0.512 | 0.548 | – | 0.677 | 0.941 | 0.762 | 0.368 | 0.435 | 0.481 | 0.500 | 0.405 | 0.815 | – | 0.419 | 0.462 | 0.436 | 0.583 | 0.679 |
| | Claude 3.5 Sonnet | 0.562 | 0.515 | – | 0.677 | 0.928 | 0.718 | 0.368 | 0.441 | 0.475 | 0.467 | 0.389 | 0.800 | – | 0.460 | 0.454 | 0.396 | 0.560 | 0.738 |
| | GPT-4o | 0.530 | 0.490 | – | 0.689 | 0.906 | 0.733 | 0.417 | 0.532 | 0.404 | 0.600 | 0.407 | 0.828 | – | 0.428 | 0.403 | 0.387 | 0.479 | 0.790 |
| | Qwen2.5-72B | 0.450 | 0.470 | – | 0.635 | 0.889 | 0.667 | 0.316 | 0.359 | 0.333 | 0.467 | 0.376 | 0.754 | – | 0.354 | 0.393 | 0.359 | 0.470 | 0.646 |
| | Aya-Vision-32B | 0.335 | 0.298 | – | 0.547 | 0.758 | 0.448 | 0.211 | 0.338 | 0.296 | 0.133 | 0.265 | 0.662 | – | 0.305 | 0.294 | 0.285 | 0.348 | 0.443 |
| | Aya-Vision-8B | 0.343 | 0.264 | – | 0.531 | 0.673 | 0.444 | 0.263 | 0.261 | 0.204 | 0.267 | 0.267 | 0.400 | – | 0.279 | 0.292 | 0.272 | 0.339 | 0.430 |
| | Molmo-7B-D | 0.273 | 0.333 | – | 0.531 | 0.595 | 0.413 | 0.211 | 0.273 | 0.309 | 0.200 | 0.255 | 0.385 | – | 0.301 | 0.292 | 0.272 | 0.339 | 0.370 |
| | Pangea-7B | 0.305 | 0.316 | – | 0.429 | 0.596 | 0.429 | 0.167 | 0.387 | 0.210 | 0.071 | 0.262 | 0.297 | – | 0.245 | 0.289 | 0.301 | 0.339 | 0.370 |
| | Qwen2.5-7B | 0.343 | 0.353 | – | 0.479 | 0.739 | 0.500 | 0.263 | 0.311 | 0.261 | 0.133 | 0.286 | 0.523 | – | 0.301 | 0.288 | 0.275 | 0.376 | 0.473 |
| | Qwen2.5-3B | 0.347 | 0.329 | – | 0.469 | 0.641 | 0.460 | 0.263 | 0.317 | 0.264 | 0.200 | 0.292 | 0.400 | – | 0.302 | 0.290 | 0.281 | 0.318 | 0.401 |
| Graph | Gemini 1.5 Pro | 0.722 | 1.000 | 0.488 | 0.505 | 0.757 | 0.846 | 0.483 | 0.557 | – | 0.870 | – | 0.745 | – | – | – | – | – | 0.778 |
| | Claude 3.5 Sonnet | 0.739 | 0.500 | 0.721 | 0.574 | 0.798 | 0.885 | 0.533 | 0.577 | – | 0.870 | – | 0.830 | – | – | – | – | – | 0.917 |
| | GPT-4o | 0.654 | 0.500 | 0.372 | 0.56 | 0.718 | 0.885 | 0.643 | 0.585 | – | 0.846 | – | 0.793 | – | – | – | – | – | 0.817 |
| | Qwen2.5-72B | 0.514 | 0.000 | 0.512 | 0.412 | 0.623 | 0.769 | 0.433 | 0.588 | – | 0.652 | – | 0.723 | – | – | – | – | – | 0.694 |
| | Aya-Vision-32B | 0.361 | 0.500 | 0.357 | 0.410 | 0.500 | 0.615 | 0.400 | 0.227 | – | 0.348 | – | 0.457 | – | – | – | – | – | 0.500 |
| | Aya-Vision-8B | 0.270 | 0.500 | 0.395 | 0.320 | 0.443 | 0.577 | 0.433 | 0.258 | – | 0.356 | – | 0.404 | – | – | – | – | – | 0.403 |
| | Molmo-7B-D | 0.432 | 0.000 | 0.186 | 0.437 | 0.538 | 0.367 | 0.268 | – | – | 0.348 | – | 0.330 | – | – | – | – | – | 0.361 |
| | Pangea-7B | 0.235 | 0.000 | 0.324 | 0.313 | 0.397 | 0.423 | 0.286 | 0.247 | – | 0.333 | – | 0.348 | – | – | – | – | – | 0.304 |
| | Qwen2.5-7B | 0.486 | 0.000 | 0.209 | 0.359 | 0.514 | 0.615 | 0.333 | 0.392 | – | 0.348 | – | 0.521 | – | – | – | – | – | 0.528 |
| | Qwen2.5-3B | 0.432 | 0.000 | 0.256 | 0.369 | 0.492 | 0.654 | 0.333 | 0.354 | – | 0.370 | – | 0.319 | – | – | – | – | – | 0.431 |
| Map | Gemini 1.5 Pro | 1.000 | – | – | 0.667 | 0.892 | 0.719 | 0.444 | 0.583 | – | 1.000 | – | 0.646 | – | – | – | – | – | 0.677 |
| | Claude 3.5 Sonnet | 1.000 | – | – | 0.833 | 0.892 | – | 0.611 | 0.667 | – | 1.000 | – | 0.835 | – | – | – | – | – | 0.783 |
| | GPT-4o | 1.000 | – | – | 0.769 | 0.841 | – | 0.533 | 0.833 | – | – | – | 0.766 | – | – | – | – | – | 0.802 |
| | Qwen2.5-72B | 1.000 | – | – | 0.722 | 0.923 | – | 0.389 | 0.583 | – | 0.000 | – | 0.658 | – | – | – | – | – | 0.601 |
| | Aya-Vision-32B | 0.000 | – | – | 0.556 | 0.662 | – | 0.389 | 0.333 | – | 0.000 | – | 0.506 | – | – | – | – | – | 0.465 |
| | Aya-Vision-8B | 0.000 | – | – | 0.333 | 0.677 | – | 0.389 | 0.250 | – | 0.000 | – | 0.367 | – | – | – | – | – | 0.384 |
| | Molmo-7B-D | 0.000 | – | – | 0.222 | 0.631 | – | 0.222 | 0.167 | – | 0.000 | – | 0.367 | – | – | – | – | – | 0.343 |
| | Pangea-7B | – | – | – | 0.444 | 0.556 | – | 0.278 | 0.222 | – | 0.000 | – | 0.377 | – | – | – | – | – | 0.347 |
| | Qwen2.5-7B | 0.000 | – | – | 0.444 | 0.708 | – | 0.389 | 0.333 | – | 0.000 | – | 0.456 | – | – | – | – | – | 0.439 |
| | Qwen2.5-3B | 1.000 | – | – | 0.611 | 0.569 | – | 0.529 | 0.250 | – | 0.000 | – | 0.342 | – | – | – | – | – | 0.369 |
| Photo | Gemini 1.5 Pro | – | – | – | 0.840 | 0.938 | 0.719 | 1.000 | – | – | 1.000 | – | 0.867 | – | – | – | – | – | 0.682 |
| | Claude 3.5 Sonnet | – | – | – | 0.860 | 0.938 | 0.761 | 0.667 | – | – | 1.000 | – | 0.867 | – | – | – | – | – | 0.699 |
| | GPT-4o | – | – | – | 0.927 | 0.969 | 0.779 | 0.500 | – | – | 1.000 | – | 0.909 | – | – | – | – | – | 0.743 |
| | Qwen2.5-72B | – | – | – | 0.800 | 0.875 | 0.648 | 1.000 | – | – | 1.000 | – | 0.600 | – | 0.000 | – | – | – | 0.661 |
| | Aya-Vision-32B | – | – | – | 0.680 | 0.821 | 0.524 | 1.000 | – | – | 1.000 | – | 0.667 | – | 1.000 | – | – | – | 0.540 |
| | Aya-Vision-8B | – | – | – | 0.580 | 0.688 | 0.400 | 1.000 | – | – | 0.000 | – | 0.333 | – | 0.000 | – | – | – | 0.395 |
| | Molmo-7B-D | – | – | – | 0.520 | 0.696 | 0.400 | 0.667 | – | – | 1.000 | – | 0.533 | – | 1.000 | – | – | – | 0.351 |
| | Pangea-7B | – | – | – | 0.560 | 0.786 | 0.372 | 0.333 | – | – | 0.000 | – | 0.357 | – | 1.000 | – | – | – | 0.336 |
| | Qwen2.5-7B | – | – | – | 0.600 | 0.795 | 0.457 | 0.667 | – | – | 1.000 | – | 0.667 | – | 1.000 | – | – | – | 0.464 |
| | Qwen2.5-3B | – | – | – | 0.540 | 0.777 | 0.376 | 0.667 | – | – | 1.000 | – | 0.467 | – | 1.000 | – | – | – | 0.418 |
| Formula | Gemini 1.5 Pro | 0.935 | – | – | 0.674 | 0.767 | 1.000 | 0.554 | 0.8 | – | 0.612 | – | 0.889 | – | – | – | – | – | 0.800 |
| | Claude 3.5 Sonnet | 0.75 | – | – | 0.771 | 0.900 | 0.667 | 0.554 | 0.391 | – | 0.374 | – | 0.944 | – | – | – | – | – | 0.633 |
| | GPT-4o | 0.909 | – | – | 0.711 | 0.800 | 0.667 | 0.531 | 0.864 | – | 0.529 | – | 1.000 | – | – | – | – | – | 0.724 |
| | Qwen2.5-72B | 0.625 | – | – | 0.604 | 0.700 | 0.333 | 0.411 | 0.553 | – | 0.381 | – | 0.667 | – | – | – | – | – | 0.667 |
| | Aya-Vision-32B | 0.419 | – | – | 0.489 | 0.567 | 0.667 | 0.268 | 0.128 | – | 0.300 | – | 0.333 | – | – | – | – | – | 0.267 |
| | Aya-Vision-8B | 0.344 | – | – | 0.396 | 0.367 | 0.333 | 0.304 | 0.234 | – | 0.278 | – | 0.278 | – | – | – | – | – | 0.267 |
| | Molmo-7B-D | 0.313 | – | – | 0.229 | 0.433 | 0.667 | 0.179 | 0.298 | – | 0.224 | – | 0.222 | – | – | – | – | – | 0.267 |
| | Pangea-7B | 0.417 | – | – | 0.356 | 0.567 | 0.333 | 0.170 | 0.333 | – | 0.265 | – | 0.412 | – | – | – | – | – | 0.393 |
| | Qwen2.5-7B | 0.375 | – | – | 0.417 | 0.700 | 0.333 | 0.321 | 0.435 | – | 0.244 | – | 0.611 | – | – | – | – | – | 0.400 |
| | Qwen2.5-3B | 0.406 | – | – | 0.354 | 0.300 | 0.333 | 0.286 | 0.404 | – | 0.323 | – | 0.500 | – | – | – | – | – | 0.433 |
| Table | Gemini 1.5 Pro | 0.880 | 0.625 | – | 0.471 | 0.875 | 0.795 | 0.333 | 0.566 | – | 0.716 | – | 0.756 | – | 0.474 | – | – | – | 0.719 |
| | Claude 3.5 Sonnet | 0.865 | 0.625 | – | 0.526 | 0.909 | 0.769 | 0.667 | 0.528 | – | 0.768 | – | 0.800 | – | 0.217 | – | – | – | 0.750 |
| | GPT-4o | 0.879 | 0.571 | – | 0.571 | 0.826 | 0.718 | 0.583 | 0.680 | – | 0.707 | – | 0.800 | – | 0.333 | – | – | – | 0.844 |
| | Qwen2.5-72B | 0.772 | 0.375 | – | 0.579 | 0.636 | 0.744 | 0.267 | 0.434 | – | 0.453 | – | 0.585 | – | 0.348 | – | – | – | 0.750 |
| | Aya-Vision-32B | 0.301 | 0.125 | – | 0.579 | 0.400 | 0.447 | 0.267 | 0.208 | – | 0.253 | – | 0.293 | – | 0.550 | – | – | – | 0.469 |
| | Aya-Vision-8B | 0.262 | 0.250 | – | 0.211 | 0.398 | 0.462 | 0.333 | 0.208 | – | 0.232 | – | 0.244 | – | 0.182 | – | – | – | 0.281 |
| | Molmo-7B-D | 0.225 | 0.375 | – | 0.421 | 0.386 | 0.538 | 0.400 | 0.226 | – | 0.263 | – | 0.293 | – | 0.391 | – | – | – | 0.344 |
| | Pangea-7B | 0.254 | 0.143 | – | 0.222 | 0.395 | 0.462 | 0.200 | 0.340 | – | 0.284 | – | 0.175 | – | 0.474 | – | – | – | 0.156 |
| | Qwen2.5-7B | 0.435 | 0.500 | – | 0.421 | 0.432 | 0.436 | 0.667 | 0.377 | – | 0.326 | – | 0.415 | – | 0.348 | – | – | – | 0.344 |
| | Qwen2.5-3B | 0.337 | 0.250 | – | 0.368 | 0.471 | 0.487 | 0.467 | 0.264 | – | 0.263 | – | 0.390 | – | 0.391 | – | – | – | 0.250 |
| Text | Gemini 1.5 Pro | 1.000 | – | – | 0.500 | 0.844 | 0.872 | 0.600 | – | – | 1.000 | – | 1.000 | – | – | – | – | – | 0.800 |
| | Claude 3.5 Sonnet | 1.000 | – | – | 1.000 | 0.867 | 0.866 | 0.200 | – | – | 1.000 | – | – | – | – | – | – | – | 0.867 |
| | GPT-4o | 1.000 | – | – | 1.000 | 0.858 | 0.858 | 0.500 | 1.000 | – | 1.000 | – | – | – | – | – | – | – | 0.917 |
| | Qwen2.5-72B | 1.000 | – | – | 1.000 | 0.844 | 0.888 | 0.400 | 1.000 | – | 0.000 | – | – | – | – | – | – | – | 0.733 |
| | Aya-Vision-32B | 0.000 | – | – | 1.000 | 0.711 | 0.720 | 0.200 | 0.000 | – | 1.000 | – | – | – | – | – | – | – | 0.400 |
| | Aya-Vision-8B | 0.000 | – | – | 1.000 | 0.636 | 0.636 | 0.000 | 0.000 | – | 0.000 | – | – | – | – | – | – | – | 0.133 |
| | Molmo-7B-D | 0.000 | – | – | 0.500 | 0.578 | 0.615 | 0.000 | 0.000 | – | 1.000 | – | – | – | – | – | – | – | 0.200 |
| | Pangea-7B | 0.000 | – | – | 1.000 | 0.578 | 0.714 | 0.200 | – | – | 0.000 | – | – | – | – | – | – | – | 0.462 |
| | Qwen2.5-7B | 0.000 | – | – | 1.000 | 0.778 | 0.818 | 0.400 | 0.000 | – | 0.000 | – | – | – | – | – | – | – | 0.267 |
| | Qwen2.5-3B | 0.000 | – | – | 1.000 | 0.622 | 0.797 | 0.200 | 1.000 | – | 0.000 | – | – | – | – | – | – | – | 0.400 |

Table 15: **Format error disaggregated analysis.** For answers which were identified as Format Error, we report how many correspond to answer refusal (R) vs another type of format errors (NR). Empty N-NR refusal pairs indicate that the model did not have Format Errors on that language. Note that for most models in all languages, format errors are not due to answer refusals but to other reasons like inability to follow MCQ tagging instructions. The only exception is Pangea, which seems to exhibit a higher refusal rate.

| Model ↔ Language ↕ | Aya 8b | | Aya 32b | | Claude 3.5 Sonnet | | Gemini 1.5 pro | | GPT-4o | | Molmo | | Pangea | | Qwen2.5 7b | | Qwen2.5 3b | | Qwen2.5 32b | | Qwen2.5 72b | |
|---|---|---|---|---|---|---|---|---|---|---|---|---|---|---|---|---|---|---|---|---|---|---|
| | R | NR | R | NR | R | NR | R | NR | R | NR | R | NR | R | NR | R | NR | R | NR | R | NR | R | NR |
| Arabic | | | 0 | 2 | | | 1 | 1 | 5 | 27 | | | 14 | 2 | | | 0 | 1 | | | | |
| Bengali | 0 | 204 | 0 | 3 | 1 | 0 | 0 | 7 | 2 | 18 | | | 38 | 13 | 1 | 0 | 0 | 1 | | | | |
| Croatian | | | 0 | 1 | 1 | 1 | 0 | 6 | 4 | 14 | | | 17 | 7 | 0 | 1 | 0 | 3 | 0 | 15 | | |
| Dutch; Flemish | | | 0 | 19 | 2 | 1 | 8 | 27 | 17 | 40 | | | 13 | 9 | | | | | 0 | 13 | 0 | 1 |
| English | | | 0 | 11 | 1 | 339 | 0 | 9 | 1 | 278 | 0 | 2 | 71 | 23 | | | | | 0 | 2 | | |
| French | | | 0 | 24 | | | 0 | 14 | 0 | 33 | 0 | 3 | 27 | 42 | 0 | 1 | 0 | 1 | 0 | 20 | 0 | 2 |
| German | 0 | | 0 | 4 | | | | | 0 | 10 | | | 41 | 3 | | | 0 | 1 | 0 | 3 | | |
| Hindi | 0 | 6 | 0 | 24 | 1 | 0 | 0 | 18 | 5 | 110 | 0 | 3 | 127 | 56 | 0 | 2 | | | 0 | 20 | 0 | 1 |
| Hungarian | | | 0 | 14 | 1 | 1 | 0 | 30 | 11 | 64 | | | 23 | 26 | 0 | 1 | 0 | 18 | 0 | 19 | 0 | 1 |
| Lithuanian | | | 0 | 7 | 0 | 1 | | | 12 | 3 | | | 6 | 2 | | | | | | | | |
| Nepali | 0 | 2 | | | 0 | 1 | 0 | 3 | 5 | 4 | | | 27 | 7 | | | | | | | | |
| Persian | 0 | 6 | 0 | 31 | | | 0 | 38 | 3 | 36 | | | 116 | 44 | | | 0 | 3 | | | 0 | 1 |
| Portuguese | | | 0 | 29 | | | 0 | 35 | 0 | 60 | | | 24 | 19 | 0 | 7 | 0 | 2 | 0 | 36 | | |
| Russian | | | 0 | 13 | 0 | 2 | 0 | 84 | 0 | 96 | | | 67 | 38 | | | | | 0 | 11 | | |
| Serbian | | | 0 | 6 | 3 | 15 | 0 | 50 | 7 | 205 | | | 74 | 51 | | | 0 | 11 | 0 | 41 | | |
| Spanish | | | 0 | 31 | 0 | 1 | 0 | 2 | 9 | 36 | | | 42 | 6 | 0 | 4 | | | 0 | 5 | | |
| Telugu | 75 | 493 | 0 | 1 | | | 0 | 4 | 6 | 126 | | | 309 | 143 | | | | | | | | |
| Ukrainian | 0 | 7 | | | 0 | 1 | 0 | 1 | 44 | 73 | 0 | 1 | 20 | 6 | | | | | | | | |

## D.4 FORMAT ERRORS

We report the distribution of format errors for all evaluated models in Figure 5. Table 15 provides a finer-grained breakdown, showing which format errors correspond to refusals versus missing answers.

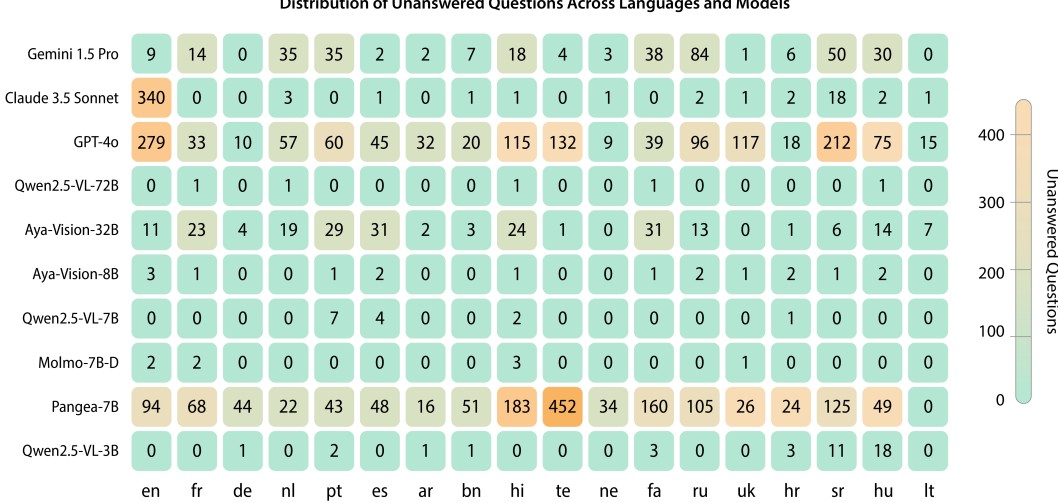

Figure 5: **Distribution of the number of format errors for each model/language combination**. The languages are represented in their ISO 639 (set 1) code.

## D.5 PROMPTS

The prompts that we used to perform all experiments were designed to ensure consistency across languages. Examples are shown both in English and Spanish as an overview. Below is a summary of the key components.

### D.5.1 SYSTEM MESSAGE

A system message sets the context for the model, instructing it to act as an expert in solving multiple-choice questions. For zero-shot CoT prompting, the message is provided in all the evaluation languages to support language-specific evaluation.

- **Zero-shot CoT**:

  - English:        You are an expert at solving multiple-choice
    questions.  Carefully analyze the question, think step
    by step, and provide your FINAL answer between the tags
    <ANSWER> X </ANSWER>, where X is ONLY the correct choice.
    Do not write any additional text between the tags.

  - Spanish:    Eres un experto en resolver preguntas de opción
    múltiple.  Analiza cuidadosamente la pregunta, piensa
    paso a paso y proporciona tu respuesta FINAL entre las
    etiquetas <ANSWER> X </ANSWER>, donde X es ÚNICAMENTE
    la opción correcta.  No escribas ningún texto adicional
    entre las etiquetas.

- **Direct answer**:

  You are a helpful assistant who answers multiple-choice
  questions.  For each question, output your final answer
  in JSON format with the following structure:  {"choice":
  "The correct option (e.g., A, B, C, or D)"}.  ONLY output
  this format exactly.  Do not include any additional text or
  explanations outside the JSON structure.

### D.5.2    KEYWORDS

Language-specific keywords are used to structure the prompts consistently across languages. These include terms for "Question," "Options," and "Answer" to be included when generating the prompt. For example:

- English:        {"question":  "Question", "options":  "Options",
  "answer":  "Answer"}

- Spanish:        {"question":  "Pregunta", "options":  "Opciones",
  "answer":  "Respuesta"}

### D.5.3    PROMPT EXAMPLES

System messages D.5.1 and Keywords D.5.2 are used to systematically craft the prompt for a model in a specific language. We show examples of both a closed and an open model in Table 16.

Table 16: **Prompt examples in** KALEIDOSCOPE. Multimodal prompt samples with interleaved image are shown for an open model and a closed model.

| Open Model | Closed Model |
|---|---|
| SYSTEM:
You are a helpful assistant who answers multiple-choice questions. For each question, output your final answer in JSON format with the following structure: "choice":"The correct option (e.g., A, B, C, or D)". ONLY output this format exactly. Do not include any additional text or explanations outside the JSON structure. Output your choice in the specified JSON format.

USER:

Question: Make CORRECT match between Group-I and Group-II, in relation to interaction between two species.
Options:
A.) P-I, Q-II, R-III, S-IV
B.) P-III, Q-II, R-IV, S-I
C.) P-IV, Q-III, R-II, S-I
D.) P-III, Q-IV, R-II, S-I
Answer: | SYSTEM:
Eres un experto en resolver preguntas de opción múltiple. Analiza cuidadosamente la pregunta, piensa paso a paso y proporciona tu respuesta FINAL entre las etiquetas <ANSWER> X </ANSWER>, donde X es ÚNICAMENTE la opción correcta. No escribas ningún texto adicional entre las etiquetas.
USER:
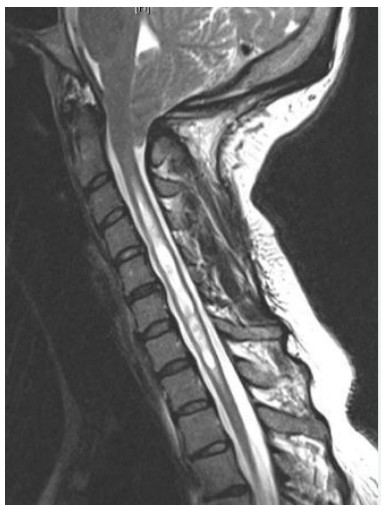
Pregunta: Ante esta imagen en un paciente con un trastorno motor en miembros inferiores, señale la respuesta INCORRECTA:
Opciones:
A.) Debemos buscar una malformación de Chiari
B.) En algunos casos se asocia a hidrocefalia
C.) Se caracteriza por una pérdida de la sensibilidad táctil y vibratoria con preservación de la sensación térmica y dolorosa
D.) Puede producirse tras traumatismos o infecciones

Respuesta: |

The embedded table in the Open Model USER prompt:

| Group-I | Group-II |
|---|---|
| **P**. Neutralism | **I**. neither can survive under natural condition without the other |
| **Q**. Allelopathy | **II**. direct inhibition of one species by the other species using toxic compound |
| **R**. Amensalism | **III**. neither is affected by the association with the other |
| **S**. Mutalism | **IV**. one is inhibited and the other is not affected |

Table 17: **Caption+OCR prompt examples in** KALEIDOSCOPE. Prompts are shown for open and closed models in English and Spanish. Caption and OCR additions are highlighted in green.

| Open Model | Closed Model |
|---|---|
| **SYSTEM:**
You are a helpful assistant who answers multiple-choice questions. For each question, output your final answer in JSON format with the following structure: "choice":"The correct option (e.g., A, B, C, or D)". ONLY output this format exactly. Do not include any additional text or explanations outside the JSON structure.

**USER:**

```c
#include<stdio.h>

int main(int argc, char *argv[]){
    char a = 'P';
    char b = 'x';
    char c = (a & b) + '*';
    char d = (a | b) - '-';
    char e = (a ^ b) + '+';
    printf("%c %c %c\n", c, d, e);
    return 0;
}
```
ASCII encoding for relevant characters is given below
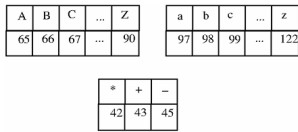

**Caption:** The code initializes character variables 'a' to 'P' and 'b' to 'x'. It then calculates 'c', 'd', and 'e' using bitwise operations (\&, \|, ^) and character addition with '*', '-', and '+', respectively. The 'printf' function (...)[a] Ellipses (...) within the tables indicate omitted values between the shown characters.
**OCR:** ##include<stdio.h>\}\n \nint main(int argc,\n \nchar a = 'P';\nchar b = 'x';\nchar c = (a &\nchar d = (a |\nchar e = (a \u 201c\n \nprintf (\"sc \%\nreturn 0;\n \n \}\n \nchar *argv[]) \{\n \nby + te;\nb ) - '-\%3\nb ) + \"Hy\n se\\\n\", c, d, e);\n \nASCII encoding for relevant characters is given below\n \n 42\| 43) 45\n \n
**Question:** What is printed by the following ANSI C program? Options: A.) z K s B.) 122 75 83 C.) * - + D.) P x +
Answer: | **SYSTEM:**
Eres un experto en resolver preguntas de opción múltiple. Analiza cuidadosamente la pregunta, piensa paso a paso y proporciona tu respuesta FINAL entre las etiquetas <ANSWER> X </ANSWER>, donde X es ÚNICAMENTE la opción correcta. No escribas ningún texto adicional entre las etiquetas.

**USER:**

| Mes | Peso total |
|---|---|
| 1 | 1.500 gramos |
| 2 | 2.600 gramos |
| 3 | 3.700 gramos |
| 4 | 4.800 gramos |

**Caption:** This table presents data on total weight, measured in grams, across four months. The table consists of two columns: M̈es¨(Month) and P̈eso total¨(Total Weight). Month 1 shows a weight of 1,500 grams, Month 2 shows 2,600 grams, Month 3 shows 3,700 grams, and Month 4 shows 4,800 grams. The table is a simple grid format with plain black text on a white background.
**OCR:** Wes \| Pesototal\n 1 [1500 gramos\n 2 \|_2600 grmos\n 3 \| 3700 gemoe\na \n \n \u 201cZOO grams\n \n
**Pregunta:** Un perro cachorro tenía un peso de 1.500 gramos al mes de nacido. En la tabla se muestra el peso del cachorro en los primeros cuatro meses. De acuerdo con la tabla, ¿Cuál es el cambio del peso del cachorro entre un mes y el mes siguiente?
Opciones:
A.) Disminuyó 1.500 gramos
B.) Disminuyó 2.600 gramos
C.) Aumentó 1.100 gramos
D.) Aumentó 3.300 gramos
Respuesta: |

[a] Caption was trimmed for visualization purposes.

## D.6 TO WHAT EXTENT DO TEXTUAL AUGMENTATIONS BOOST VLM CAPABILITIES?

The significant performance gap between text-only and multimodal responses raises critical questions about the strengths and weaknesses of the visual processing in the tested models. In this analysis, we investigate to what extent do visual processing constraints limit multimodal capabilities, and conversely, can automatically generated textual augmentation improve model performance?

To explore this direction, we generate synthetic captions (using Gemini 1.5 Pro) and Optical Character Recognition (OCR) text (Tesseract (Smith, 2007)) for all images in KALEIDOSCOPE, aligning with the methodology of (Das et al., 2024). Unlike prior work that completely replaces images with text, we evaluate whether a VLM *augmented* with these textual inputs can boost performance.

Table 18: **Accuracy on augmented multimodal inputs with image captions.** Results are grouped by image type. We report **Valid Accuracy (%)**; the highest scores are highlighted in bold for each model. Macro averaged accuracy is reported over language for both methods.

|  |  | Qwen2.5-VL-7B | | Gemini 1.5 Pro | |
| --- | --- | --- | --- | --- | --- |
|  | Samples | Image | +Caption | Image | +Caption |
| Diagram | 2,182 | **38.0** | 37.9 | 59.4 | **59.6** |
| Figure | 6,178 | 34.0 | **34.8** | **51.3** | 50.0 |
| Graph | 733 | 44.3 | **45.2** | 67.9 | **68.2** |
| Map | 392 | **48.0** | 46.7 | 69.4 | **70.9** |
| Photo | 631 | 53.9 | **54.1** | **75.8** | 74.3 |
| Formula | 487 | 34.9 | **37.3** | 68.3 | **68.7** |
| Table | 597 | **40.9** | 34.6 | 76.0 | **76.1** |
| Text | 257 | 76.3 | **79.8** | **85.2** | 83.7 |
| Macro Avg. | 11,457 | **36.88** | 36.83 | **55.71** | 54.81 |

Table 18 shows the results of augmenting visual inputs with synthetic captions and OCR text across diverse image types in KALEIDOSCOPE, measured by valid accuracy (%). Overall, the addition of a caption and OCR text improves the performance of the selected models in 5 out of 8 image types. Both models experienced a performance boost coordinately for *Graph* and *Formula*. The experiment reveals that the utility of textual augmentation depends critically on image content type. While Gemini 1.5 Pro dominates overall performance, Qwen2.5-VL-7B demonstrates selective gains when provided with captions and OCR: improvements in *Graph* (+0.9%), *Photo* (+0.2%), *Formula* (+2.4%), and *Text* (+3.5%) suggest that textual augmentation aids interpretation of content where visual elements are tightly coupled with symbolic or linguistic features (e.g., labeled axes, embedded text, or mathematical notation). Conversely, performance declines for *Diagram* (−0.1%), *Map* (−1.3%), and *Table* (−6.3%) with augmentation, implying that synthetic captions may introduce noise or fail to capture structural relationships critical to these categories. Gemini's robustness across modalities ($\leq 2\%$ variation in most categories) suggests its stronger native visual understanding reduces reliance on supplementary text. The results underscore that captioning effectiveness is context-dependent: text augmentation benefits models most when (1) visual content inherently contains extractable text (e.g., *Photo* with signs, *Text* regions) or (2) symbolic patterns (e.g., formulas, graphs) require disambiguation. However, for structurally complex or text-sparse images (e.g., *Map*, *Diagram*), captioning may not compensate for deficiencies in spatial or relational reasoning. Full results, including total accuracy and format error, can be found in Table 19.

Table 19: **Total Accuracy %**, **Valid Accuracy %** and **Format Error % (FE)** grouped by Image Type in KALEIDOSCOPE for captioning + OCR experiment.

| | Qwen2.5-VL-7B | | | Gemini 1.5 Pro | | |
|---|---|---|---|---|---|---|
| | *Total Acc.* | *Valid Acc.* | *FR* | *Total Acc.* | *Valid Acc.* | *FR* |
| Diagram | 37.8 | 37.9 | 0.3 | 58.9 | 59.6 | 1.2 |
| Figure | 34.6 | 34.8 | 0.7 | 49.0 | 50.0 | 1.9 |
| Graph | 45.2 | 45.2 | 0.0 | 67.4 | 68.2 | 1.2 |
| Map | 46.7 | 46.7 | 0.0 | 70.9 | 70.9 | 0.0 |
| Photo | 53.9 | 54.1 | 0.5 | 74.2 | 74.3 | 0.2 |
| Formula | 37.0 | 37.3 | 0.8 | 66.7 | 68.7 | 2.9 |
| Table | 34.5 | 34.6 | 0.2 | 74.7 | 76.1 | 1.8 |
| Text | 79.8 | 79.8 | 0.0 | 83.7 | 83.7 | 0.0 |

### D.6.1 CAPTIONING & OCR

We instantiated Gemini 1.5 Pro with the following instructions to generate synthetic captions from the images in KALEIDOSCOPE. Prompts with image augmentations are shown in Table 17.

**Gemini 1.5 Pro's prompt for captioning:**

```
**Instruction:**
You are an expert image captioner. Generate highly detailed, precise,
and academically relevant textual descriptions of images sourced from
exam questions, ensuring all critical visual elements are captured for
accurate problem-solving.

**Guidelines:**

Exam-Specific Analysis:

- Primary Elements: Identify and describe key components (e.g.,
diagrams, charts, graphs, labels, symbols, annotations) and their
exact attributes (e.g., numerical values, units, directional arrows,
text annotations).

- Secondary Details: Note stylistic features (e.g., "black-and-white
schematic," "color-coded bars in a graph"), spatial relationships
(e.g., "force vectors pointing northwest"), and contextual clues
(e.g., axes labels, legends, scales).

- Textual Elements: Explicitly transcribe all visible text (e.g.,
labels like "Mitochondria," numbers like "5V," titles like "Figure 2:
Velocity vs. Time").

Academic Precision:

- Technical Focus: Prioritize details critical to exam questions (e.g.,
"a right triangle with hypotenuse labeled c = 10 cm," "a bar graph
comparing GDP of 5 countries,with Japan's bar shaded blue at 4.3
trillion").

- Diagrams/Charts: Specify type (e.g., "pie chart," "circuit diagram")
and components (e.g., "resistor symbol connected to a battery").

- Scientific Relevance: Highlight measurements, units, symbols (e.g.,
"T = 25°C," "a pulley system with frictionless ropes").
```

Structure & Clarity:

– Begin with the image's purpose (e.g., "A biology diagram of a plant cell") followed by a systematic breakdown (left-to-right, top-to-bottom, or by functional layers).

– Use neutral, objective language. Avoid assumptions unless implied by context (e.g.,"a downward arrow labeled 9.8 m/s² likely representing gravitational acceleration").

**Output Format:**

– Single paragraph (4-6 sentences).

– Example:
"A physics diagram depicts two blocks on a frictionless inclined plane: Block A (5 kg) is connected via a rope to Block B (3 kg) over a pulley. Angle theta = 30°, with vectors labeled F_normal and F_gravity. A scale beside the plane shows time t = 0sto t = 5s. Text at the bottom reads: 'Calculate tension in the rope.' The image is monochrome, with dashed lines indicating motion direction."

Constraints:

– Avoid Omissions: Ensure no labels, numbers, or symbols are overlooked, even if small or peripheral.

– Neutral Tone: Exclude subjective interpretations (e.g., "messy handwriting" or "complex diagram") unless style is exam-relevant (e.g., "a hand-drawn sketch with annotations").

## D.7 CROSS-BENCHMARK PERFORMANCE

Table 20 presents a comparison of Kaleidoscope's overall valid accuracy with MMMU-Pro and MMMU-Validation on the overlapping subset of models. The results show that Kaleidoscope occupies a difficulty range comparable to existing high-quality multimodal benchmarks: most models perform slightly above their MMMU-Pro scores but below their MMMU-Validation scores. Importantly, the rank ordering of models is not identical across benchmarks. Several models with similar performance on MMMU diverge noticeably on Kaleidoscope, reflecting the benchmark's unique emphasis on culturally grounded, in-language multimodal reasoning. These discrepancies reinforce that Kaleidoscope is not merely a parallel version of existing datasets, but a complementary evaluation that probes multilingual and cross-cultural generalization more directly.

Table 20: **Cross-benchmark comparison on** KALEIDOSCOPE**, MMMU-Pro, and MMMU**. Results are reported as overall accuracy (%). KALEIDOSCOPE values use **valid accuracy**. Asterisks (*) indicate results provided directly by the original authors.

| Model | Kaleidoscope (Valid Acc.) | MMMU-Pro (Acc.) | MMMU Validation (Acc.) |
|---|---|---|---|
| Claude 3.5 Sonnet | 63.87 | 51.5 | 68.3 |
| Gemini 1.5 Pro | 62.95 | 46.9 | 65.8 |
| GPT-4o | 62.10 | 51.9 | 69.1 |
| Qwen2.5-VL-72B | 53.00 | 46.2 | 64.5 |
| Qwen2.5-VL-32B | 48.64 | N/A | N/A |
| Aya-Vision-32B | 39.66 | 45.11* | N/A |
| Aya-Vision-8B | 35.11 | 39.9* | N/A |
| Qwen2.5-VL-7B | 39.60 | 38.3* | 58.6* |
| Qwen2.5-VL-3B | 35.63 | 31.6* | 53.1* |
| Molmo-7B-D | 32.88 | N/A | N/A |
| Pangea-7B | 34.02 | N/A | N/A |

# E DATASET SAMPLES

## E.1 CATEGORIES OF VISUAL ELEMENTS

We group the visual elements into eight primary categories in KALEIDOSCOPE in Table 21. If an image falls into multiple categories, we assign the most representative based on the image's content.

## E.2 SELECTED DATASET SAMPLES

Table 22 presents one sample from each dataset, including the question, the associated image, the provided answers, and the correct answer highlighted in green.

## E.3 DATASET FIELDS

We provide a description of the fields in KALEIDOSCOPE in Table 23.

Table 21: Types of visual elements or images in the KALEIDOSCOPE benchmark. The correct answer is highlighted in **Bold Green**. Some samples are reformatted for better presentation.

| Visual Element Category | Question Image | Question and Answer |
|---|---|---|
| **Diagram.** Technical or schematic drawings illustrating processes, structures, or concepts. |  | **Question:** Wie verhält sich die Verarmungszone in der hier dargestellten Halbleiterdiode? **Options:** **A. Sie erweitert sich.** B. Sie verengt sich. C. Sie verändert sich nicht. D. Sie verschwindet. |
| **Figure.** Illustrations, drawings, or visual representations of objects, patterns, or symbols. |  | **Question:** Applicable for D of stem 'B'- **Options:** A. contains more genes B. unable to replicate C. present in the nucleus **D. used as a vector** |
| **Charts.** Images showing data plotted on axes, such as line graphs, bar charts, scatter plots, pie charts, flowcharts, organizational charts, and so on. |  | **Question:** Em uma xícara que já contém certa quantidade de açúcar, despeja-se café. A curva abaixo representa a função exponencial M(t), que fornece a quantidade de açúcar não dissolvido (em gramas), t minutos após o café ser despejado. Pelo gráfico, podemos concluir que. **Options:** **A.** $m(t) = 2^{(4-t/75)}$ B. $m(t) = 2^{(4-t/50)}$ C. $m(t) = 2^{(5-t/50)}$ D. $m(t) = 2^{(5-t/150)}$ |
| **Map.** Geographical or spatial representations. |  | **Question:** Діяльність якого гетьмана можна характеризувати, спираючись на подану карту? **Options:** A. Б. Хмельницького **B. I. Виговського** C. Д. Многогрішного D. I. Самойловича |

| Visual Element Category | Question Image | Question and Answer |
|---|---|---|
| **Photographs.** Photographic images of real-world scenes, objects, or people. | 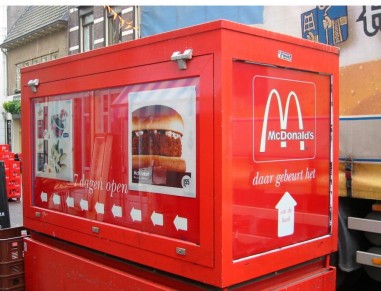 | **Question:** Wat kun je zeggen over het verzorgingsgebied van deze McDonald's in Arnhem?
**Options:**
**A. Het verzorgingsgebied beperkt zich tot de stad Arnhem.**
B. Het verzorgingsgebied beperkt zich tot de provincie Gelderland.
C. Het verzorgingsgebied beperkt zich tot Nederland.
D. Het verzorgingsgebied beperkt zich tot de regio Arnhem en omstreken. |
| **Formula.** Mathematical equations, chemical formulas, mathematical diagrams, or related concepts. | $$2\mathrm{HI}(g) \rightleftharpoons \mathrm{H}_2(g) + \mathrm{I}_2(g)$$ | **Question:** अभिक्रिया इस छवि में दिखाए गए समीकरण की विघटन की कोटि, साम्यावस्था स्थिरांक $\mathrm{Kp}$ में सम्बद्ध है।
**Options:**
A. $\sqrt{\frac{1+2K_p}{2}}$
B. $\frac{1+2K_p}{2}$
C. $\frac{2K_p}{1+2K_p}$
**D.** $\frac{2\sqrt{K_p}}{1+2\sqrt{K_p}}$ |
| **Table.** Structured data arranged in rows and columns. | | **Question:**
به چند طریق می‌توان جدول نیم‌پر روبه‌رو را با عددهای ۱ تا ۴ طوری پر کرد که در هیچ سطر و ستونی عدد تکراری نداشته باشیم؟
**Options:**
A. 0
B. 1
**C. 2**
D. !4 |
| **Text.** Images containing primarily textual information. | ABC ত্রিভুজে B কোণের পরিমাণ ৪৮° এবং AB=AC। | **Question:** যদি E এবং F AB এবং AC-কে এমনভাবে ছেদ করে যেন EF ‖ BC হয়, তাহলে
**Options:**
**A. ১৩২°**
B. ১৬০°
C. ১৮০°
D. ১০৮° |

Table 22: Samples from various exams in the KALEIDOSCOPE benchmark. The correct answer is highlighted in **Bold Green**. Some samples are reformatted for better presentation.

| Language | Question Image | Question and Answer |
|---|---|---|
| | | |

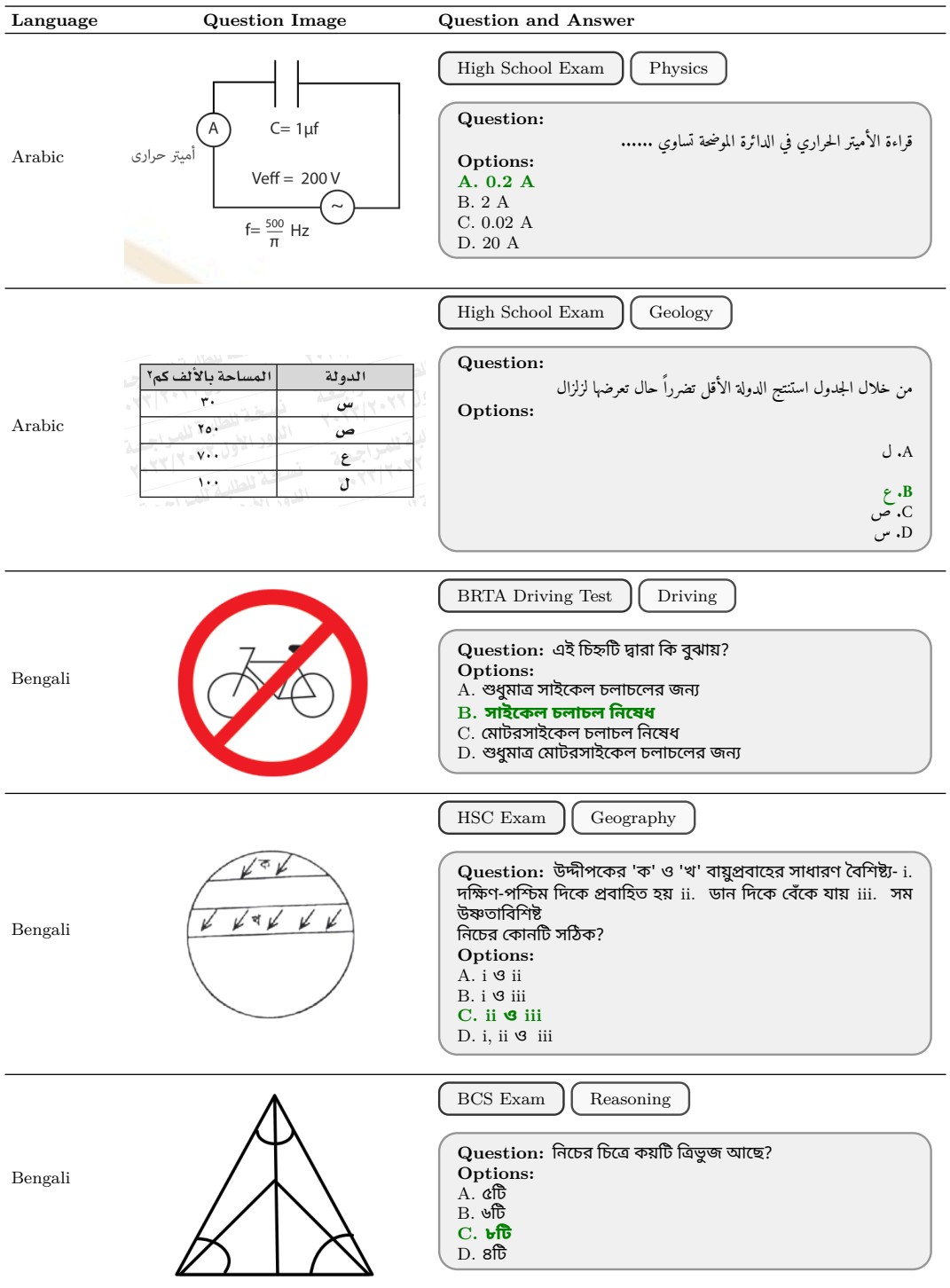

Arabic

High School Exam | Physics

**Question:**
قراءة الأميتر الحراري في الدائرة الموضحة تَساوي ......

**Options:**
**A. 0.2 A**
B. 2 A
C. 0.02 A
D. 20 A

---

Arabic

High School Exam | Geology

**Question:**
من خلال الجدول استنتج الدولة الأقل تضرراً حال تعرضها لزلزال

**Options:**

A. ل

**B. ع**
C. ص
D. س

---

Bengali

BRTA Driving Test | Driving

**Question:** এই চিহ্নটি দ্বারা কি বুঝায়?
**Options:**
A. শুধুমাত্র সাইকেল চলাচলের জন্য
**B. সাইকেল চলাচল নিষেধ**
C. মোটরসাইকেল চলাচল নিষেধ
D. শুধুমাত্র মোটরসাইকেল চলাচলের জন্য

---

Bengali

HSC Exam | Geography

**Question:** উদ্দীপকের 'ক' ও 'খ' বায়ুপ্রবাহের সাধারণ বৈশিষ্ট্য- i. দক্ষিণ-পশ্চিম দিকে প্রবাহিত হয় ii. ডান দিকে বেঁকে যায় iii. সম উষ্ণতাবিশিষ্ট
নিচের কোনটি সঠিক?
**Options:**
A. i ও ii
B. i ও iii
**C. ii ও iii**
D. i, ii ও iii

---

Bengali

BCS Exam | Reasoning

**Question:** নিচের চিত্রে কয়টি ত্রিভুজ আছে?
**Options:**
A. ৫টি
B. ৬টি
**C. ৮টি**
D. ৪টি

| Language | Question Image | Question and Answer |
|---|---|---|
| Dutch | 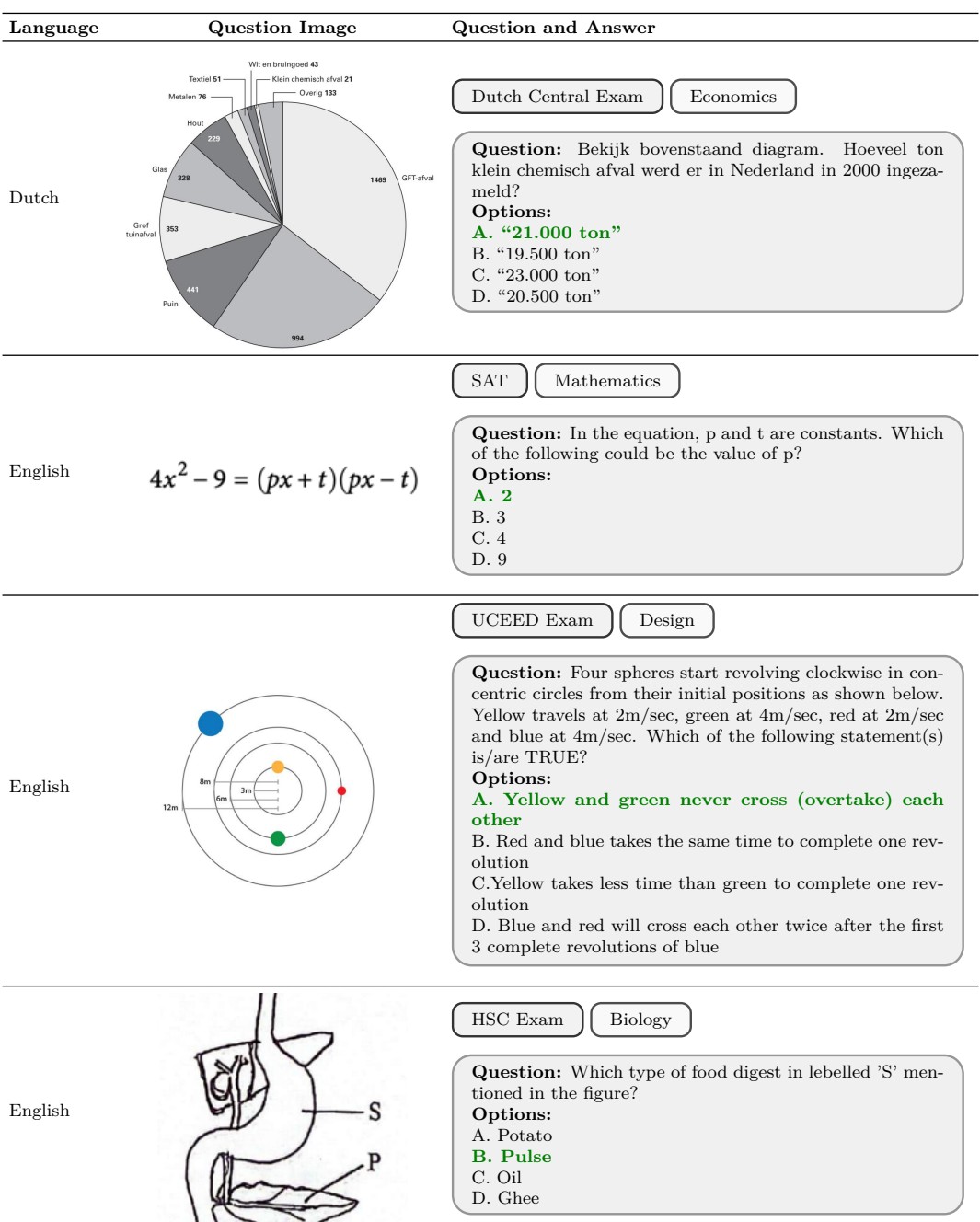 | **Dutch Central Exam**   **Economics**

**Question:** Bekijk bovenstaand diagram. Hoeveel ton klein chemisch afval werd er in Nederland in 2000 ingezameld?
**Options:**
**A. "21.000 ton"**
B. "19.500 ton"
C. "23.000 ton"
D. "20.500 ton" |
| English | $4x^2 - 9 = (px + t)(px - t)$ | **SAT**   **Mathematics**

**Question:** In the equation, p and t are constants. Which of the following could be the value of p?
**Options:**
**A. 2**
B. 3
C. 4
D. 9 |
| English | | **UCEED Exam**   **Design**

**Question:** Four spheres start revolving clockwise in concentric circles from their initial positions as shown below. Yellow travels at 2m/sec, green at 4m/sec, red at 2m/sec and blue at 4m/sec. Which of the following statement(s) is/are TRUE?
**Options:**
**A. Yellow and green never cross (overtake) each other**
B. Red and blue takes the same time to complete one revolution
C.Yellow takes less time than green to complete one revolution
D. Blue and red will cross each other twice after the first 3 complete revolutions of blue |
| English | | **HSC Exam**   **Biology**

**Question:** Which type of food digest in lebelled 'S' mentioned in the figure?
**Options:**
A. Potato
**B. Pulse**
C. Oil
D. Ghee |

| Language | Question Image | Question and Answer |
|---|---|---|

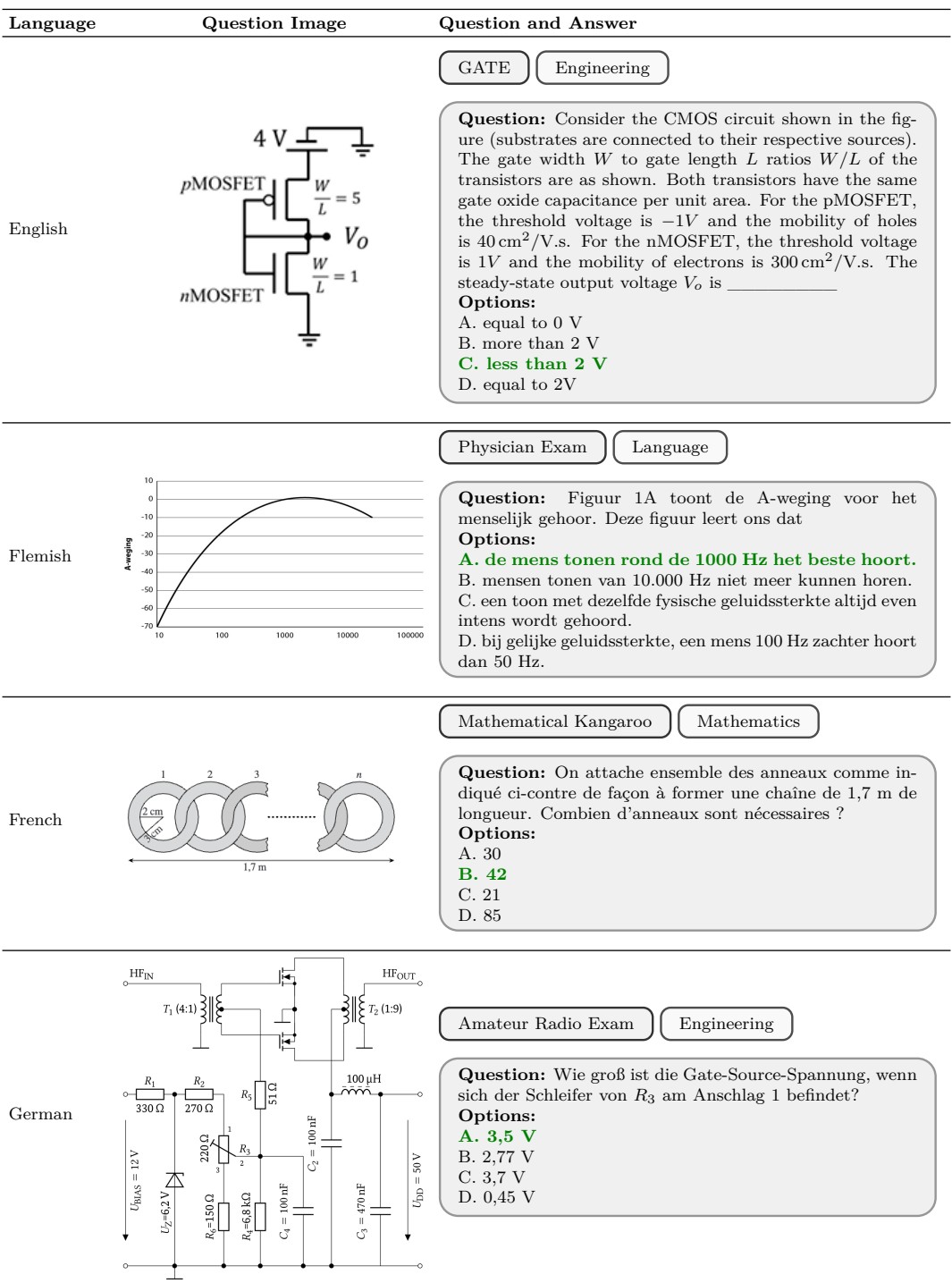

| Language | Question Image | Question and Answer |
|---|---|---|
| Hindi | 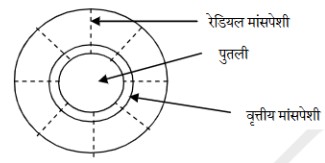 | 

**Question:** अजय अपने घर से 20 मिनट पैदल चलकर दोपहर के 3.30 बजे सिनेमा हॉल पहुँचा। वह जल्दी-जल्दी सिनेमा हॉल में घुस गया। इसे अस-पास साफ-साफ देखने में कुछ समय लगा। इस दौरान उसकी आँखों में किस तरह के परिवर्तन आए होंगे?

**Options:**
A. वृत्तीय और रेडियल मांसपेशियां शिथिल होती हैं जबकि पुतलियां संकुचित होती हैं।
**B. वृत्तीय मांसपेशियां शिथिल होती हैं, रेडियल मांसपेशियां संकुचित होती हैं और पुतलियां फैलती हैं।**
C. वृत्तीय और रेडियल मांसपेशियां संकुचित होती हैं जबकि पुतलियां फैलती हैं।
D. वृत्तीय मांसपेशियां संकुचित होती हैं, रेडियल मांसपेशियां शिथिल होती हैं और पुतलियां संकुचित होती हैं। |
| Hindi | $I$ vs $V$ graphs (a), (b), (c), (d) | JEE (Main) · Physics

**Question:** चित्र (a), (b), (c), (d) देखकर निर्धारित करें कि ये चित्र क्रमशः किन सेमीकंडक्टर डिवाइसों के अभिलक्षणांक ग्राफ हैं :
**Options:**
**A. साधारण डायोड, जीनर डायोड, सोलर सेल, LDR (लाइट डिपेंडेंट रेजिस्टेंस)**
B. जीनर डायोड, साधारण डायोड, LDR (लाइट डिपेंडेंट रेजिस्टेंस), सोलर सेल
C. सोलर सेल, LDR (लाइट डिपेंडेंट रेजिस्टेंस), जीनर डायोड, साधारण डायोड
D. जीनर डायोड, सोलर सेल, साधारण डायोड, LDR (लाइट डिपेंडेंट रेजिस्टेंस) |
| Hindi | 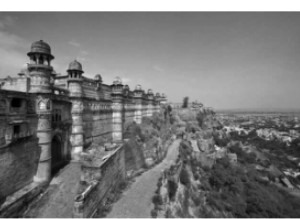 | UP-CET · Social Sciences

**Question:** चित्र में दिए किले को पहचानिये।
**Options:**
A. जोधपुर किला
**B. ग्वालियर किला**
C. लाल किला
D. आमेर किला |
| Hindi | $$\int_1^2 (x^2 - 2x + 4)^{3/2}\, dx = \frac{k}{k+5}$$ | JEE (Main) · Mathematics

**Question:** यदि इस छवि में दिखाए गए समीकरण के अनुसार, तो $k$ बराबर है :
**Options:**
A. 1
B. 2
**C. 3**
D. 4 |

| Language | Question Image | Question and Answer |
|---|---|---|

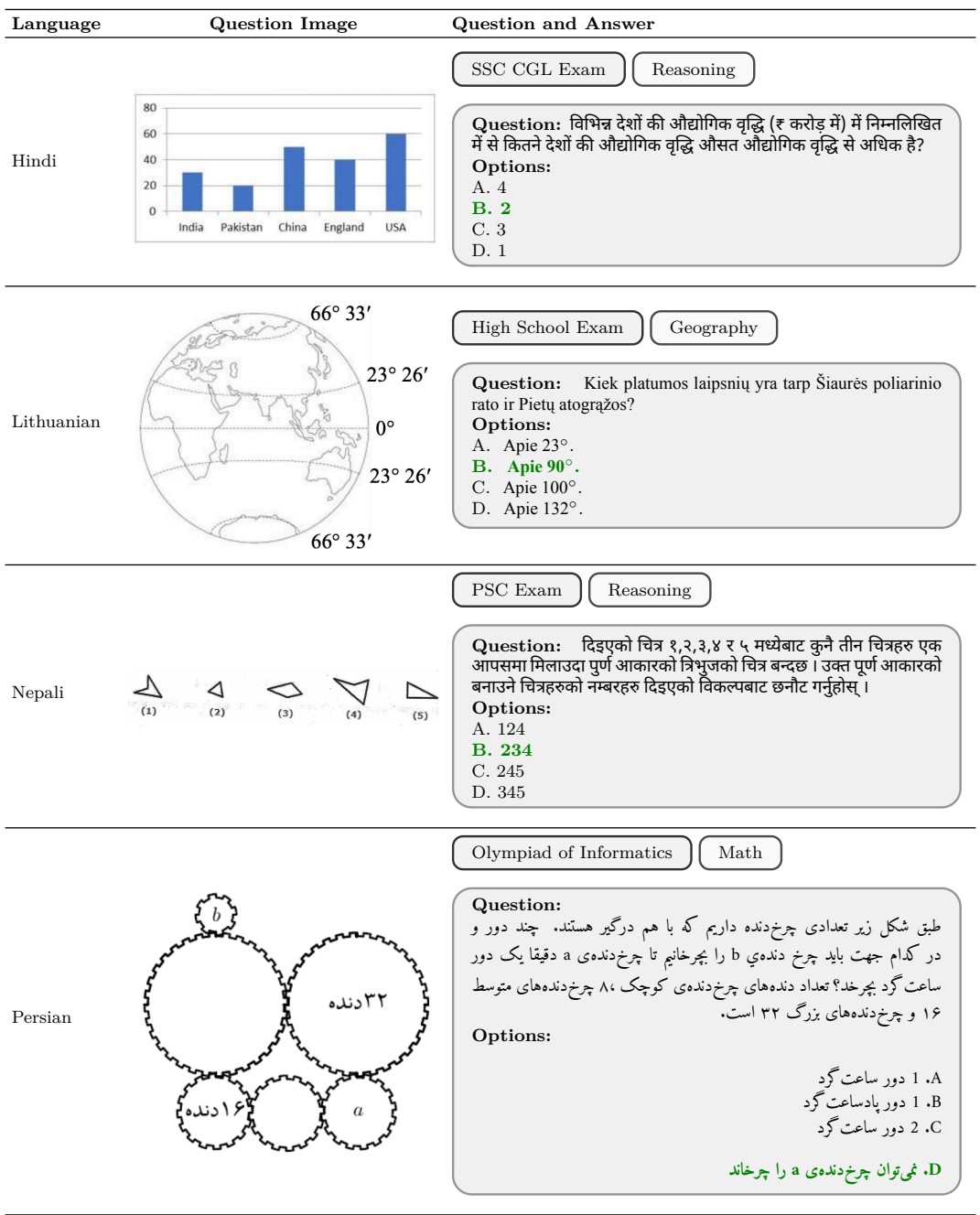

**Hindi**

**Question:** विभिन्न देशों की औद्योगिक वृद्धि (₹ करोड़ में) में निम्नलिखित में से कितने देशों की औद्योगिक वृद्धि औसत औद्योगिक वृद्धि से अधिक है?
**Options:**
A. 4
**B. 2**
C. 3
D. 1

---

**Lithuanian**

High School Exam · Geography

**Question:** Kiek platumos laipsnių yra tarp Šiaurės poliarinio rato ir Pietų atogrąžos?
**Options:**
A. Apie 23°.
**B. Apie 90°.**
C. Apie 100°.
D. Apie 132°.

---

**Nepali**

PSC Exam · Reasoning

**Question:** दिइएको चित्र १,२,३,४ र ५ मध्येबाट कुनै तीन चित्रहरु एक आपसमा मिलाउदा पूर्ण आकारको त्रिभुजको चित्र बन्दछ । उक्त पूर्ण आकारको बनाउने चित्रहरुको नम्बरहरु दिइएको विकल्पबाट छनौट गर्नुहोस् ।
**Options:**
A. 124
**B. 234**
C. 245
D. 345

---

**Persian**

Olympiad of Informatics · Math

**Question:**
طبق شکل زیر تعدادی چرخ‌دنده داریم که با هم درگیر هستند. چند دور و در کدام جهت باید چرخ دنده‌ی b را بچرخانیم تا چرخ‌دنده‌ی a دقیقاً یک دور ساعت‌گرد بچرخد؟ تعداد دنده‌های چرخ‌دنده‌ی کوچک ۸، چرخ‌دنده‌های متوسط ۱۶ و چرخ‌دنده‌های بزرگ ۳۲ است.

**Options:**

A، ۱ دور ساعت‌گرد
B، ۱ دور پادساعت‌گرد
C، ۲ دور ساعت‌گرد

**D، نمی‌توان چرخ‌دنده‌ی a را چرخاند**

| Language | Question Image | Question and Answer |
|---|---|---|

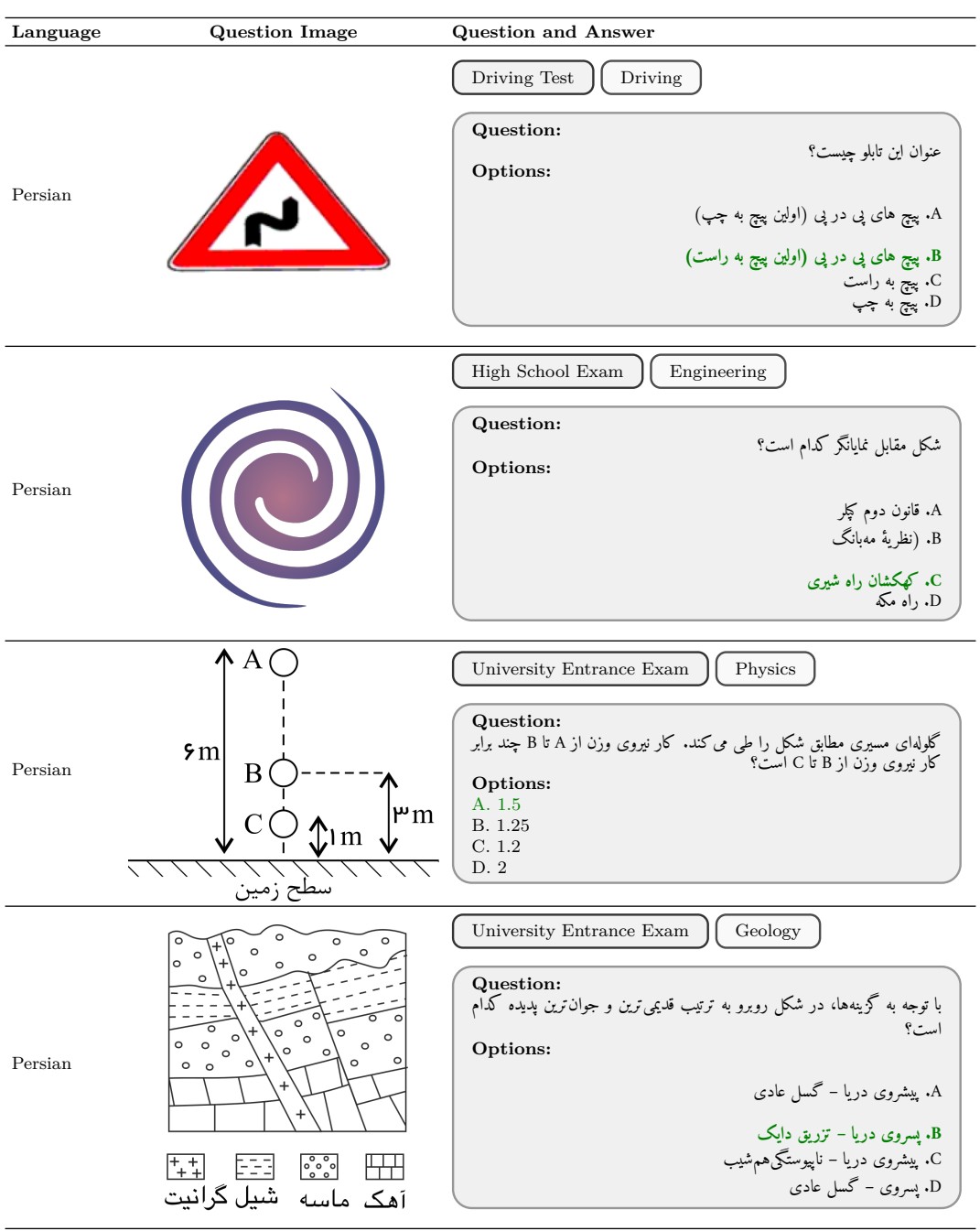

Persian

**Question:**

عنوان این تابلو چیست؟

**Options:**

A، پیچ های پی در پی (اولین پیچ به چپ)

**B، پیچ های پی در پی (اولین پیچ به راست)**
C، پیچ به راست
D، پیچ به چپ

---

Persian

High School Exam    Engineering

**Question:**

شکل مقابل نمایانگر کدام است؟

**Options:**

A، قانون دوم کپلر
B، (نظریۀ مه‌بانگ)

**C، کهکشان راه شیری**
D، راه مکه

---

Persian

University Entrance Exam    Physics

**Question:**
گلوله‌ای مسیری مطابق شکل را طی می‌کند. کار نیروی وزن از A تا B چند برابر کار نیروی وزن از B تا C است؟

**Options:**
A. 1.5
B. 1.25
C. 1.2
D. 2

---

Persian

University Entrance Exam    Geology

**Question:**
با توجه به گزینه‌ها، در شکل روبرو به ترتیب قدیمی‌ترین و جوان‌ترین پدیده کدام است؟

**Options:**

A، پیشروی دریا – گسل عادی

**B، پسروی دریا – تزریق دایک**
C، پیشروی دریا – ناپیوستگی‌هم‌شیب
D، پسروی – گسل عادی

| Language | Question Image | Question and Answer |
|---|---|---|

Portuguese

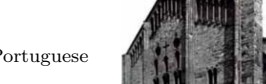 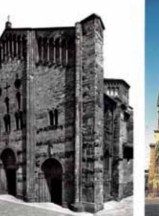 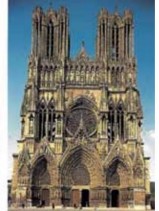

UNESP | Social Sciences

**Question:** Observe as fachadas de duas igrejas. À esquerda, a Basílica de San Michele, construída no século XII em Pavia, na Itália. À direita, a Catedral de Reims, erguida a partir do século XIII em Reims, na França.
(Georges Duby e Michel Laclotte (orgs.). História artística da Europa: a Idade Média II, 1998.)
As duas fachadas
**Options:**
**A. diferenciam-se pela pouca ornamentação de San Michele, que expressa o estilo românico, e pela monumentalidade e sofisticação de Reims.**
B. diferenciam-se pela solidez de San Michele, que simboliza a força espiritual do catolicismo, e pela carência de detalhes na sede papal em Reims.
C. igualam-se na suntuosidade e no rebuscamento arquitetônico, indicando o poderio econômico da Igreja católica.
D. diferenciam-se pela discrição de San Michele, que revela o rigor na conduta dos protestantes, e pela ostentação da riqueza católica de Reims.

---

Portuguese

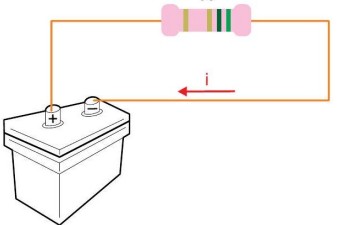

FAMERP Entrance Exam | Physics

**Question:** Quando um gerador de força eletromotriz 12 V é ligado a um resistor R de resistência $5,8\Omega$, uma corrente elétrica i de intensidade 2,0 A circula pelo circuito.
R
A resistência interna desse gerador é igual a
**Options:**
A. $0,40\Omega$.
**B. $0,20\Omega$**
C. $0,10\Omega$.
D. $0,30\Omega$.

---

Portuguese

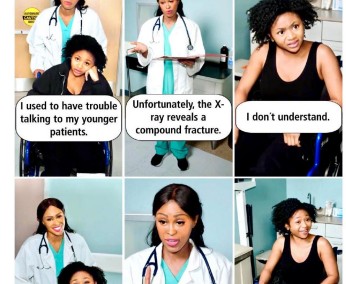

Unicamp Entrance Exam | Language

**Question:** A imagem a seguir apresenta a transcrição de um diálogo em um vídeo publicado no Instagram.
No diálogo, a principal característica da reformulação da fala da médica é a inserção de
**Options:**
A. expressões que utilizam verbos frasais para recontextualizar o tratamento da paciente.
B. abreviações de substantivos, através das quais a médica amplia as informações do caso.
C. gírias que utilizam diversas classes de palavras para especificar melhor o diagnóstico da paciente.
**D. vocábulos marcados pela oralidade, através dos quais a médica atualiza os procedimentos futuros.**

| Language | Question Image | Question and Answer |
|---|---|---|

| Language | Question Image | Question and Answer |
|---|---|---|
| Portuguese |  | **ENEM, Brazil** \| **Mathematics**

**Question:** Um segmento de reta está dividido em duas partes na proporção áurea quando o todo está para uma das partes na mesma razão em que essa parte está para a outra. Essa constante de proporcionalidade é comumente representada pela letra grega $\varphi$, e seu valor é dado pela solução positiva da equação $\varphi^2 = \varphi + 1$.
Assim como a potência $\varphi^2$, as potências superiores de $\varphi$ podem ser expressas da forma $a\varphi + b$, em que a e $b$ são inteiros positivos, como apresentado no quadro.
A potência $\varphi^7$, escrita na forma $a\varphi + b$ ( $a$ e $b$ são inteiros positivos), é
**Options:**
A. $7\varphi + 2$
B. $9\varphi + 6$
C. $11\varphi + 7$
**D.** $13\varphi + 8$ |
| Serbian |  | **Mathematical Kangaroo** \| **Mathematics**

**Question:** Колико процената површине троугла на слици је осенчено?
**Options:**
**A. 88 %**
B. 90 %
C. 85 %
D. 80 % |
| Russian |  | **Mathematical Kangaroo** \| **Mathematics**

**Question:** Каких геометрических фигур нет на рисунке?
**Options:**
A. кругов
B. все эти фигуры есть
C. прямоугольников
**D. треугольников** |
| Spanish |  | **Medicine Exam** \| **Pulmonology**

**Question:** Varón de 60 años, fumador activo, que presenta tos y expectoración diaria de años de evolución, ocasionalmente hemoptoica. En los últimos meses se añade disnea progresiva. Presenta acropaquia y en la auscultación pulmonar destacan roncus y sibilantes teleinspiratorios en pulmón izquierdo. La TC pulmonar de alta resolución se muestra en la imagen adjunta. ¿Cuál es el diagnóstico más probable?

**Choices:**
A. Carcinoma quístico.
B. Enfisema pulmonar.
C. Tuberculosis cavitada.
**D. Bronquiectasias.** |

| Language | Question Image | Question and Answer |
|---|---|---|
| Spanish |  | 

**Question:** Calcule el valor de la primera resistencia (R1)
**Options:**
A. 42 $\Omega$
B. 6 $\Omega$
**C. 12 $\Omega$**
D. 24 $\Omega$ |
| Spanish |  | High School Exam, Colombia    Biology

**Question:** En un laboratorio se estudia el comportamiento del volumen de un gas ideal al variar su temperatura, obteniendo la siguiente gráfica: Teniendo en cuenta la información de la gráfica, si la temperatura aumenta de -153 °C a -33 °C, ¿qué pasa con el volumen del gas?
**Options:**
A. Disminuye de 30 L a 25 L.
B. Disminuye de 10 L a 5 L.
C. Aumenta de 0 L a 10 L.
**D. Aumenta de 10 L a 20 L.** |
| Telugu |  | Undergraduate Exam    Chemistry

**Question:** ఇచ్చిన చిత్రంలో సమ్మేళనం యొక్క మోలార్ ద్రవ్యరాశి ఎంత?
**Options:**
A. 304.9
B. 304.4
C. 301.9
**D. 303.4** |
| Ukrainian |  | ZNO Vision    Mathematics

**Question:** Пластикові кульки радіуса 6 см зберігають у висувній шухлядці, що має форму прямокутного паралелепіпеда (див. рисунок). Якою з наведених може бути висота $h$ цієї шухлядки?
**Options:**
A. 3 см
B. 6 см
C. 10 см
**D. 13 см** |
| Ukrainian |  | Driving Test    Driving

**Question:** По якій траєкторії можна продовжити рух праворуч легковому автомобілю?
**Options:**
A. Тільки по А.
**B. Тільки по Б.**
C. По А і Б.
D. По будь-якій. |

Table 23: Structured dataset fields with descriptions used in data collection protocol.

| Field | Description |
|---|---|
| `language` | The language in which the question is written (e.g., `"en"` for English). |
| `country` | The country where the exam originated (e.g., `"United States"`). |
| `contributor_country` | The contributor's country of residence (e.g., `"Spain"`). |
| `file_name` | The internal database filename for the original exam document. |
| `source` | The URL or reference to the original exam document. |
| `license` | Licensing information of the exam (e.g., `"Unknown"` if not stated). |
| `level` | The educational level of the exam (e.g., `"University Entrance"`). |
| `category_en` | The exam subject category in English (e.g., `"Chemistry"`). |
| `category_source _lang` | The subject category as written in the original language (`language`). |
| `original_question _num` | The original question number in the source document. |
| `question` | The text of the question. |
| `options` | A list of possible answer choices. For example, `["Option A", "Option B", "Option C", "Option D"]`. |
| `answer` | The index of the correct answer (e.g., `3` for the fourth option). |
| `question_image` | The extracted diagram, graph, or table associated with the `question`. |
| `image_information` | A label indicating the importance of the `question_image` for answering the question. Possible values include:
• `"useful"` - The image provides additional clarification.
• `"essential"` - The image is necessary to answer the question. |
| `image_type` | The category of `question_image` (e.g., `"figure"`, `"graph"`, `"table"`) as described in Appendix E.1. |

