# OpenReview forum: "Kaleidoscope: In-language Exams for Massively  Multilingual Vision Evaluation"
_ICLR.cc/2026/Conference — ICLR 2026 Poster_

### Official Review · Reviewer_KzYF · 2025-10-28

**Soundness:** 4
**Presentation:** 4
**Contribution:** 3
**Rating:** 8
**Confidence:** 4

**Summary:**

This paper presents KALEIDOSCOPE, a large-scale exam-style benchmark to evaluate vision-language models across diverse languages and visual inputs. The proposed benchmark contains 18 languages, and 14 subjects with 55% requiring image understanding. The authors collect real exams via a global open-science effort. They evaluate closed (such as GPT-4o, and Claude 3.5 Sonnet) and open models (such as Qwen2.5-VL, Aya-Vision and others). After evaluation the authors describe different findings, among them, (i) all evaluated models perform substantially better on text-only questions than in multimodal ones. (ii) The type of visual data contained in exams affect performance depending if the question contains tables, diagrams, photos, etc. (iii) Performance depends on the domain of the questions. (iV) And finally the well known problem of crosslingual disparities in high vs low resource languages.

**Strengths:**

Paper Strengths:
1. The authors tackle the gap of evaluation at the intersection of multilingual and multimodal tasks. Kaleidoscope proposes in language exam questions blending image and text modalities, covering 18 languages and 8 image categories (such as diagrams, graphs, tables, etc). Compared to other benchmarks on the field, Kaleidoscope is more linguistically diverse and highly focused on multimodal evaluation.
2. The paper provides extensive evaluations by language, subject and image type.
3. Overall, the paper shows interesting insights and evaluations of open vs close models on multimodal QA , such as multimodal vs text only across models, resource gaps, scaling trends and domain disparities. These insights can be valuable for the community and future work in VLMs training.

**Weaknesses:**

Paper Weaknesses:
1. Table 1 compares closed models that use CoT prompts, while smaller models use a direct answer prompt due to CoT instability, however this is not an apples-vs-apples comparison, it would be interesting to see the performances of closed models using the direct answer generation approach. Now Qwen2.5-VL-72B is already close in performance compared to GPT-4o, with this direct comparison maybe Qwen could surpass GPT.
2. I think the paper needs to show some statistics related to the difficulty level of the questions, this will give readers an overall idea of the benchmark. For example Table 4, could have how many grad, undergrad questions each language contains and in this way there is some information (maybe not really precise) but gives an overall idea of how difficult the questions are.
3. Personally, for me it was not really clear when the authors describe that Kaleidoscope allows cultural evaluation by combining regionally sourced multimodal questions. The proposed benchmark contains different categories, so I expect in fields like STEM to contain common universal topics, and in others such as Arts & Humanities and Social Sciences to contain some cultural related topics, but the question is was this quantified in some way? like how many of those cultural related questions are containing information from their specific country-language pair?, I think that exam-questions could be more global in some situations and those will not be related to the culture of each specific country-language pair.

**Questions:**

Please refer to weaknesses for my questions and doubts. Overall I think the paper brings good contributions and really interesting insights for current open and close vision-language models that can be useful for the community. Now I'm accepting the paper, but I look forward to see the author's responses and clarifications, and sorry if I misunderstood something, after rebuttal I will revise my decision.

---

> ### Author Response · Authors · 2025-11-22
> **Response to Reviewer KzYF**
>
> We thank the reviewer for the thoughtful and constructive review, and for the positive assessment of KALEIDOSCOPE’s motivation, scope, and analyses. We especially appreciate the highlighting of  (i) the intersection of multilingual and multimodal evaluation, (ii) the breadth of languages, subjects, and image types, and (iii) the value of the open vs. closed model comparisons and the resulting insights on modality, domain, and resource-level gaps.
>
> Below we address each raised concern.
>
> **1) CoT vs. direct-answer prompting.**
>
> We agree that using CoT for closed models and direct answering for open-weight models introduces a confounding factor. As shown in Appendix D.2 (Table 7), several open-weight models struggled to follow the CoT prompt, whereas the direct-answer prompt produced higher accuracy. Closed models (e.g., GPT-4o, Claude, Gemini) worked well with CoT, and prior work shows that CoT generally yields higher accuracy for stronger models; we report the strongest results (as further discussed in Appendix D.2).
>
> Following the reviewer’s suggestion, we added results for GPT-4o using the same direct prompt used for open models. Table 8 (also included below) shows that GPT-4o performs worse with the direct prompt than with CoT, supporting our choice to use the best-performing strategy in the main paper. Unfortunately, after submission, Claude 3.5 Sonnet and Gemini 1.5 Pro became unavailable, and re-running both prompting strategies on their updated versions would entail substantial additional cost. As with GPT-4o, we expect these models would likewise perform better under CoT.
>
> **2) Difficulty levels and question statistics.**
>
> We acknowledge that exposing difficulty-related statistics is helpful for interpreting the benchmark. We fully agree, and we have added a difficulty analysis to address this concern. The original metadata contained nine “level’’ categories mixing educational stages with exam types; to enable cross-language comparability, we standardized these into a four-tier taxonomy (Basic, Intermediate, Advanced, Expert). The mapping was created by sampling examples from each category and determining the typical human age and educational stage associated with those questions, providing a more consistent and interpretable difficulty framework.
>
> Table 4 (above and in the paper) now reports difficulty distributions for each language, in both percentages and raw counts. As the reviewer suggested, this shows that differences in observed accuracy can be partly explained by differences in the underlying difficulty of available exams. We discuss this in Appendix B.2 and emphasize that performance differences reflect both language and content variation.
>
> **3) Cultural evaluation.**
>
> We appreciate the reviewer’s question and agree that cultural evaluation must be defined carefully, especially since domains like STEM include many seemingly universal topics. Our claim is not that every question is culturally specific, but that real, in-language exams inevitably embed cultural context through curricula, framing, examples, notation, and assumptions about background knowledge. This cultural grounding is preserved in KALEIDOSCOPE because each item is sourced from a specific national or regional exam and is tagged with country and language in the metadata.
>
> We did not perform an explicit annotation pass to quantify “culturally dependent” vs. “culturally agnostic” items, as this would require substantial expert labeling. However, prior work (e.g., Global MMLU, Singh et al. 2025) shows that culturally sensitive questions constitute a meaningful portion of real exam datasets and significantly affect model rankings. KALEIDOSCOPE follows this same design philosophy by using authentic, locally authored exams rather than translations.
>
> We agree that an explicit cultural-sensitivity annotation layer, similar to the one in Global , would strengthen the benchmark, and we see this as a natural direction for future extensions of KALEIDOSCOPE.
>
> We thank the reviewer for pointing towards these additions and we hope to have addressed the reviewer’s requests.

---

> > ### Comment · Reviewer_KzYF · 2025-11-24
> >
> > Thanks to the authors for their responses and clarifications, everything has been solved from my side. I will give my final score at the end of the rebuttal period. Good luck with the paper !

---

### Official Review · Reviewer_PSVd · 2025-11-01

**Soundness:** 3
**Presentation:** 3
**Contribution:** 4
**Rating:** 6
**Confidence:** 3

**Summary:**

Kaleidoscope is a new large-scale benchmark designed to evaluate multimodal + multilingual reasoning in vision-language models (VLMs), claiming to be the largest benchmark of its sort. It contains around 21k multiple choice questions (MCQ) across 18 languages and 14 subjects, with about half including images. The dataset is built from real-world, in-language (that is, not machine translated) educational exams contributed by native speakers from more than 20 countries. The authors evaluate both commercial API and open-source models under standardized protocols. Their analyses show large modality gaps (text-only vs. image+text), STEM-specific difficulty (compared to humanities subjects), and cross-lingual disparities (Latin script advantage over non-Latin). Despite some limitations (language balance, difficulty calibration, and reliance on MCQA format), Kaleidoscope aims to serve as a diagnostic benchmark to reveal weaknesses in a niche but increasingly important task where both multimodal and multilingual reasoning are required.

**Strengths:**

- Most importantly, the core contribution is the dataset itself. It targets a niche area that is difficult to produce synthetically (e.g., LLM-generated).

- The scope of this benchmark (# languages, # subjects, and a substantial proportion of image-based questions) is impressive. Having personally experienced very large, multi-national collaborations, I am genuinely impressed by the coordination and collective effort that made this possible.

- Although I wish the paper provided more details about the data collection and quality control processes (including license verification), my general impression is that the authors have made a serious effort to standardize collection and validation across languages and annotator teams.

- Each item includes detailed metadata (17 fields including country, subject, image type, and image importance), enabling detailed diagnostic analyses. This metadata can itself serve as a valuable resource for future research.

- The paper provides interesting empirical insights that can guide the development of future model architectures or training strategies: the modality gap, STEM vs. humanities performance disparity, and Latin vs. non-Latin script biases.

**Weaknesses:**

- Each language has independent questions rather than parallel translations, so performance differences reflect a mixture of linguistic and content variation _(I note that the authors were fair to list this point in the limitations section)_. This complicates direct interpretation of multilingual gaps. It would be helpful if the authors could provide some indication or proxy of relative difficulty across languages.

- Related to the above comment, I was curious to see a translation baseline. The authors deliberately avoid machine-translated versions, but a translated-to-English baseline would, to some extent, help isolate the effect of language from content or domain differences.

- I also wish the paper included cross-benchmark context. Table 2 reports results only on Kaleidoscope, without comparisons to other multimodal or multilingual benchmarks (e.g., MMMU, SEED-Bench, M3Exam). Without this, it is difficult to assess the dataset's relative difficulty and unique diagnostic value.

- Intended-use guidance (e.g., do’s and don’ts) would have been nice. Benchmarks are often misused or overinterpreted; an explicit section clarifying what Kaleidoscope can and cannot measure would help practitioners use it responsibly.

**Questions:**

See the above weakness sections. Besides, I have a couple of questions / suggestions.

- Given the benchmark’s collaborative nature, do you plan to release a contribution protocol or submission guideline so external contributors can add new languages or exams in future versions? I think this could turn Kaleidoscope into a true community-science platform and make it a living benchmark.

- The appendix is lengthy and dense. I'd like to suggest that the authors add an overview or roadmap at the beginning of the appendices. This would increase readability.

---

> ### Author Response · Authors · 2025-11-22
> **Response to Reviewer PSVd**
>
> We thank the reviewer for the thoughtful and constructive assessment. We appreciate the recognition of the dataset as the core contribution, the scale and coordination of the multi-national collaboration, the standardized collection and validation process, the value of the rich metadata, and the relevance of our empirical findings for future model design.
>
> **1) Independent questions vs. parallel translations & relative difficulty across languages.**
>
> We agree that using independent in-language exams means cross-language gaps reflect a mix of language, curriculum, and difficulty. Our goal was to preserve the authenticity of real exams rather than impose artificial parallelism through rewritten questions.
> To provide a proxy for difficulty, we have added a new difficulty metadata and analysis. The original nine “level’’ categories mixed educational stages with exam types, so we standardized them into a four-tier taxonomy (Basic, Intermediate, Advanced, Expert). The mapping was created by sampling examples from each category and estimating the typical age and educational stage associated with those questions, yielding a more consistent and interpretable difficulty framework.
>
> Table 4 now reports difficulty distributions for each language (percentages and counts). As the reviewer suggested, this shows that differences in observed accuracy can partly reflect differences in exam difficulty. We discuss this in Appendix B.2 and emphasize that performance differences reflect both language and content variation.
>
> **2) Translated-to-English baseline.**
>
> We agree that translation-to-English baselines are often useful in multilingual benchmarks. However, KALEIDOSCOPE’s multimodal image–text nature makes such a baseline non-trivial: many items contain essential textual cues embedded in images, and producing an English-only version would require OCR and image editing across 18 languages and scripts, introducing significant and hard-to-control noise. Prior work (e.g., M3Exam[Zhang et al., NeurIPS 2023]) likewise notes that translation can alter difficulty, remove culturally grounded knowledge, and reduce comparability with native exam data. For these reasons, we do not include a translated-to-English version in the main benchmark and instead view it as complementary future work that KALEIDOSCOPE enables.
>
> **3) Cross-benchmark context and relative difficulty.**
>
> We appreciate the reviewer’s suggestion. We have added a brief cross-benchmark comparison situating Kaleidoscope alongside MMMU and MMMU-Pro on the overlapping set of models. The new Table 20 makes two patterns clear:
> - Relative difficulty. Kaleidoscope is a non-trivial benchmark: its overall accuracy sits in roughly the same difficulty regime as MMMU/MMMU-Pro. Across models, Kaleidoscope typically falls between MMMU-Validation and MMMU-Pro, though the exact ordering varies by model.
> - Complementary diagnostic value. Despite similar accuracies, Kaleidoscope measures a distinct axis of capability. Unlike MMMU/MMMU-Pro, which rely heavily on English or translated content, Kaleidoscope consists of in-language exam questions from 18 languages and cultures. As commented in Appendix D.7, models with comparable MMMU scores diverge more strongly on Kaleidoscope, highlighting the dataset’s ability to reveal cross-lingual and culturally dependent weaknesses that MMA benchmarks cannot detect.
>
> **4) Intended-use.**
>
> We agree that explicit intended-use guidance is important for preventing over-interpretation. Our current Limitations section already notes the constraints of KALEIDOSCOPE, such as its language imbalance, uncontrolled difficulty distribution, and the use of MCQA with known issues (e.g., guessing, distractor sensitivity, and format-compliance effects).
>
> We have included a concise “Intended Use” paragraph (placed at the end of the Limitations, for now in Appendix A) that clearly specifies what KALEIDOSCOPE is and is not designed to measure. In particular, we will highlight that the benchmark is meant for evaluating cross-lingual multimodal reasoning under exam-style conditions, diagnosing modality gaps, and analyzing performance by subject, image category, and language family via our metadata. We will also state explicitly that KALEIDOSCOPE is not suitable for assessing free-form generation, long-context interaction, or for any training/fine-tuning purposes.

---

> > ### Author Response · Authors · 2025-11-22
> > **Response to Reviewer PSVd - Part 2**
> >
> > **5) “Living benchmark” and external contributions.**
> >
> > We appreciate the interest in turning KALEIDOSCOPE into a community-driven benchmark, this aligns closely with the collaborative spirit in which the dataset was initially collected. We share the enthusiasm about enabling future contributions, but designing a fully open submission pipeline is beyond the scope of the present study. The main practical challenge is maintaining the level of quality control we applied during data collection: sourcing legally usable exams, verifying metadata consistency, performing multi-stage validation, and ensuring cross-language comparability. Replicating this standard in an open, decentralized workflow is non-trivial.
> >
> > To balance openness with reliability, our current plan is to release the complete protocol we used for sourcing, licensing, annotating, and validating exams. This allows external groups to create KALEIDOSCOPE-compatible extensions following the same methodological guidelines.
> >
> > **6) Appendix roadmap and data-collection/licensing details.**
> >
> > We have added a short table of contents at the beginning of the appendix to provide a clearer roadmap and improve overall navigability, as well we have reorganized the complete appendix section for clarity. Regarding licensing and quality control, we note that Appendix B.1 already describes our licensing policy, and we have now expanded that section to provide additional detail. Specifically, we added the following subsection: *To further guarantee compliance, we employ a two-stage validation process in which two blinded annotators independently verify the license of each exam included in our dataset. Only items that pass both validations are included in the final dataset.*
> >
> > We believe these additions directly address the reviewer’s request for greater transparency and strengthen the documentation of our data-collection process.

---

### Official Review · Reviewer_6PmD · 2025-11-01

**Soundness:** 4
**Presentation:** 4
**Contribution:** 4
**Rating:** 6
**Confidence:** 4

**Summary:**

This paper addresses the problem that VLM evaluation is heavily dominated by English-language and Western-centric benchmarks. To overcome the limitations of simply translating existing datasets, the authors introduce KALEIDOSCOPE, a new benchmark composed of over 20,000 "in-language" multiple-choice questions from real-world exams. The benchmark spans 18 languages and 14 different subjects, with a significant portion (55%) of the questions requiring multimodal reasoning (image and text). The dataset was constructed through a large-scale open science collaboration with contributors from around the world, ensuring linguistic and cultural authenticity. The authors evaluate a wide range of state-of-the-art VLMs on KALEIDOSCOPE, including both large closed-source models and smaller open-weight models. Their findings reveal substantial performance gaps: all models struggle more with multimodal questions than text-only ones, performance is significantly weaker on low-resource languages and questions from STEM subjects, and models exhibit a clear bias towards Latin scripts.

**Strengths:**

- The motivation for this work is very clear and addresses a critical gap in the field. As vision-language models become more widespread, it is essential to evaluate them beyond English. The paper makes a strong case against the "translate-test" paradigm and for the necessity of "in-language" data that preserves cultural and educational context.

- The design and scale of the KALEIDOSCOPE benchmark is good. It is a comprehensive resource, covering 18 languages and a wide range of academic subjects. The use of a multiple-choice question format provides a structured and scalable evaluation framework. The inclusion of rich metadata for each question, such as image type and educational level, is also a very valuable feature that enables fine-grained analysis.

- The collaborative, open-science approach to data collection is particularly commendable. By involving a diverse group of researchers worldwide, the authors have created a more authentic and representative dataset than would be possible with automated translation. This methodology directly confronts the known biases in dataset creation and should be seen as a model for future work in this area.

- The paper's analysis is cmprehensive and provides several insightful findings that go beyond a simple ranking of models. The clear performance gap between text-only and multimodal questions is quantified well. Furthermore, the analysis of performance by image type (e.g., diagrams vs. photos), by subject (STEM vs. humanities), and by script (Latin vs. non-Latin) offers a deep look into the specific weaknesses of current VLMs.

**Weaknesses:**

- The reliance on a Multiple-Choice Question Answering (MCQA) format, while practical for evaluation, comes with inherent limitations. Models can sometimes guess the correct answer or exploit statistical cues in the options without genuine understanding. The authors acknowledge this point in the appendix, but this is a fundamental aspect of their evaluation design and should be more prominently discussed in the main paper's limitations section (Section A).

- The data collection process, despite being a large-scale effort, could benefit from more transparency regarding the source material's representativeness and difficulty. The paper notes that performance in some languages (like Lithuanian) may be higher due to the subject matter of the available exams. This suggests that the benchmark measures a combination of model capability and the specific difficulty of the sourced exams, which can be hard to disentangle. A more detailed discussion of the potential imbalances in difficulty across languages and subjects would be helpful.

- A potential confounding factor in the experimental results is the use of two different prompting strategies for closed and open-weight models (Chain-of-Thought for large models, direct answering for smaller ones). The authors justify this as a pragmatic choice, which is understandable. However, this means the comparison between these two groups of models is not perfectly controlled. The performance gap could be influenced by the different evaluation protocols. This should be more clearly stated and discussed as a limitation of the current experimental setup.

- The analysis of format errors (Section 5.3 and Table 1) is interesting, but its presentation could be more integrated. For example, the fact that GPT-4o's accuracy improves significantly when format errors are excluded is a very important finding that highlights a specific weakness of that model. This point, and the high refusal rate of Pangea, could be woven more directly into the main results discussion to give a clearer picture of model behavior.

**Questions:**

N/A

---

> ### Author Response · Authors · 2025-11-22
> **Response to Reviewer 6PmD**
>
> We thank the reviewer for the positive evaluation of our work. We are glad that the motivation behind critiquing translate-test, the design and scale of KALEIDOSCOPE, and the value of its metadata and analyses were clearly recognized. These were core goals of the project, and we appreciate the reviewer’s appreciation of them.
>
> Below we address the raised concerns and specify the exact changes that will be incorporated into the camera-ready version.
>
> **1) MCQA format and its limitations**
>
> We agree that the limitations of the MCQA format should be more prominent in the main paper. As discussed in Appendix C.3, MCQA is widely used in human exam settings and it enables standardized evaluation across all 18 languages on our dataset. However, as the reviewer notes, it also introduces risks such as guessing or exploiting statistical cues rather than demonstrating true understanding.
>
> We have reorganized the appendix to make the discussion of evaluation format more accessible. Additionally, for the camera-ready version, we will include the limitations section in the main paper, where we directly discuss the constraints of the MCQA format and its implications for interpreting model performance.
>
> **2) Representativeness and difficulty of the sourced exams**
>
> We have added a difficulty analysis to address this concern. The original metadata contained nine “level’’ categories mixing educational stages with exam types; to enable cross-language comparability, we standardized these into a four-tier taxonomy (Basic, Intermediate, Advanced, Expert). The mapping was created by sampling examples from each category and determining the typical human age and educational stage associated with those questions, providing a more consistent and interpretable difficulty framework.
>
> Table 4 (in the comment above) now reports difficulty distributions for each language, in both percentages and raw counts. As the reviewer suggested, this shows that differences in observed accuracy can be partly explained by differences in the underlying difficulty of available exams. We discuss this in Appendix B.2 and emphasize that performance differences reflect both language and content variation.
>
> **3) Different prompting strategies for closed vs open-weight models**
>
> We agree that using CoT for closed models and direct answering for open-weight models introduces a confounding factor. As shown in Appendix D.2 (Table 7), several open-weight models struggled to follow the CoT prompt, whereas the direct-answer prompt produced higher accuracy. Closed models (e.g., GPT-4o, Claude, Gemini) worked well with CoT, and prior work shows that CoT generally yields higher accuracy for stronger models; we report the strongest results (as further discussed in Appendix D.2).
>
> Following the reviewer’s suggestion, we added results for GPT-4o using the same direct prompt used for open models. Table 8 (also included below) shows that GPT-4o performs worse with the direct prompt than with CoT, supporting our choice to use the best-performing strategy in the main paper. Unfortunately, after submission, Claude 3.5 Sonnet and Gemini 1.5 Pro became unavailable, and re-running both prompting strategies on their updated versions would entail substantial additional cost. As with GPT-4o, we expect these models would likewise perform better under CoT.
>
> **4) Integration of format-error and refusal analysis into the main results**
>
> We agree that this analysis should be integrated more directly into the main narrative. In Section 5 (Results), we will directly reference Table 1, emphasizing that:
> *“GPT-4o’s accuracy is underestimated when format-error cases are included, and Pangea’s notably high refusal rate represents a distinct behavioral pattern relevant for interpreting accuracy.”*
>
> We thank the reviewer again for the thoughtful feedback. We believe these clarifications and additions, primarily in the Limitations and Results discussions, will directly address the concerns raised while preserving the core contributions of KALEIDOSCOPE.

---

### Author Response · Authors · 2025-11-22
**New Tables Added to the Paper**

The following tables were added to the paper.

# Difficulty Levels
**Table 4: Difficulty distribution across languages in KALEIDOSCOPE.**
Each cell shows percentages with raw counts.

| **Language** | **Basic** | **Intermediate** | **Advanced** | **Expert** | **Total** |
|--------------|-----------|------------------|--------------|------------|-----------|
| Arabic | 0.0% (0) | 100.0% (382) | 0.0% (0) | 0.0% (0) | 382 |
| Bengali | 0.0% (0) | 72.6% (581) | 18.0% (144) | 9.4% (75) | 800 |
| German | 0.0% (0) | 0.0% (0) | 100.0% (722) | 0.0% (0) | 722 |
| English | 0.0% (0) | 65.4% (1,065) | 17.0% (277) | 17.6% (286) | 1,628 |
| Spanish | 0.0% (0) | 58.9% (873) | 3.8% (56) | 37.3% (553) | 1,482 |
| Persian | 0.0% (0) | 76.3% (1,526) | 23.7% (474) | 0.0% (0) | 2,000 |
| French | 0.0% (0) | 100.0% (762) | 0.0% (0) | 0.0% (0) | 762 |
| Hindi | 13.1% (248) | 28.6% (540) | 21.2% (399) | 37.1% (699) | 1,886 |
| Croatian | 0.0% (0) | 100.0% (324) | 0.0% (0) | 0.0% (0) | 324 |
| Hungarian | 0.0% (0) | 100.0% (1,120) | 0.0% (0) | 0.0% (0) | 1,120 |
| Lithuanian | 0.0% (0) | 100.0% (680) | 0.0% (0) | 0.0% (0) | 680 |
| Nepali | 0.0% (0) | 100.0% (126) | 0.0% (0) | 0.0% (0) | 126 |
| Dutch | 0.0% (0) | 24.2% (246) | 75.8% (772) | 0.0% (0) | 1,018 |
| Portuguese | 0.0% (0) | 0.0% (0) | 100.0% (2,000) | 0.0% (0) | 2,000 |
| Russian | 0.0% (0) | 100.0% (1,744) | 0.0% (0) | 0.0% (0) | 1,744 |
| Serbian | 0.0% (0) | 100.0% (2,000) | 0.0% (0) | 0.0% (0) | 2,000 |
| Telugu | 0.0% (0) | 0.0% (0) | 100.0% (1,000) | 0.0% (0) | 1,000 |
| Ukrainian | 35.3% (437) | 0.0% (0) | 64.7% (800) | 0.0% (0) | 1,237 |
| **All languages** | **3.3% (685)** | **62.8% (12,869)** | **39.5% (8,094)** | **9.4% (1,929)** | **20,911** |

# Closed Models Direct Prompt
**Table 8: Comparison of different prompting strategies on GPT-4o.**
Results are shown disaggregated by language. Global accuracy, valid answer accuracy and format error rate are reported.

| **Language** | **Direct: Total Acc.** | **Direct: Valid Acc.** | **Direct: F.E.** | **CoT: Total Acc.** | **CoT: Valid Acc.** |**CoT: F.E.** |
|--------------|---------|---------|-----|---------|---------|-----|
| Arabic | 49.7 | 50.4 | 1.3 | 52.9 | **57.7** | 8.4 |
| Bengali | 57.4 | 57.5 | 0.2 | 65.6 | **67.3** | 2.5 |
| Croatian | 33.3 | 33.8 | 1.2 | 49.7 | **52.6** | 5.6 |
| Dutch | 57.1 | 58.1 | 1.8 | 58.9 | **62.4** | 5.6 |
| English | 63.9 | 64.0 | 0.1 | 60.8 | **73.4** | 17.1 |
| French | 37.7 | 37.7 | 0.1 | 61.8 | **64.6** | 4.3 |
| German | 70.8 | 70.8 | 0.0 | 71.6 | **72.6** | 1.4 |
| Hindi | 48.6 | 48.7 | 0.2 | 60.1 | **64.0** | 6.1 |
| Hungarian | 34.0 | 34.4 | 1.1 | 47.2 | **50.6** | 6.7 |
| Lithuanian | 83.4 | 83.4 | 0.0 | 86.5 | **88.4** | 2.2 |
| Nepali | 23.8 | **24.0** | 0.8 | 19.0 | 20.5 | 7.1 |
| Persian | 40.2 | 40.3 | 0.3 | 47.0 | **47.9** | 2.0 |
| Portuguese | 75.4 | 75.4 | 0.0 | 82.6 | **85.2** | 3.0 |
| Russian | 36.6 | 36.6 | 0.1 | 51.5 | **54.5** | 5.5 |
| Serbian | 33.1 | 33.4 | 1.0 | 47.0 | **52.6** | 10.6 |
| Spanish | 75.4 | 75.5 | 0.2 | 77.7 | **80.1** | 3.0 |
| Telugu | 42.1 | 44.3 | 4.9 | 41.6 | **47.9** | 13.2 |
| Ukrainian | 71.4 | 72.2 | 1.1 | 68.1 | **75.3** | 9.5 |
| **Overall** | 51.9 | 52.2 | 0.7 | 58.3 | **62.1** | 6.5|

# Cross-Benchmark Evaluation
**Table 20: Cross-benchmark comparison on KALEIDOSCOPE, MMMU-Pro, and MMMU.** Results
are reported as overall accuracy (%). KALEIDOSCOPE values use valid accuracy. Asterisks (*)
indicate results provided directly by the original authors.
| Model | Kaleidoscope (Valid Acc.) | MMMU-Pro (Acc.) | MMMU Validation (Acc.) |
|-------|---------------------------|-----------------|------------------------|
| Claude 3.5 Sonnet | 63.87 | 51.5 | 68.3 |
| Gemini 1.5 Pro | 62.95 | 46.9 | 65.8 |
| GPT-4o | 62.10 | 51.9 | 69.1 |
| | | | |
| Qwen2.5-VL-72B | 53.00 | 46.2 | 64.5 |
| | | | |
| Qwen2.5-VL-32B | 48.64 | N/A | N/A |
| Aya-Vision-32B | 39.66 | 45.11* | N/A |
| | | | |
| Aya-Vision-8B | 35.11 | 39.9* | N/A |
| Qwen2.5-VL-7B | 39.60 | 38.3* | 58.6* |
| Qwen2.5-VL-3B | 35.63 | 31.6* | 53.1* |
| | | | |
| Molmo-7B-D | 32.88 | N/A | N/A |
| Pangea-7B | 34.02 | N/A | N/A |

*Results are reported as overall accuracy (%). Kaleidoscope values use valid accuracy. Asterisks indicate results from original authors.*

---

### Meta-Review · Area_Chair_wu1A · 2026-01-09

**Summary:**

- For evaluation at the intersection of multilingual and multimodal tasks,  Kaleidoscope is proposed, covering 18 languages and 8 image categories. Compared to other benchmarks on the field, Kaleidoscope is more linguistically diverse and highly focused on multimodal evaluation.
- The paper provides extensive evaluations by language, subject and image type.
Overall, the paper shows interesting insights and evaluations of open vs close models on multimodal QA , such as multimodal vs text only across models, resource gaps, scaling trends and domain disparities. These insights can be valuable for the community and future work in VLMs training.

**Reviewer Concerns:**

The following concerns have been well addressed in the rebuttal.

- CoT vs. direct-answer prompting created unfairness.
- The joint analysis of difficulty levels and question statistics is missing.
- Translated-to-English baseline is missing.
- explicit intended-use guidance is missing.

**Reviewer Scores:**

The reviewers would have maintained their high sores.

---

### Decision · Program_Chairs · 2026-01-26

Accept (Poster)